# Sympathetic neuropeptide Y protects from obesity by sustaining thermogenic fat

Yitao Zhu[1], Lu Yao[1], Ana L. Gallo-Ferraz[2], Bruna Bombassaro[2], Marcela R. Simões[2], Ichitaro Abe[3,4], Jing Chen[5], Gitalee Sarker[1], Alessandro Ciccarelli[6], Linna Zhou[7], Carl Lee[8], Davi Sidarta-Oliveira[1], Noelia Martínez-Sánchez[9], Michael L. Dustin[8], Cheng Zhan[10], Tamas L. Horvath[11], Licio A. Velloso[2], Shingo Kajimura[3] & Ana I. Domingos[1 ✉]

Human mutations in neuropeptide Y (NPY) have been linked to high body mass index but not altered dietary patterns[1]. Here we uncover the mechanism by which NPY in sympathetic neurons[2,3] protects from obesity. Imaging of cleared mouse brown and white adipose tissue (BAT and WAT, respectively) established that NPY+ sympathetic axons are a smaller subset that mostly maps to the perivasculature; analysis of single-cell RNA sequencing datasets identified mural cells as the main NPY-responsive cells in adipose tissues. We show that NPY sustains the proliferation of mural cells, which are a source of thermogenic adipocytes in both BAT and WAT[4–6]. We found that diet-induced obesity leads to neuropathy of NPY+ axons and concomitant depletion of mural cells. This defect was replicated in mice with NPY abrogated from sympathetic neurons. The loss of NPY in sympathetic neurons whitened interscapular BAT, reducing its thermogenic ability and decreasing energy expenditure before the onset of obesity. It also caused adult-onset obesity of mice fed on a regular chow diet and rendered them more susceptible to diet-induced obesity without increasing food consumption. Our results indicate that, relative to central NPY, peripheral NPY produced by sympathetic nerves has the opposite effect on body weight by sustaining energy expenditure independently of food intake.

Sympathetic nerves within both BAT and WAT release noradrenaline locally to trigger lipolysis and thermogenesis[7,8]. NPY is coreleased with noradrenaline[9], but its specific role in adipose tissue remains unclear. Although many reports indicate that NPY in the brain stimulates appetite[10,11], knockout of NPY has no effect on daily food consumption[12,13]. In addition, mice without NPY receptors develop late-onset obesity despite eating less[14–16]. In humans, mutations in NPY have been linked to high body mass index (BMI) but not to an altered dietary pattern (Extended Data Fig. 1m) (https://hugeamp.org/gene.html?gene=NPY)[1]. These findings suggest contrasting roles for NPY from the brain and sympathetic nervous system in the regulation of body weight. To test this hypothesis we used animal models in which NPY was removed from sympathetic neurons while remaining intact in the brain.

## NPY+ sympathetic innervation in WAT and BAT

The public single-cell RNA-sequencing (scRNA-seq) dataset of sympathetic ganglia[17] shows that only two out of five clusters of sympathetic neurons express elevated levels of NPY (Extended Data Fig. 1a–c). This proportion of NPY+ sympathetic neurons, approximately 40%, was confirmed by immunofluorescent costaining of NPY and the sympathetic neuron marker tyrosine hydroxylase (TH) in superior cervical ganglia and stellate ganglia (Extended Data Fig. 1d–f). A similar TH/NPY overlap was observed in sympathetic axon bundles dissected from inguinal white adipose tissue (iWAT; Fig. 1a). In iWAT, NPY+ axons are associated with the vasculature and form a plexus around the vessels (Fig. 1b).

To determine the distribution of TH+ and NPY+ axons throughout adipose tissue, we performed tissue clearing of iWAT[18] and immunolabelled for TH, NPY and CD31, the last of which marks the vascular endothelium. We confirmed that NPY+ axons are a subgroup of sympathetic axons (Fig. 1c,d). Quantifying the images, we found that TH+NPY+ axons make up 32.4% of all labelled axons whereas 59.3% are TH+NPY− (Fig. 1e,f). The images also indicate that NPY+ axons preferentially innervate CD31+ endothelium, and this was confirmed by quantification of images (Fig. 1g). Consistent with iWAT, in interscapular brown adipose tissue (iBAT), NPY+ axons mainly innervate the vasculature (Fig. 1h).

To further determine which vessel branches are innervated by NPY+ axons, we quantified light-sheet images (Extended Data Fig. 1g) and found that the density of NPY+ axons is highest in fifth-order

[1]Department of Physiology, Anatomy and Genetics, University of Oxford, Oxford, UK. [2]Laboratory of Cell Signaling, Obesity and Comorbidities Research Center, University of Campinas, Campinas, Brazil. [3]Beth Israel Deaconess Medical Center, Division of Endocrinology, Diabetes & Metabolism, Harvard Medical School, Boston, MA, USA. [4]Department of Cardiology and Clinical Examination, Oita University, Faculty of Medicine, Oita, Japan. [5]School of Sport Science, Beijing Sport University, Beijing, China. [6]Advanced Light Microscopy, The Francis Crick Institute, London, UK. [7]Ludwig Institute for Cancer Research, Nuffield Department of Medicine, University of Oxford, Oxford, UK. [8]Kennedy Institute of Rheumatology, University of Oxford, Oxford, UK. [9]Oxford Centre for Diabetes, Endocrinology and Metabolism Radcliffe Department of Medicine, University of Oxford, Oxford, UK. [10]Department of Haematology, Division of Life Sciences and Medicine, University of Science and Technology of China, Hefei, China. [11]Department of Obstetrics/Gynecology and Reproductive Sciences, Yale University School of Medicine, New Haven, CT, USA. ✉e-mail: ana.domingos@dpag.ox.ac.uk

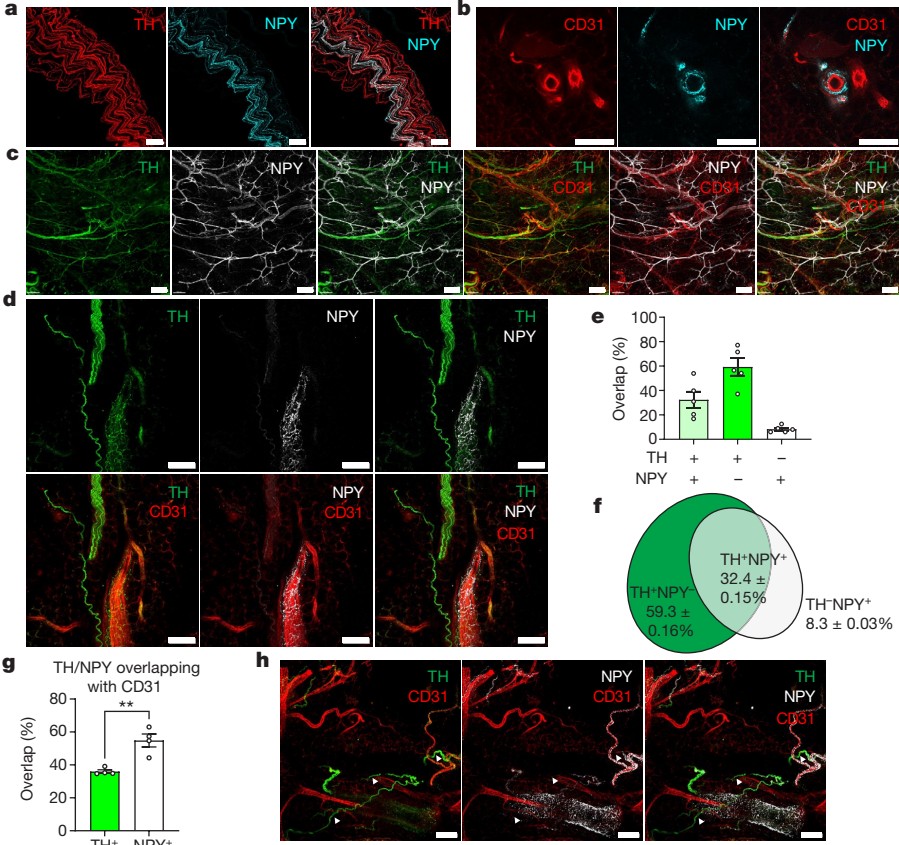

**Fig. 1 | One-third of sympathetic axons in iWAT and iBAT are NPY⁺ and preferentially innervate the vasculature. a**, Confocal images of an axonal bundle dissected from iWAT of a male lean adult mouse, stained with anti-TH (red) and anti-NPY (cyan). **b**, Confocal images of a capillary within cleared iWAT from a lean adult mouse, stained with anti-CD31 (red) and anti-NPY (cyan). **c**, Confocal images of cleared iWAT from lean adult male WT mice, stained with anti-TH (green), anti-NPY (white) and anti-CD31 (red). **d**, Confocal images of cleared iWAT in **c** from lean adult male mice, stained with anti-TH (green), anti-NPY (white) and anti-CD31 (red). **e**,**f**, Quantification of images in **d** for TH/NPY overlap: TH⁺NPY⁻ (dark green), TH⁺NPY⁺ (light green) and TH⁻NPY⁺ (white); percentages shown on a histogram (**e**) and Venn diagram (**f**); $n$ = 5 views from three mice. **g**, Quantification of images in **d** showing the proportions of TH⁺ and NPY⁺ axons overlapping with CD31⁺ endothelial cells ($n$ = 4 views from three mice; $P$ = 0.0035). **h**, Confocal images of cleared iBAT from lean adult male mice stained with anti-TH (green), anti-NPY (white) and anti-CD31 (red). Arrowheads indicate TH⁺NPY⁻ axons. **a–d**,**h**, Representative images are shown from three experiments. All values shown are mean ± s.e.m. Statistical comparisons were made using two-tailed Student's $t$-tests, **$P$ < 0.01. Scale bars, 50 µm (**a**,**c**), 100 µm (**b**,**d**,**h**).

branches (Extended Data Fig. 1i,j) whereas NPY⁺ axons do not innervate eighth-order branches (Extended Data Fig. 1h–j). To identify whether arteries or veins are innervated by NPY⁺ axons in adipose tissue, we costained iWAT for NPY and EPHB4, a vein marker[19], or SOX17, an artery marker[20]. The images show that NPY⁺ axons do not innervate EPHB4⁺ vessels whereas they innervate SOX17⁺ vessels (Extended Data Fig. 1k,l). Based on these observations, we conclude that NPY⁺ sympathetic axons in adipose tissue preferentially innervate arterioles but not veins or venules.

In addition, our NPY stains are exclusively axonal. Previous studies suggested that adipose tissue macrophages (ATMs) can express *Npy*[21], but this is negated by two scRNA-seq datasets of the stromal-vascular fraction (SVF) of mouse iWAT and iBAT[5,22] (Supplementary Fig. 1a–d). Consistently, using quantitative PCR (qPCR), we did not detect *Npy* expression in sorted ATMs from *Cx3cr1*^GFP/+ reporter mice despite the positivity of the *Adgre1* (F4/80) control (Supplementary Fig. 1e,f).

### NPY1R⁺ progenitors of thermogenic fat

We then investigated the type of perivascular cells that are the post-synaptic targets of NPY⁺ axons in adipose tissues. By reanalysis of the scRNA-seq dataset of iWAT[22] and iBAT[5], we found that *Npy1r* is highly expressed in mural cells that are marked by *Des*, *Myh11*, *Acta2* (αSMA),

*Tagln*, *Cspg4* (NG2) and *Pdgfrb*[4–6], and that NPY1R is a bona fide marker of mural cells within iWAT and iBAT (Fig. 2a–d and Supplementary Fig. 1a,b). To validate the scRNA-seq datasets by qPCR we sorted mural cells (live CD31⁻CD45⁻PDGFRα⁻NG2⁺ cells) and immune cells (live CD31⁻CD45⁺ cells) from the SVF of iWAT and iBAT (Supplementary Fig. 1g) and performed qPCR. We independently confirmed that *Npy1r* and *Des* are highly expressed in mural cells but not in immune cells (Fig. 2e).

To confirm the existence of NPY1R in mural cells at the protein level, we dissected capillaries from adipose tissue and stained them with anti-CD31 and anti-NPY1R. We detected an NPY1R⁺ cell wrapping around the capillary (Fig. 2f), the morphology of which matches that of mural cells of fine capillaries, conventionally named pericytes[23]. To confirm the colocalization of NPY1R with multiple mural cell markers observed in scRNA-seq datasets (Extended Data Fig. 2a), we immuno-labelled vessel-containing samples dissected from adipose tissues of *Npy1r*^Cre;*Rosa26*^tdTomato mice for multiple mural cell markers including DES, αSMA and NG2. The images indicate that DES⁺, αSMA⁺ and NG2⁺ mural cells are all labelled with NPY1R-tdTomato (Fig. 2g and Extended Data Fig. 2b–e). Based on the observations above, we demonstrate that mural cells are the target of NPY⁺ axons and that they are NPY1R⁺.

It is worth noting that earlier reports of *Npy2r* and *Npy5r* expression by immune cells, preadipocytes and adipocytes[24–27] are negated by

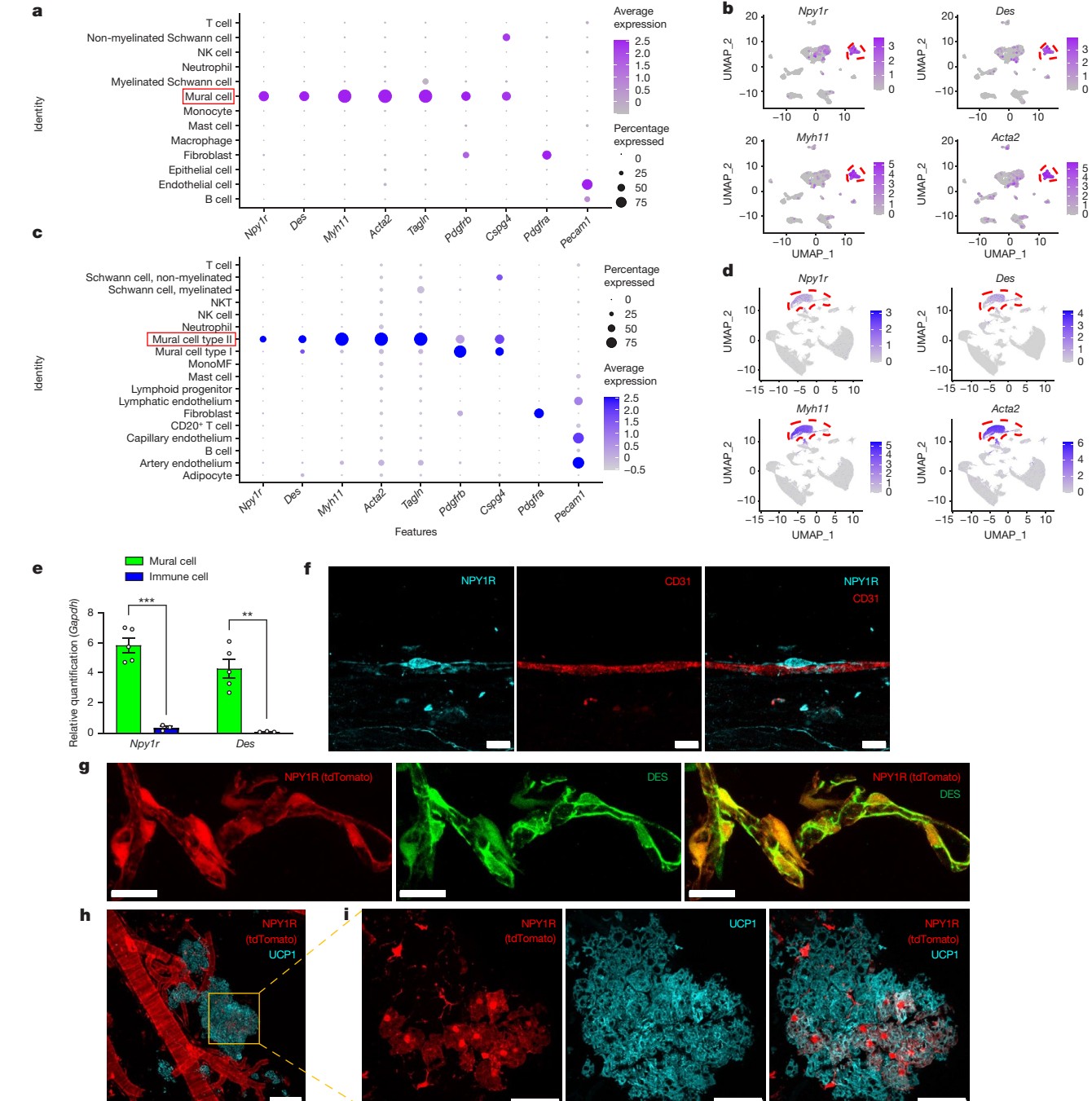

**Fig. 2 | *Npy1r* is mainly expressed in mural cells that are a source of thermogenic adipocytes. a,c**, Dot plots showing the expression of *Npy1r*, *Des*, *Myh11*, *Acta2* (αSMA), *Tagln*, *Cspg4* (NG2), *Pdgfrb*, *Pdgfra* and *Pecam1* (CD31) in the SVF of iWAT (**a**)[22] and iBAT (**c**)[5]. **b,d**, Embedding plots showing the expression of *Npy1r*, *Des*, *Myh11* and *Acta2* (αSMA) in the SVF of iWAT (**b**)[22] and iBAT (**d**)[5]. Mural cell clusters are highlighted by red dashed lines. **e**, Expression of *Npy1r* (*P* = 0.0002) and *Des* (*P* = 0.0024) in sorted CD31⁻CD45⁻PDGFRα⁻NG2⁺ mural cells and CD45⁺ immune cells (*n* = 5 and *n* = 3 biologically independent samples, respectively). *Gapdh* was used as the reference gene. **f**, Confocal images of capillary dissected from iWAT of WT mice and stained with

anti-NPY1R (cyan) and anti-CD31 (red). **g**, Confocal images of vessels dissected from *Npy1r^Cre^;Rosa26^tdTomato^* mice stained with anti-DES (green). **h**, Confocal images of vessels dissected from iBAT of ND-treated, room temperature-housed, 12-week-old *Npy1r^Cre^;Rosa26^tdTomato^* mice stained with anti-UCP1 (cyan). **i**, Zoomed-in images of **h**. **f–i**, Representative images from two experiments. All values mean ± s.e.m. Statistical comparisons were made using two-tailed Student's *t*-tests, **\*\*P* < 0.01, \*\*\*P* < 0.001. NK, natural killer cells; NKT, nature killer T cells; MonoMF, monocytes and macrophages. UMAP, uniform manifold approximation and projection. Scale bars, 10 μm (**f**), 20 μm (**g**), 100 μm (**h**), 50 μm (**i**).

our analysis of the scRNA-seq datasets of both iBAT and iWAT, where *Npy5r* is undetectable and *Npy2r* expression is detectable only in iWAT but negligible (Supplementary Fig. 2a,b). Also, some previous reports of *Npy1r* expression by adipocytes or macrophages[28] are negated by a single-nucleus atlas of mouse WAT and a scRNA-seq dataset of iBAT[5,29],

as well as our qPCR result for sorted ATMs (Supplementary Fig. 1f). Therefore, other than mural cells, no immune cells or adipocytes are likely to be the postsynaptic targets of NPY⁺ axons in adipose tissue.

Mural cells can give rise to thermogenic adipocytes[4–6], and previously identified progenitors of beige cells (Lin⁻CD81⁺) express *Npy1r* and

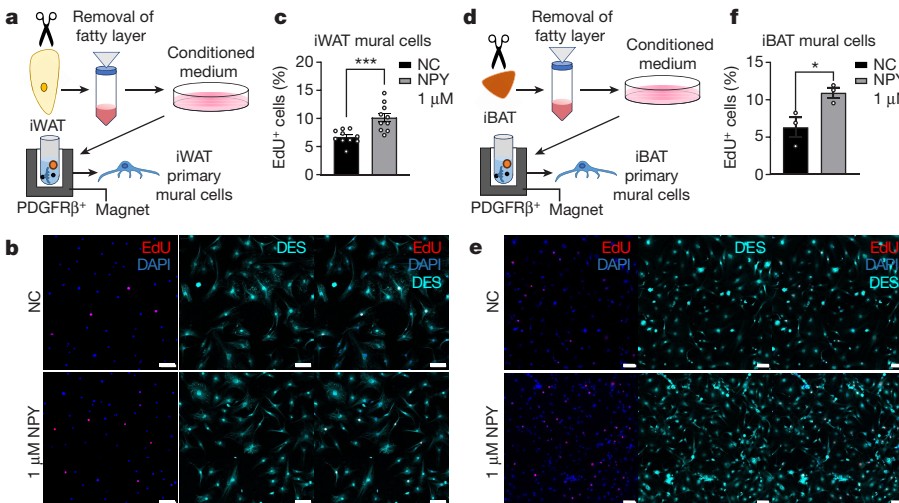

**Fig. 3 | NPY promotes the proliferation of both iWAT and iBAT mural cells.**
**a**, Schematic illustrating isolation of mural cells from iWAT. **b**, Confocal images showing iWAT mural cells treated with or without 1 μM NPY and stained with DES (cyan) or DAPI; 5-ethynyl-2′-deoxyuridine (EdU) (red) indicates proliferating cells. **c**, Percentage of EdU[+] mural cells quantified based on images in **b** (n = 10 biologically independent samples; P = 0.0008). **d**, Schematic showing isolation of mural cells from iBAT. **e**, Confocal images showing iBAT mural cells treated

with or without 1 μM NPY and stained with DES (cyan) and DAPI. EdU (red) indicates proliferating cells. **f**, Percentage of EdU[+] mural cells quantified based on images in **d** (n = 3 biologically independent samples; P = 0.0370). **b**,**e**, Representative images are shown from three and two experiments, respectively. All values mean ± s.e.m. Statistical comparisons were made using two-tailed Student's t-tests, *P < 0.05, ***P < 0.001. NC, negative control (without NPY). Scale bars, 80 μm.

mural cell markers[6] (Extended Data Fig. 3a–c), which is further proven via our analysis of iWAT scRNA-seq data (Extended Data Fig. 3d). We confirmed that mural cells from both iWAT and iBAT are progenitors of thermogenic adipocytes by their isolation and differentiation in vitro[30] (Extended Data Fig. 3e,f,j,i), and found they are more potent in differentiating into thermogenic adipocytes than SVF (Extended Data Fig. 3g–j). To demonstrate that NPY1R[+] mural cells are progenitors of thermogenic adipocytes under physiological conditions, we immunolabelled iBAT of *Npy1r Cre;Rosa26 tdTomato* room temperature-housed mice for uncoupling protein 1 (UCP1). Using this lineage-tracing approach, we observed that both mural cells and a subgroup of UCP1[+] adipocytes were colabelled with NPY1R-tdTomato (Fig. 2h,i). Because adipocytes do not express *Npy1r* (Fig. 2c), we reasoned that these NPY1R-tdTomato[+]UCP1[+] thermogenic adipocytes are lineage traced to NPY1R[+] mural cells. It is worth noting that NPY1R-tdTomato labels only multilocular UCP1[+] adipocytes in iBAT and iWAT, but not UCP1[−] unilocular white adipocytes (Supplementary Fig. 3a–c), which indicates that NPY1R is specifically expressed in the progenitor of thermogenic adipocytes. Based on both the observations above and published literature[4–6], we conclude that NPY1R[+] mural cells are progenitors of thermogenic adipocytes.

## NPY sustains mural cells

NPY1R is a type of G-protein-coupled receptor (GPCR) that can activate mitogen-activated protein kinase (MAPK) pathways following binding to NPY, with the MAPK downstream of NPY1R being extracellular signal-regulated kinase (ERK)[31–33]. Because the MAPK pathway is a canonical regulator of cell proliferation[34], we questioned whether NPY could promote the proliferation of mural cells. By isolation of mural cells and the addition of NPY, we observed that NPY promoted the proliferation of both iWAT and iBAT mural cells (Fig. 3a–f). Furthermore, ERK inhibitor PD98059 blocked the mitogenic effect of NPY (Extended Data Fig. 4a,b), indicating that NPY promotes mural cell proliferation via ERK. Consistent with this, we found that NPY increased the proportion of αSMA[+] mural cells in cultured SVF and CD81[+] cells (Extended Data Fig. 4c–g).

Because mural cells are proven to be the progenitors of thermogenic adipocytes (Extended Data Fig. 3g–j), we reasoned that NPY could

facilitate the neogenesis of thermogenic adipocytes (Extended Data Fig. 4g). To prove this directly, we differentiated iWAT SVF and progenitors of thermogenic adipocytes isolated from iWAT and iBAT, finding that NPY facilitated their differentiation into thermogenic adipocytes by upregulation of thermogenic genes without affecting adipogenic genes (Extended Data Fig. 5a–d,f,g). In addition, NPY increased the maximal respiratory capacity of adipocytes differentiated from iWAT beige adipocyte progenitors (Extended Data Fig. 5e). We further demonstrated that NPY does not affect adipogenesis, using NPY1R[+] 3T3-L1 preadipocytes (Supplementary Fig. 4a–d), confirming that NPY specifically promotes the differentiation of thermogenic adipocytes.

Previous reports have shown that mural cells can be regulated by PDGF-B[35]. PDGF-B is required for the recruitment of mural cells, but elevated levels can decrease mural cells both in vitro and in vivo by upregulation of fibroblast markers and promotion of fibroblast proliferation[36,37]. We questioned whether NPY could sustain mural cells when challenged with PDGF-B, which is physiologically relevant because diet-induced obesity (DIO) increases PDGF-B in adipose tissue[37]. By treating isolated mural cells with PDGF-BB (dimerized PDGF-B), we found that PDGF-BB decreased the proportion of mural cells and increased the number of spindle-shaped PDGFRα[+] fibroblasts, an effect that is counteracted by NPY (Extended Data Fig. 6a,b). Using qPCR, we demonstrated that NPY restored the expression of *Npy1r* and other mural cell markers including *Pdgfrb* and *Rgs5* that are downregulated by PDGF-BB (Extended Data Fig. 6c–e). Because mural cells are identified as the progenitor of thermogenic adipocytes, we examined whether NPY can preserve their beiging ability against PDGF-BB. We observed that PDGF-BB treatment downregulated thermogenic and adipogenic genes and that NPY restored their expression (Extended Data Fig. 6f–h). Based on the observations above, we demonstrate in vitro that NPY can sustain mural cells as the progenitors of thermogenic adipocytes.

## DIO depletes NPY[+] axons

To test whether NPY[+] sympathetic axons are affected by the obesity-induced sympathetic neuropathy in adipose tissue[38], we immunolabelled cleared iWAT of normal-diet (ND)- and high-fat-diet

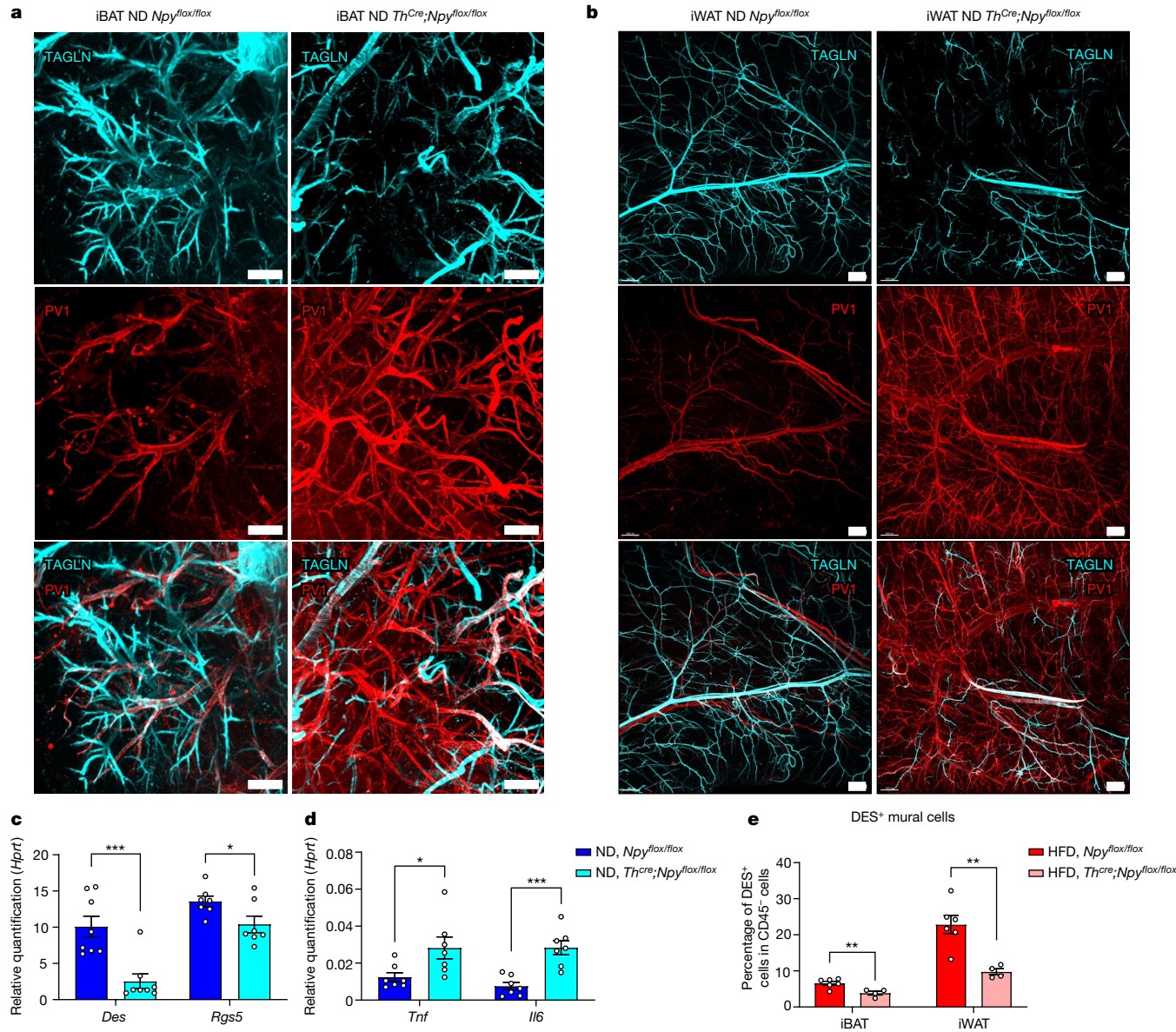

**Fig. 4 | Loss of NPY from sympathetic neurons depletes mural cells and increases vascular leakiness in both iBAT and iWAT. a,b**, Light-sheet images of cleared iBAT from 15-week-old ND-treated mice (**a**) and of iWAT from 30-week-old ND-treated mice (**b**) stained with anti-PV1 (red) and anti-TAGLN (cyan). **c,d**, Expression levels of mural cell markers *Des* (*n* = 8 mice; *P* = 0.0007) and *Rgs5* (*n* = 7 mice; *P* = 0.0374) (**c**) and proinflammatory genes *Tnf* (*n* = 7 mice, *P* = 0.0284) and *Il6* (*n* = 7 mice, *P* = 0.0004) (**d**) in BAT of ND-treated, 12-week-old, male *Th^{Cre};Npy^{flox/flox}* and *Npy^{flox/flox}* mice. **e**, Percentage of DES⁺ mural cells in iBAT (*Npy^{flox/flox}* versus *Th^{Cre};Npy^{flox/flox}*, *P* = 0.0084) and iWAT (*Npy^{flox/flox}* versus *Th^{Cre};Npy^{flox/flox}*, *P* = 0.0038) of 17-week-old, HFD-treated mice, measured by flow cytometry (*n* = 4 mice). **a,b**, Representative images from three experiments. All values mean ± s.e.m. Statistical comparisons were made using two-tailed Student's *t*-tests, *\*P* < 0.05, *\*\*P* < 0.01, *\*\*\*P* < 0.001, *\*\*\*\*P* < 0.0001. Scale bars, 200 μm (**a**), 500 μm (**b**). Hypoxanthine–guanine phosphoribosyltransferase (*Hprt*) was used as the reference gene for qPCR.

(HFD)-treated mice for NPY and CD31. Our light-sheet images show a decreased density of NPY⁺ axons and a decreased percentage of vessels innervated by NPY⁺ axons in the iWAT of HFD-treated mice compared with ND-treated lean mice (Extended Data Fig. 7a,b). Consistently, NPY concentration also diminished in the iWAT of HFD-induced obese mice and leptin-deficient obese mice (Extended Data Fig. 7c). This decrease is not due to any changes in *Npy* expression in the ganglionic neuronal soma (Supplementary Fig. 5a,b) or systemic NPY level (Extended Data Fig. 7d). These results indicate that obesity-induced sympathetic neuropathy reduces NPY⁺ innervation in adipose tissue, which in turn decreases the local concentration of NPY in iWAT.

We then investigated whether mural cells, the postsynaptic targets of NPY⁺ innervation in iWAT, are also affected by obesity.

By immunolabelling cleared iWATs for DES and CD31, we found that there are fewer mural cells covering the vasculature of the iWAT of DIO mice compared with age-matched lean mice (Extended Data Fig. 7e,f). Given that mural cells are essential for vascular integrity[39], we observed increased vascular leakiness in the iWAT of DIO mice, as shown by PV1 staining[40] (Extended Data Fig. 7g,h). To determine whether this phenomenon is related to NPY levels, we immunolabelled sympathetic ganglia for DES to ascertain whether mural cell coverage changes when NPY levels are unaffected. We observed no significant difference in mural cell coverage of ganglionic vasculature between DIO and lean mice (Supplementary Fig. 5b–e), confirming that mural cell coverage positively correlates with local NPY level.

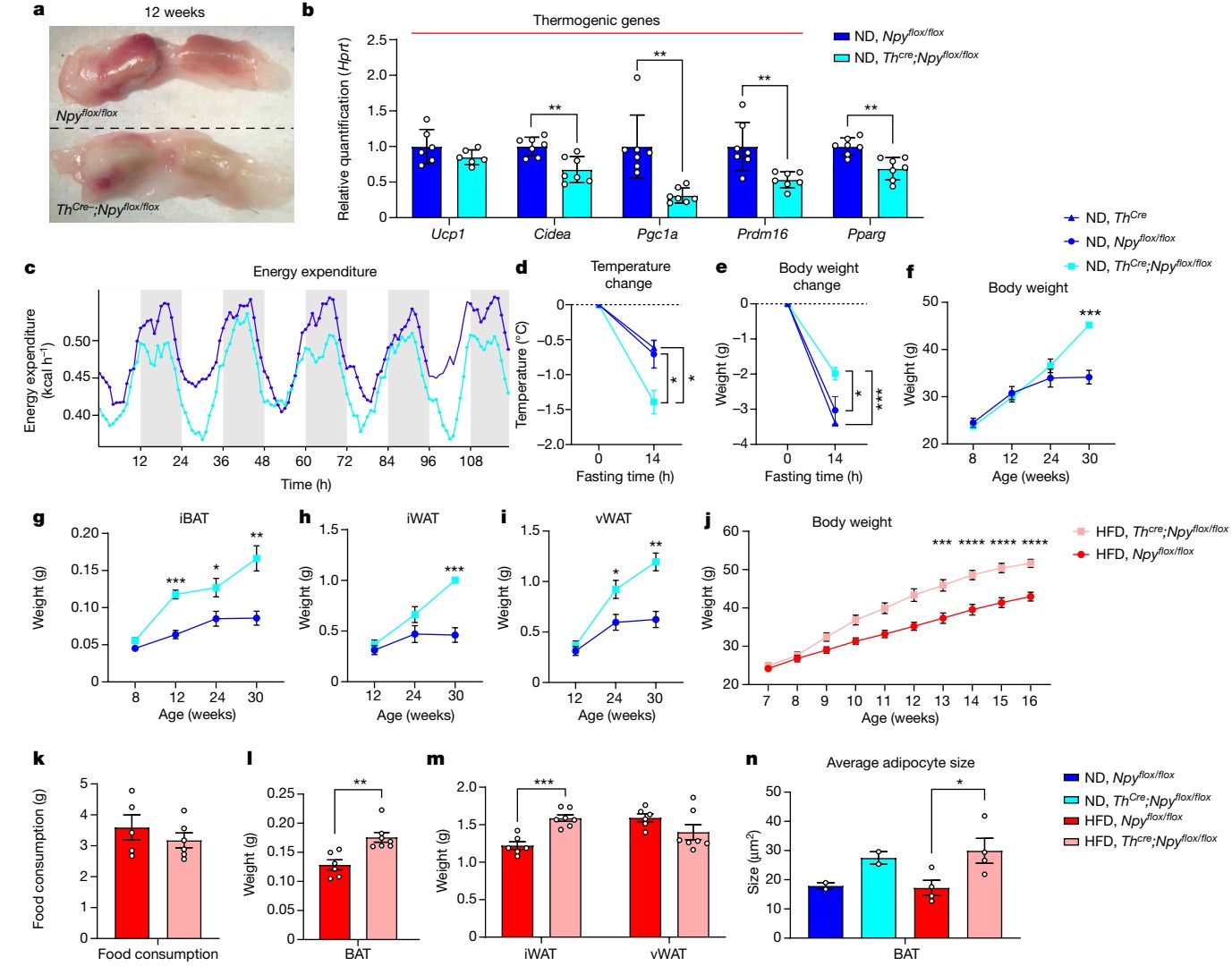

**Fig. 5 | Loss of NPY from sympathetic neurons whitens BAT, decreases energy expenditure and thermogenesis and renders mice more susceptible to DIO without increasing food intake. a**, iBAT dissected from ND-treated, 12-week-old, male *Npy*[flox/flox] and *Th*[Cre]*;Npy*[flox/flox] mice. Dashed lines delineate iBAT lobes. **b**, Expression of thermogenic genes *Ucp1*, *Cidea* (*P* = 0.0024), *Pgc1a* (*P* = 0.0018) and *Prdm16* (*P* = 0.0048), and of adipogenic gene *Pparg* (*P* = 0.0012), in iBAT from ND-treated, 12-week-old, male *Th*[Cre]*;Npy*[flox/flox] and *Npy*[flox/flox] mice (*n* = 6 mice for *Ucp1*, *n* = 7 mice for other genes). **c**, Daily energy expenditure of 7-week-old male *Th*[Cre]*;Npy*[flox/flox] and *Npy*[flox/flox] mice, measured using an indirect calorimetry system (*n* = 9). **d,e**, Changes in iBAT temperature (*Th*[Cre] versus *Th*[Cre]*;Npy*[flox/flox] mice, *P* = 0.0198; *Npy*[flox/flox] versus *Th*[Cre]*;Npy*[flox/flox] mice, *P* = 0.0440) (**d**) and body weight (*Th*[Cre] versus *Th*[Cre]*;Npy*[flox/flox] mice, *P* = 0.0008; *Npy*[flox/flox] versus *Th*[Cre]*;Npy*[flox/flox] mice, *P* = 0.0225) (**e**) of 12-week-old mice following a 14-h fast (*n* = 6 for *Th*[Cre]*;Npy*[flox/flox] mice and *n* = 3 for *Th*[Cre] and *Npy*[flox/flox] mice). **f–i**, Body weight (week 30,

*P* = 0.0003) (**f**) and weights of iBAT (week 12, *P* = 0.0006; week 24, *P* = 0.0387; week 30, *P* = 0.0058) (**g**), iWAT (week 30, *P* = 0.0004) (**h**) and visceral WAT (vWAT) (week 24, *P* = 0.0338; week 30, *P* = 0.0030) (**i**) from ND-treated, male *Th*[Cre]*;Npy*[flox/flox] and *Npy*[flox/flox] mice (*n* = 4 mice). **j,k**, Weekly body weight (*n* = 8; week 13, *P* = 0.0007; weeks 14–16, *P* < 0.0001) (**j**) and averaged daily food consumption (*n* = 4 and *n* = 6 cages) (**k**) from male, HFD-treated, *Th*[Cre]*;Npy*[flox/flox] and *Npy*[flox/flox] mice. **l,m**, Weights of iBAT (*n* = 5 and *n* = 6 mice, *P* = 0.0023) (**l**), and iWATs (*n* = 6 and *n* = 7 mice, *P* = 0.0002) and vWAT (*n* = 6 and *n* = 7 mice) (**m**), from HFD-treated, 17-week-old, male *Th*[Cre]*;Npy*[flox/flox] and *Npy*[flox/flox] mice. **k–m**, Colour coding as in **j**. **n**, Average adipocyte size in iBAT from ND- and HFD-treated male *Th*[Cre]*;Npy*[flox/flox] and *Npy*[flox/flox] mice (*n* = 2 for ND-treated mice, *n* = 4 for HFD-treated mice, *P* = 0.0436). **a**, Representative images from three experiments. All values mean ± s.e.m. Statistical comparisons were made using two-tailed Student's *t*-tests, *\*P* < 0.05, *\*\*P* < 0.01, *\*\*\*P* < 0.001, *\*\*\*\*P* < 0.0001. *Hprt* was used as the reference gene for qPCR.

## Sympathetic NPY regulates leakiness

To test the hypothesis that sympathetic neuron-derived NPY is required to sustain mural cells in vivo, we generated a mouse model in which NPY was abrogated from sympathetic neurons: *Th*[Cre]*;Npy*[flox/flox] mice. We first confirmed that, in *Th*[Cre]*;Npy*[flox/flox] mice, NPY was abrogated from sympathetic ganglia (Extended Data Fig. 8a,b) and sympathetic axons innervating iWAT and iBAT (Extended Data Fig. 8e,g), whereas general TH[+] sympathetic innervation was not affected (Extended Data Fig. 8f,h). Using this mouse model, we demonstrated that the loss of NPY in sympathetic neurons downregulated mural cell markers in iBAT of

ND-treated mice (Fig. 4c) and depleted mural cells in iBAT and iWAT of both ND-treated (Fig. 4a,b and Extended Data Fig. 8c) and HFD-treated mice (Fig. 4e).

Because mural cells are required for vascular integrity[39], we observed increased vascular leakiness in both iBAT and iWAT of *Th*[Cre]*;Npy*[flox/flox] mice, as demonstrated by PV1 staining (Fig. 4a,b). We further evaluated vascular leakiness in epididymal WAT using an intravital method by intravenous injection of Dextran-70kDa. Increased vascular leakiness in *Th*[Cre]*;Npy*[flox/flox] mice was shown by leakage of Dextran-70kDa into the parenchyma (Extended Data Fig. 9a–d) whereas blood flow remained unchanged (Extended Data Fig. 9e). Consistent with increased vascular

leakiness, we observed upregulated proinflammatory genes in iBAT (Fig. 4d) and increased immune infiltration in both iBAT and iWAT of ND-treated $Th^{Cre};Npy^{flox/flox}$ mice (Extended Data Fig. 8d).

These observations confirmed that NPY is required for sustaining mural cells in adipose tissue and preventing immune infiltration.

## Sympathetic NPY protects from obesity

To test the hypothesis that NPY locally sourced from sympathetic axons in adipose tissues preserves the mural cell pool and is required for energy homeostasis, we characterized the metabolism of $Th^{Cre};Npy^{flox/flox}$ mice. We discovered that ND-treated, lean $Th^{Cre};Npy^{flox/flox}$ mice had whitened and enlarged iBAT (Fig. 5a,g) and that their iBAT was dysfunctional in thermogenesis, as shown by lower levels of thermogenic genes (Fig. 5b). Consistent with this, $Th^{Cre};Npy^{flox/flox}$ mice have lower daily energy expenditure (Fig. 5c and Extended Data Fig. 9f) and higher respiratory exchange (Extended Data Fig. 9i,j), indicating reduced usage of fat as a metabolic substrate. When food restricted, $Th^{Cre};Npy^{flox/flox}$ mice showed lower iBAT temperature and lost less weight during a 14-h fast, indicative of energy conservation (Fig. 5d,e). Following 8 h cold exposure, $Th^{Cre};Npy^{flox/flox}$ mice showed lower rectal temperature (Extended Data Fig. 9k) and downregulation of thermogenic genes in iBAT and $Prdm16$ in iWAT (Extended Data Fig. 9l,m) compared with their wild-type (WT) littermates. These alterations are not likely to be related to muscular activity because $Th^{Cre};Npy^{flox/flox}$ mice have the same lean mass and locomotor activity as WT mice (Extended Data Fig. 9g,h).

As a cumulative result of deficient thermogenic ability and reduced energy expenditure, ND-treated $Th^{Cre};Npy^{flox/flox}$ mice developed adult-onset obesity at 30 weeks, with heavier adipose depots compared with WT mice (Fig. 5f–i). When on a HFD, $Th^{Cre};Npy^{flox/flox}$ mice became obese earlier and faster than WT mice without increasing food intake (Fig. 5j,k). In addition, HFD-treated $Th^{Cre};Npy^{flox/flox}$ mice had significantly heavier adipose depots (Fig. 5l,m) and their iBAT contained larger lipid droplets on average (Fig. 5n and Extended Data Fig. 9n,o). Lastly, metabolic phenotypes were consistent across genders because HFD-treated female $Th^{Cre};Npy^{flox/flox}$ mice also had heavier body weight, WAT and iBAT compared with $Npy^{flox/flox}$ control mice (Supplementary Fig. 6a–c).

We can ascertain that the metabolic phenotypes observed in $Th^{Cre};Npy^{flox/flox}$ mice were not caused by NPY$^+$ neurons in the brain or systemic NPY, because (1) no catecholaminergic neurons coexpress NPY in the hypothalamus[41] or ventral tegmental area (Extended Data Fig. 10a–d) and $Npy$ expression was not changed in the hypothalamus of $Th^{Cre};Npy^{flox/flox}$ mice (Extended Data Fig. 10e). (2) The metabolic phenotypes were not caused by systemic NPY because NPY concentration in the blood plasma of $Th^{Cre};Npy^{flox/flox}$ mice was unchanged (Extended Data Fig. 10f). (3) Even though there are NPY$^+$ catecholaminergic neurons in the hindbrain[42], knocking out NPY from the hindbrain has no effect on either thermogenesis or iBAT (Extended Data Fig. 10g–m). Taken together, our findings demonstrate that sympathetic neuron-derived NPY in adipose tissue is required for sustaining energy expenditure and leanness.

## Discussion

Mural cells are contractile cells that wrap around the vasculature to regulate the diameter of vessels[43] and have been described as a source of thermogenic adipocytes in both WAT and iBAT[4–6], which are subject to the influence of noradrenaline released from adjacent sympathetic nerve terminals. These autonomic nerves corelease NPY[2,3,9], but their function in the development and activity of thermogenic adipocytes is hitherto unknown. In humans, mutations in NPY have been linked to high BMI but not to dietary patterns[1] (https://hugeamp.org/gene.html?gene=NPY; Extended Data Fig. 1m). In mice, the loss of NPY receptors leads to late-onset obesity without increase in food intake[14–16]. Taken together, the experimental evidence points to a role of NPY in body weight homeostasis that is independent of appetite regulation. Three studies have implicated NPY in directing lipogenic actions in white adipocytes, which favoured the growth of white adipose mass[24,25,27]. However, some of these studies are not replicated here[24,25] and others are not supported by modern evidence provided by our imaging data and several single-cell sequencing datasets[27]. Specifically, we show in our supplementary data that NPY, directly, promotes neither adipogenesis[24] nor the proliferation of 3T3-L1 preadipocytes[25]. Thus the evidence disfavours a pro-obesity role for peripheral NPY, contradicting the current perception.

This study provides anatomical maps of NPY$^+$ innervation in both WAT and BAT that demonstrate that only one-third of sympathetic innervation is neuropeptidergic and that it preferentially targets arterioles where mural cells are present. Consistent with recent studies showing that sympathetic axons within adipose tissues retract and degenerate with the onset of obesity[8,38], our data show that the axonal source of NPY, locally, within these tissues is also obliterated and this, again, refutes a direct adipogenic role of peripheral NPY. Moreover, several modern datasets omit NPY receptors in adipocytes or immune cells whereas $Npy1r$ expression is robustly detected in mural cells, identifying these cells as the postsynaptic targets of NPY$^+$ innervation within adipose tissues. We show that NPY sustains mural cells that are a source of thermogenic adipocytes by promotion of their proliferation in vitro. In vivo, NPY sourced from sympathetic neurons is necessary for support of mural cells and the normal development of thermogenic adipocytes. Conditional knockout of NPY (NPY-cKO) in sympathetic neurons depletes mural cells and whitens brown fat before the onset of obesity, rendering this tissue hypertrophic and with reduced thermogenic ability. These NPY-cKO mice have reduced thermogenic ability and lower energy expenditure, show cold intolerance, develop late-onset obesity on a regular chow diet and rapidly develop obesity when fed a high-fat diet, without eating more. These metabolic phenotypes match those of whole-body-NPY-receptor-knockout mice[14–16], which also develop late-onset obesity on a regular chow diet even though they eat slightly less.

Notwithstanding this, disrupted mural cell proliferation and adipocyte differentiation are reasons for adipose dysfunction in NPY-cKO mice and are among many possibilities, including a reciprocal causality between vascular leakiness and adipose tissue inflammation. Taken together, we clarify that central and peripheral sources of NPY have antipodal roles in body weight homeostasis. The biological mechanism described in this study is a potential explanation for why mutations of the orexigenic NPY paradoxically associate with high BMI in humans (https://hugeamp.org/gene.html?gene=NPY)[1]. Lastly, it highlights that energy dissipation may be more important than appetite in some individuals via the mechanism identified herein.

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

# Methods

## Animals

$Th^{Cre}$ mice (B6.Cg-$7630403G23Rik^{Tg(Th\text{-}cre)1Tmd}$/J, stock no. 008601) and $Cx3cr1^{GFP/+}$ mice ($Cx3cr1^{tm1Litt}$/LittJ, stock no. 008451) were purchased from the Jackson Lab. $Npy^{flox/flox}$ mice were a donation from the I. Kalajzic Laboratory at the Department of Reconstructive Science, University of Connecticut[44] under the material transfer agreement (MTA). Tissues of NPY-GFP mice (B6.FVB-Tg(Npy-hrGFP)1Lowl/J) were sourced from the Tamas Horvath Laboratory at Brandy Memorial Laboratory, Yale University. Tissues from $Npy1r^{Cre}$;$Rosa26^{tdTomato}$ mice were provided by M. Roberts at the Department of Otolaryngology–Head and Neck Surgery, University of Michigan. Sympathetic neuron-specific NPY-cKO mice were generated by crossing $Th^{Cre}$ mice with $Npy^{flox/flox}$ mice. Diet-induced obesity was achieved by feeding mice an HFD (Diet Research, product no. D12492) when they were 7 weeks old, and this feeding regime lasted for 10 weeks. The body weight of each mouse and food consumption in each cage were recorded weekly. All mice were group housed in standard housing under controlled room temperature (21–23 °C) and 50% humidity under a 12/12 h light/dark cycle and given access to diet and water ad libitum. All experimental procedures were performed on living animals in accordance with the UK ANIMALS ACTS 1986 under the project licence (PPL no. P80EDA9F7) and personal licences granted by the UK Home Office. Ethical approval was provided by the Ethical Review Panel at the University of Oxford.

## Energy expenditure and measurement of respiratory exchange rate

Animals aged 6 weeks were analysed for oxygen consumption, carbon dioxide production, energy expenditure and respiratory exchange rate using an indirect calorimetry system (Panlab, Harvard Apparatus, LE405 Gas Analyzer and Air Supply & Switching). Animals were maintained in individual cages following a 12/12 h light/dark cycle with water and food supplied ad libitum, and under controlled room temperature (21–23 °C) and 50% humidity. Metabolic data were collected for 5 days following a 2 day acclimatation period. Animals' body weight (g) and food (g) were measured before entering and after exiting the cage. Results were normalized by weight, and graphs and statistical analysis were obtained using the CalR Web-based Analysis Tool for Indirect Calorimetry Experiments (v.1.3)[45].

## Locomotor activity and measurement of body composition

Animals aged 8–9 weeks were analysed for spontaneous locomotor activity (Panlab, Harvard Apparatus, LE001 PH Multitake Cage). Animals were maintained in individual cages following a 12/12 h light/dark cycle with water and food supplied ad libitum, and under controlled room temperature (21–23 °C) and humidity. Data were collected for 72 h following a 1 day acclimatation period. Activity was recorded using COMPULSE v.1.0 software (Panlab). Animals aged 7–9 weeks were analysed for body composition using a Minispec LF50 (Brucker).

## Fasting and thermoimaging

Animals aged 12–14 weeks were used for thermoimaging. The data shown in Fig. 5e were recorded using an Optris thermocamera (Optris PI 160 with a standard 61 lens), and the data in Extended Data Fig. 10l were recorded using a FOTRIC 225 s infrared camera and analysed with FOTRIC software (v.5.0.8.214). Animals were single housed and placed under the thermocameras, with their back shaved to expose the skin above iBAT. Animals were acclimatized for 4 days in individual cages with ad libitum access to food and water at room temperature under a 12/12 h light/dark cycle before a 14-h fast. Mice had ad libitum access to water during fasting.

iBAT temperature was recorded every 1 s over a 6-day period by storing the temperature of the warmest pixel in view using the software provided by the camera manufacturer (rel. 2.0.6, Optris PIX Connect).

iBAT temperature at 0 and 14 h was taken as an average of temperatures during a 1-h period.

## Cold-exposure experiment

Animals aged 20 weeks were single housed in a thermoneutral environment (31 °C, 50% humidity) under a 12/12 light/dark cycle for 1 week, with ad libitum access to food and water. Mice were then cold-challenged at 4 °C and their rectal temperature measured every 1 h using a rectal probe. Animals were killed immediately following 8 h of cold exposure and their iBAT harvested for qPCR with reverse transcription analysis.

## Stereotaxic injection

For viral injection, mice were anaesthetized with avertin (240 mg kg$^{-1}$ intraperitoneally) and fixed on a stereotaxic holder (RWD, Life Science). Either AAV2/9-hSyn-Cre-EGFP-WPRE-pA (Taitool, catalogue no. S0230-9, 2 × 1,012 viral genomes ml$^{-1}$, 200 nl per side) or AAV2/9-hSyn-EGFP-WPRE-pA (catalogue no. S0237-9, Taitool, 2 × 1,012 viral genomes ml$^{-1}$, 200 nl per side) was bilaterally injected into the nucleus tractus solitarius (NTS) (NTS coordinates anteroposterior, mediolateral, dorsoventral −7.5, ±0.35 and −4.75 mm, respectively) of $Npy^{flox/flox}$ mice.

## Immunofluorescent staining

Superior cervical ganglia and sympathetic axon bundles were dissected from adipose tissues of mice perfused with 20 ml of PBS, and the tissues were fixed in 4% paraformaldehyde (PFA, 043368.9 M, ThermoFisher) for at least 4 h. Cells collected following in vitro experiments were washed once with PBS and fixed using 4% PFA for at least 4 h. Axon bundles were first stained by incubation overnight at 4 °C with primary antibodies dissolved in the permeabilizing buffer (3% bovine serum albumin, 2% goat serum, 0.1% Triton X-100 and 0.1% sodium azide in PBS), followed by incubation with secondary antibodies and DAPI dissolved in the permeabilizing buffer for 1 h with gentle agitation. As for immunofluorescence in fixed cells, cells were first incubated with the permeabilizing buffer for 1 h at room temperature and then stained overnight at 4 °C with primary antibodies diluted in the permeabilizing buffer. Samples were then washed and stained with DAPI and secondary antibodies for 1 h at room temperature. Sympathetic fibres or fixed cells were then mounted on slides with Fluoromount-G mounting medium (ThermoFisher, catalogue no. 00-4958-02).

For cryosection, samples were first embedded in OCT (VWR, catalogue no. 361603E) and frozen at −20 °C until cryosectioning. Samples were cut into 10-µm-thick sections and mounted on charged SuperFrost Plus slides (VWR, catalogue no. 631-0108), followed by staining using the same procedure as for fixed cells.

For paraffin embedding, fixed samples were processed in 70, 80 and 90% ethanol and then in 100% ethanol, 100% xylene and 60 °C paraffin wax, three times for each condition. Samples were embedded in paraffin wax, sectioned at 7 µm, mounted on coverslips and dried in an oven at 45–50 °C overnight. Before staining, samples were deparaffinized twice in 100% xylene for 5 min, rehydrated twice in 100% ethanol and then once each in 95 and 80% ethanol and water for 10 min. Antigen retrieval was done by the addition of 0.05% trypsin to each slide and incubation at 37 °C for 20 min. After washing with running water, samples were stained using the same procedure as for fixed cells.

For staining, whole-mount, PFA-fixed ganglia were dehydrated once in each of 30, 50, 70 and 90% ethanol, twice in 100% ethanol−with shaking for 20 min at each concentration−and then rehydrated in 90, 70, 50 and 30% ethanol. Ganglia were then digested with 1.5 U ml$^{-1}$ dispase-1 (Roche, catalogue no. 04942078001), 0.5 mg ml$^{-1}$ collagenase (Sigma, catalogue no. C2674) and 300 µg ml$^{-1}$ hyaluronidase (Sigma, catalogue no. H3884) diluted in PBS, with shaking at 37 °C in a water bath for 30 min. Ethanol-treated ganglia were blocked in blocking solution

(3% bovine serum albumin, 2% goat serum, 0.1% Triton X-100 and 0.1% sodium azide in PBS) for 2 h and then incubated with primary antibodies diluted in the permeabilizing buffer for 3 days at 4 °C. Three days later, ganglia were incubated in secondary antibodies diluted in the permeabilizing buffer for a further 3 days at 4 °C. Finally, ganglia were dehydrated once each in 30, 50, 70 and 90% ethanol and twice in 100% ethanol then cleared using ethyl cinnamate (ECi). Cleared ganglia were mounted in a coverslip slide sandwich filled with ECi.

Images were acquired using a Zeiss LSM880 confocal microscope with Zen-black software (v.2.1). Samples were imaged using either (1) a ×10/0.45 numerical aperture (NA) objective with a voxel size of 1.04 µm ($x$), 1.04 µm ($y$) and 3.64 µm ($z$), (2) a ×20/0.8 NA objective with voxel size of 0.52 µm ($x$), 0.52 µm ($y$) and 1.00 µm ($z$) or (3) a ×63/1.4 NA oil-immersion objective with a voxel size of 0.13 µm ($x$), 0.13 µm ($y$) and 1.00 µm ($z$). Solid-state lasers of 405, 561 and 633 nm and an Argon 488 laser were used for DAPI, AF488, AF546 and AF647 fluorophores, respectively.

## Clearing of adipose tissue

Whole fat pads were stained and cleared using a modified version of iDISCO as previously described[18,46]. In brief, WAT or BAT was dissected from mice perfused with 20 ml of PBS, and the tissues were fixed with 4% PFA overnight. Tissues were pretreated once each with 20, 40, 60 and 80% ethanol and twice in 100% methanol for 30 min, all at room temperature, and then bleached with 5% $H_2O_2$ diluted in 100% methanol for 24 h at 4 °C with shaking. Bleached samples were rehydrated using 80, 60, 40 and 20% methanol for 30 min each and then in PTwH buffer (0.2% Tween-20, 10 µg ml$^{-1}$ heparin and 0.02% sodium azide in PBS) for 30 min. Afterwards, samples were incubated overnight in permeabilizing solution (20% DMSO, 0.2% Triton X-100, 0.3 M glycine in PBS) at 37 °C in a water bath and blocked using blocking buffer (10% DMSO, 2% Triton X-100, 0.02% sodium azide and 5% goat serum in PBS). For immunolabelling, samples were incubated for 5 days, with shaking, in primary antibodies diluted in antibody dilution buffer (5% DMSO, 5% goat serum, 0.2% Tween-20 and 10 µg ml$^{-1}$ heparin in PBS) at 37 °C, washed with PTwH for 1 day and then incubated for a further 3 days with secondary antibodies and DAPI diluted using antibody dilution buffer at 37 °C, with shaking. Immunolabelled tissues were embedded in 1% agarose and dehydrated in 20, 40, 60, 80 and 100% methanol for 1 h each, and then in 100% methanol overnight at room temperature. Samples were incubated in dichloromethane (Sigma, catalogue no. 270997) until they sank and were then incubated in dibenzyl ether (Sigma, catalogue no. 179272) until clear. Samples were stored in dibenzyl ether at room temperature and transferred into ECi before imaging. Images of cleared tissues were acquired using a Miltenyi Biotec Ultramicroscope II light-sheet microscope; step size was 8 µm, thickness of the light sheet was 3.98 µm, horizontal dynamic focusing was set to eight steps and exposure time was 180 ms. Samples were illuminated using a bidirectional light sheet and scanned under a ×2/0.5 NA objective with voxel size 4.03 µm ($x$), 4.03 µm ($y$) and 8 µm ($z$). A 488-nm laser with a 525/50 filter, a 561-nm laser with a 620/60 filter and a 638-nm laser with a 680/30 filter were used for AF488, AF546 and AF647, respectively.

## Dextran-70kDa injection and intravital microscopy

Animals were anaesthetized with ketamine (100 mg kg$^{-1}$) and xylazine (10 mg kg$^{-1}$) and a small incision was made to expose epididymal white adipose tissue for microscopy analysis. Intravital images of space resolution 1,024 × 1,024 pixels from epididymal WAT were obtained simultaneously using an optical microscope with coherent anti-Stokes Raman scattering, two-photon excitation fluorescence and a bright-field microscope, with a confocal LSM 780-NLO Zeiss inverted microscope Axio Observer Z.1 (Carl Zeiss). All procedures were performed at the National Institute of Science and Technology on Photonics Applied to Cell Biology at the State University of Campinas.

Adipocyte images from coherent anti-Stokes Raman scattering were acquired using two lines of lasers in wavelengths $\lambda_{pump}$ = 803 nm and $\lambda_{Stokes}$ = 1,040 nm, and fluorescence images were acquired by two-photon excitation fluorescence using the exogenous fluorescence dye tetramethylrhodamine isothiocyanate dextran (Sigma, catalogue no. T1162) in wavelength of excitation $\lambda_{Stokes}$ = 1,040 nm. Following identification of blood vessels and acquisition of an initial image the video was started and, after ten frames, 50 µl (30 mg kg$^{-1}$) of Dextran-70KDa (Sigma, catalogue no. T1162) was injected into the orbital plexus of the mouse. The video continued recording for approximately 12 min to capture vessel fluorescence and leakage. Fluorescence intensity over time was analysed for the whole duration of the video. A static image was analysed following dextran administration using Fiji[47]. For evaluation of tissue leakage, a region of interest was selected inside the vessel and another in the tissue (outside the vessel). Fluorescence intensity was measured in both areas, and the intensity ratio between the tissue and within the vessel was calculated. Blood flow was calculated using the average of the raw intensity of dextran inside the artery over 100 s immediately following stabilization of the dextran signal after administration.

## Antibodies

The following primary antibodies were used for immunofluorescent staining: rat anti-CD31 (BioLegend, catalogue no. 102501, MEC13.3, 1:100 dilution), rat anti-PLVAP (BioLegend, catalogue no. 120503, MECA32, 1:100 dilution), rabbit anti-DES (abcam, catalogue no. ab15200, 1:500 dilution), rabbit anti-NPY (Cell Signaling, catalogue no. D7Y5A, 1:500 dilution), rabbit anti-NPY (abcam, catalogue no. ab30914, 1:500 dilution), chicken anti-TH (Aves Labs, catalogue no. TYH73787982, 1:500 dilution), rabbit anti-TH (Sigma, catalogue no. Ab152, 1:500 dilution), mouse anti-NPY1R (Santa Cruz, catalogue no. sc-393192, 1:200 dilution), rat anti-PDGFRα (BioLegend, catalogue no. 135902, APA5, 1:200 dilution), rabbit anti-TAGLN (abcam, catalogue no. ab14106, 1:250 dilution), Cy3 anti-αSMA (Sigma, catalogue no. C6198, 1A4, 1:250 dilution), goat anti-SOX17 (R&D, catalogue no. AF1924, 1:250 dilution), goat anti-EPHB4 (R&D, catalogue no. AF3034, 1:250 dilution) and rabbit anti-UCP1 (abcam, catalogue no. ab10983, 1:500 dilution).

The following antibodies were used for fluorescent activated cell sorting (FACS) and flow cytometry: AF700 anti-CD45 (BioLegend, catalogue no. 103128), BUV395 anti-CD45 (BD Horizon, catalogue no. 564279), Pacific Blue anti-CD31 (BioLegend, catalogue no. 102421), APC anti-PDGFRa (BioLegend, catalogue no. 135907), AF488 anti-NG2 (Sigma, catalogue no. MAB5384A4) and AF488 anti-DES (abcam, catalogue no. ab185033, Y66). All antibodies for FACS and flow cytometry were diluted at 1:500. The LIVE/DEAD Fixable Near-IR Dead Cell Stain Kit (1:1,000, ThermoFisher, catalogue no. L10119) was used for live/dead staining.

## qPCR

RNA extraction was performed using Trizol reagent (ThermoFisher, catalogue no. 15596026), complementary DNA was synthesized using SuperScript II Reverse Transcriptase (Invitrogen, catalogue no. 18064022) and qPCR was performed using Power SYBR Green PCR Master Mix (LifeTech, catalogue no. 4368706). Data were recorded using a StepOne qPCR system and analysed with StepOne Software v.2.3. Primers used in this research are listed in Supplementary Table 1.

The ΔCt method was used to quantify gene expression using the following formula: relative expression = 2^ ($-(Ct_{target gene} - Ct_{reference gene})$). The ΔΔCt method was used to compare the thermogenic and adipogenic gene expression shown in Fig. 5b and Extended Data Figs. 5b,d,g, 9l,m and 10m.

The qPCR data shown in Extended Data Fig. 10m were collected using the CFX96 Real-Time PCR Detection System (Bio-Rad) with TB Green Premix Ex Taq II (TaKaRa, catalogue no. RR820A).

## Protein extraction and blood plasma isolation for NPY ELISA

Proteins were extracted from iWAT as previously described[48]. In brief, dissected iWAT was placed in homogenizing tubes filled with 300 μl 50 mg$^{-1}$ RIPA lysis buffer (Sigma, catalogue no. 20188) containing 50 μM DPP-IV inhibitor (Sigma, catalogue no. DPP4-010) and 500,000 IU ml$^{-1}$ aprotinin (Sigma, catalogue no. A6103-1MG) and homogenized using a Precellys 24 homogenizer. Lipid in the tissue homogenize was removed by centrifuging at 20,000× rcf at 4 °C for 15 min, and the clear portion retained. The process was repeated three times to ensure complete removal of lipid.

For terminal blood collection, mice euthanized by intraperitoneal injection of 10 μl g$^{-1}$ pentobarbital and blood was then collected from the left ventricle using a 25 G needle and a syringe coated with 100 mM EDTA. Blood was then placed in tubes with 5 μl of 100 mM EDTA and centrifuged at 1,000× rcf at 4 °C for 15 min to separate plasma from blood cells, and then the supernatant was transferred to fresh tubes with DPP-IV inhibitor (final concentration 50 μM) and aprotinin (final concentration 500,000 IU ml$^{-1}$). The concentration of NPY in mouse iWAT and blood plasma was determined using an NPY ELISA kit (Merck, catalogue no. EZRMNPY-27K). ELISA data were recorded using a FLUOstar Omega microplate reader with Omega v.6.20 software.

## Single-cell suspension and flow cytometry

Dissected adipose tissues were minced and digested in an enzyme mixture (for each sample, 500 μl of collagenase II (4 mg ml$^{-1}$, Sigma, catalogue no. C6885), 500 μl of hyaluronidase (5.3 mg ml$^{-1}$, equivalent to 40,000 U ml$^{-1}$, Sigma, catalogue no. H3884) and 5 μl of Dnase I (BioLabs, catalogue no. M0303L)) in a 37 °C water bath, with shaking, for 45 min, and samples pipetted every 10 min. Digestion was stopped by the addition of FACS buffer (PBS containing 2% fetal bovine serum (FBS)), and single-cell suspensions were collected by filtering the digested sample using EASYStrainer cell sieves (70 μm mesh; Greiner, catalogue no. 542070).

For preparation of samples for sorting or flow cytometry, cells were first treated with red blood cell lysis buffer (BioLegend, catalogue no. 420301) to remove red blood cells and then treated with Fc block (ThermoFisher, catalogue no. 14-9161-73) before staining with antibodies. Before immunolabelling for intracellular markers, cells were fixed and permeabilized using the eBioscience Intracellular Fixation and Permeabilization Buffer Set (ThermoFisher, catalogue no. 88-8824-00). Cell sorting was performed using a BD FACSAria III sorter, and flow cytometry data were acquired with either a BD FACSAria III sorter or a BD LSRFortessa X20 cytometer with BD FACSDiva v.6.0 software. Cytometry data were analysed using FlowJo v.10.8.1.

## Isolation and primary culture of mural cells and in vitro stimulation

Mural cells were isolated and cultured following a modified version of the protocol previously published[49]. In brief, mice were euthanized intraperitoneally with 10 μl g$^{-1}$ pentobarbital and perfused with 15 ml of PBS to remove blood. The iBAT SVF was isolated using the method described above. Cells were then seeded in 10 cm dishes with 10 ml of DMEM with 10% FBS for 2 days for the cells to attach to the bottom. Afterwards, 5 ml of the medium was replaced with mural cell-specific medium (DMEM with 2% FBS, supplied with 1% mural cell growth supplement (Science Cell, catalogue no. 1252)). Every other day, 5 ml of medium in dishes was refreshed until the cells grew confluent. For acquisition of pure mural cells, cells were labelled with an APC anti-CD104b antibody (1:100, BioLegend, catalogue no. 136007) and mural cells were sorted using the MojoSort magnetic sorting protocol with MojoSort buffer (BioLegend, catalogue no. 480017) and anti-APC nanobeads (BioLegend, catalogue no. 480071).

To perform in vitro coculturing experiments, cells were seeded at $5 \times 10^4$ per millilitre on glass coverslips and the indicated concentrations of PDGF-BB (R&B, catalogue no. 220-BB) and NPY (Cayman Chemical, catalogue no. CAY15071) were added to cells 24 h following cell seeding. Cells were collected 5 days later and either fixed with 4% PFA for imaging or lysed with Trizol for RNA extraction and qPCR.

## Cell lines

The RAW264.7 mouse macrophage cell line (Sigma, catalogue no. 91062702) was purchased without further authentication or testing for mycoplasma contamination. The complete medium for cell culture includes high-glucose DMEM with glutamate (Sigma, catalogue no. 41965039) and 10% FBS (Sigma, catalogue no. 12133 C).

The 3T3-L1 preadipocyte cell line was a gift from R. Klemm at the Department of Physiology, Anatomy and Genetics, University of Oxford, and used without further authentication or testing for mycoplasma contamination. The complete medium for cell culture includes high-glucose DMEM with glutamate (ThermoFisher, catalogue no. 41965039) and 10% calf serum (Sigma, catalogue no. 12133 C). For subculture of preadipocytes the medium was transferred, and cells were washed with 1 ml of prewarmed 0.05% trypsin (ThermoFisher, catalogue no. 25300054) and digested with 200 μl of trypsin at 37 °C for 5–10 min. Afterwards, digestion was stopped and cells resuspended using the complete medium.

## Induction to white and thermogenic adipocytes

For differentiation of 3T3-L1 cells to white adipocytes, cells were seeded in 12-well plates at a density of $1 \times 10^5$ per millilitre. The medium was refreshed until cells were confluent. Two days following cell confluence, induction medium (10% FBS, 500 μM 3-isobutyl-methylxanthine and 1 μM dexamethasone in DMEM) with or without 1 μg ml$^{-1}$ insulin was added to each well; 3 days later the induction medium was replaced with maintenance medium (DMEM with 10% FBS with or without 1 μg ml$^{-1}$ insulin). Subsequently the medium was refreshed with maintenance medium every 2 days. Total differentiation time for each well was 8 days. NPY treatments in experimental groups started when the induction medium was added and lasted throughout the whole differentiation process.

SVF and mural cells were differentiated to thermogenic adipocytes as previously reported[6,30]. SVF and mural cells were isolated and seeded in 12-well plates at a density of $1 \times 10^5$ per millilitre. When cells reached 95% confluency, beige cell induction medium (DMEM with 10% FBS, 125 μM indomethacin, 5 μM dexamethasone, 500 μM 3-isobutyl-methylxanthine and 0.5 μM rosiglitazone) was added to each well; 2 days later the induction medium was replaced with maintenance medium (DMEM with 10% FBS, 5 μg ml$^{-1}$ insulin and 1 nM triiodothyronine). Maintenance medium was refreshed every 2 days. Total differentiation time for each well was 8 days. NPY treatments in the experimental groups started following the addition of induction medium and lasted throughout the whole differentiation process. Oxygen consumption was measured using a Seahorse XF Analyzer.

## Cell proliferation assay

The cell proliferation assay was performed with the Click-iT EdU Cell Proliferation Kit (ThermoFisher, catalogue no. C10340) for imaging. Mural cells were seeded at $0.5 \times 10^5$ per millilitre in 12-well plates on coverslips and cultured overnight before experiments. Cells were then labelled with EdU and cultured with or without 1 μM NPY or 2 μM PD98059 (Cell Signaling, catalogue no. 9900 S) for 6 h in medium for mural cells (low-glucose DMEM with 2% FBS). Cells were then fixed with 4% PFA and permeabilizing buffer for 30 min each, and EdU was labelled with AF647 using the Click-iT Plus reaction cocktail. Following immunofluorescent staining with anti-DES, cells were imaged using a Zeiss LSM880 confocal microscope with Zen-black software (v.2.1), with a ×20/0.8 NA objective and voxel size of 0.52 μm ($x$), 0.52 μm ($y$)

and 1.00 μm ($z$). An area of 3,072 × 3,072 pixels was imaged for each biological replicate.

## scRNA-seq dataset analysis

Public scRNA-seq datasets were downloaded from Gene Expression Omnibus (GEO) and analysed using the following method. Cells with fewer than 200 unique detected genes or over 5% mitochondrial counts were discarded. After filtering, the gene $x$ cell matrix was normalized using 'NormalizeData()' in Seurat v.4.2.0 (ref. 50) in R (v.4.2.2). Data were then scaled using 'ScaleData()', and linear dimensional reduction performed by principal component analysis and calculation of UMAP coordinates for all cells using Seurat v.4.2.0. Cells were clustered using 'FindNeighbour()', with dimensions set to 15, and 'FindClusters()', with resolution set to 0.5. Each cluster was identified based on differentially expressed genes and known markers in the published literature[5,22,29,51–53].

## Figure quantification

Overlapping between TH, NPY and CD31 was calculated using JACoP[54], a Fiji plug-in[47]. To make this calculation, 472.33, 472.33 and 30 μm $xyz$ regions were randomly picked and maximally projected to $z$ using Fiji script. Labelled areas were automatically segmented using a threshold set by default, and overlapping percentages were calculated automatically by JACoP.

Innervation of NPY$^+$ axons was quantified using the 'Surface' tool in Imaris v.9.2. Labelled areas of NPY$^+$ axons and CD31$^+$ vessels in whole cleared iWAT were automatically segmented, and the innervation of NPY in vasculature was calculated as volume$_{NPY+}$/volume$_{CD31}^+$. The coverage of DES$^+$ mural cells in iWAT was calculated similarly, as volume$_{DES}^+$/volume$_{CD31}^+$. Confocal images showing NPY$^+$ innervation were quantified using Fiji with an automatic unbiased method. Area$_{NPY+}$ and area$_{CD31+}$ were segmented using the Otsu thresholding method and measured with the 'measure' program.

The percentage of EdU$^+$ cells was counted with an unbiased automatic method using 'detect particles' in Fiji[47]. Threshold was set automatically using the Otsu thresholding method.

The percentages of PDGFRα$^+$ and DES$^+$ cells were counted manually based on PDGFRα and DES signals. The percentages of TH$^+$ and NPY$^+$ neurons in the hypothalamus and ventral tegmental area were also counted manually. To calculate ganglionic mural cell coverage, 531, 531 and 30 μm $xyz$ views were randomly picked and maximally projected to $z$ using Fiji script. Labelled areas were segmented using a threshold set by default, and coverage was calculated as area $_{NPY}^+$/area $_{CD31}^+$.

The size of adipocytes in WAT and iBAT was quantified as previously described[55] using Fiji. In brief, WAT and iBAT were processed to paraffin-embedded sections (3 μm), stained with haematoxylin and eosin and scanned into digital images using a NanoZoomer-SQ Digital slide scanner (Hamamatsu). Images were randomly picked and the plug-in Analyse Particles was used to count the number and measure the size of adipocytes and droplets in iBAT. For each iBAT, 11,000 lipid droplets were measured.

## Statistics and reproducibility

GraphPad Prism v.9.5.0 was used for statistical analysis. Mean and s.e.m. were used to represent a sample. Chi-squared and paired two-tailed Student's $t$-tests were used to compare two samples; analysis of variance was used to compare multiple data points. No statistical methods were used to predetermine sample sizes.

All experiments were replicated at least twice with the same conclusion, and all in vivo experiments were replicated in at least two independent cohorts. The exact numbers of replicated experiments are stated in each figure legend. Samples and animals were randomly allocated to experimental groups and proceeded in experiments, and data were collected and analysed blind and ad hoc registered to groups or genotypes.

## Reporting summary

Further information on research design is available in the Nature Portfolio Reporting Summary linked to this article.

## Data availability

The public scRNA-seq dataset of sympathetic neurons can be viewed on the Linnarson Lab website (https://linnarssonlab.org/sympathetic/), with raw data deposited at GEO under accession no. GSE78845. The scRNA-seq datasets for iWAT and iBAT were deposited at GEO under accession nos. GSE154047 and GSE160585. Genome-Wide Association Study data of the association between genomic variants and metabolic traits are publicly available at the Common Metabolic Diseases Knowledge Portal (https://hugeamp.org/). All numerical data supporting this research are available within the paper and its Supplementary Information.

## Code availability

The code used for scRNA-seq analysis is deposited at GitHub (https://github.com/PLAVRVSO/scRNA-Seq-analysis-of-BAT-SVF).

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

**Acknowledgements** We thank I. Kalajzic for sharing *Npy^flox/flox* mice; M. Silveira and M. Roberts for sharing *Npy1r^Cre*;*Rosa26^tdTomato* mice; T. Hovarth for sharing NPY-GFP mice; R. Klemm for the 3T3-L1 cell line; V. Vyazovskiy and S. Wilcox for access to thermocamera and technical assistance; M. Dustin at the Kennedy Institute of Rheumatology for access to the light-sheet microscope, in particular C. Lagerholm for his technical assistance; the Dunn School of Pathology for confocal microscopy, in particular A. Wainman for his technical assistance; and K. Zhang for her technical assistance. We thank the National Institute of Science and Technology on Photonics Applied to Cell Biology for their assistance with the inverted microscope Axio Observer Z.1, in particular M. O. Baratti and A. T. P. Campos for their help. The icons of scissors in Extended Data Figs. 3a,g,i and 4c are from Flaticon (https://flaticon.com). We thank all members of Domingos Laboratory for ther discussions and advice on the project. Y.Z. receives a scholarship from Shineroad Industry Developments Co. Ltd. L.Y. is supported by the China Scholarship Council. J.C. is supported by the National Natural Science Foundation of China (no. 32100821). This research is funded by the Wellcome Trust–Howard Hughes Medical Institute International Research Scholar Award (no. 208576/Z/17/Z), the ERC consolidator award (no. ERC-2017 COG 771431), the Pfizer ASPIRE Obesity award and the Next Iteration of the Type2 Diabetes Knowledge Portal (no. 2UM1DK105554).

**Author contributions** Y.Z. designed and performed all experiments, with guidance from A.I.D., unless otherwise stated. Y.Z. wrote the first draft of the manuscript and A.D. wrote subsequent versions with Y.Z. L.Y. assisted Y.Z. and performed the experiments shown in Fig. 3 and Extended Data Figs. 1g–k, 3g–j, 4c–e, 5a,b,f,g, 6f,g and 9l,m, with guidance from Y.Z. and A.D. A.L.G.-F., B.B. and M.R.S. conducted experiments shown in Fig. 5c and Extended Data Fig. 9a–j, with guidance from L.A.V. G.S. helped Y.Z. conceive this project and provided advice on scRNA-seq data analysis. D.S.-O. made intellectual contributions to scRNA-seq dataset analysis. A.C. acquired light-sheet images at the Francis Crick Institute on samples cleared and stained by Y.Z. and shown in Extended Data Fig. 7a,e, and gave technical advice on light-sheet imaging. N.M.-S. helped with mouse colony management. C.L. imaged the first samples cleared and stained by Y.Z. (not shown in this manuscript). M.L.D. permitted access to the

Kennedy Imaging Facility. T.L.H. provided the brains of NPY-GFP mice shown in Extended Data Fig. 10a,c. S.K. gave intellectual contributions pertaining to beige fat biology. I.A. conducted the experiment shown in Extended Data Figs. 3a–c and 5c–e, with guidance from S.K. J.C. conducted the experiments shown in Extended Data Fig. 10g–m, with guidance from C.Z. L.Z. helped with the histology shown in Supplementary Fig. 3a,b.

**Competing interests** All authors declare no competing interests.

**Additional information**
**Correspondence and requests for materials** should be addressed to Ana I. Domingos.

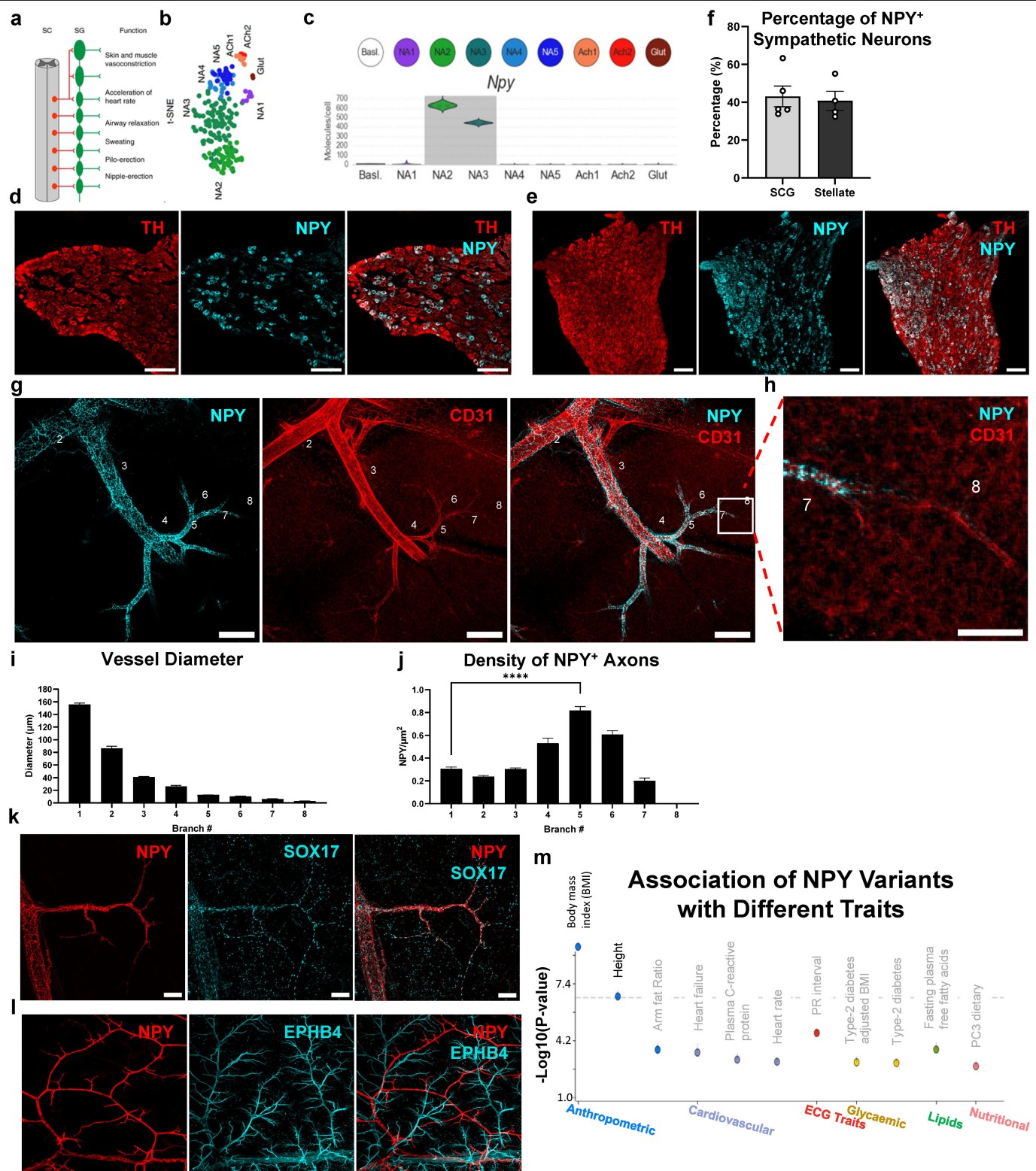

**Extended Data Fig. 1** | See next page for caption.

**Extended Data Fig. 1 | *Npy* is expressed by a subpopulation of sympathetic neurons, and NPY⁺ axons preferentially innervate the 5ᵗʰ order arteriole branches; NPY variants are significantly associated with human body mass index (BMI).** (**a**) Schematic showing the tissues innervated by each sympathetic ganglia[17]. (**b**) Single-cell RNA-Seq dataset of sympathetic ganglia chain published by Linnarsson Lab to show the clustering of neurons[17]. (**c**) Violin plot showing the expression of *Npy*[17]. (**d-e**) Confocal image of cryo-sectioned (d) superior-cervical ganglion (SCG) and (e) stellate ganglia (SG) of ND-treated WT male 8-week-old mice stained with anti-TH (red) and anti-NPY (cyan). Scale bar = 100 μm. (**f**) The percentage of NPY⁺ sympathetic neurons quantified based on images as in Extended Data Fig. 5d,e (n = 5 sections from 3 mice). (**g**) Confocal images of cleared adipose tissue stained with anti-NPY (cyan) and anti-CD31 (red). The numbers indicate the branch orders. Scale bar = 150 μm.

(**h**) Zoom-in of Extended Data Fig. 1g, scale bar = 50 μm. (**i-j**) (i) The diameters of vessels and (j) the density of NPY⁺ axons at each branch level quantified based on images as in Extended Data Fig. 1g (n = 6 views from 2 mice); NPY⁺ axon density on Branch #1 vs. #5, P < 0.0001. (**k**) Light-sheet images of cleared adipose tissue stained with anti-NPY (red) and anti-EPHB4 (cyan). Scale bar = 500 μm. (**l**) Confocal images of cleared adipose tissue stained with anti-NPY (red) and anti-SOX17 (cyan). Scale bar = 100 μm. (**m**) Plot showing associations of human NPY gene variants with different traits[1] (https://hugeamp.org/gene. html?gene=NPY). For **d-h**, representative images were shown from 3 experiments; for **k,l**, representative images were shown from 2 experiments. All values are expressed as mean ± SEM. Statistical comparisons were made using 2-tailed Student T-tests, ****P < 0.0001.

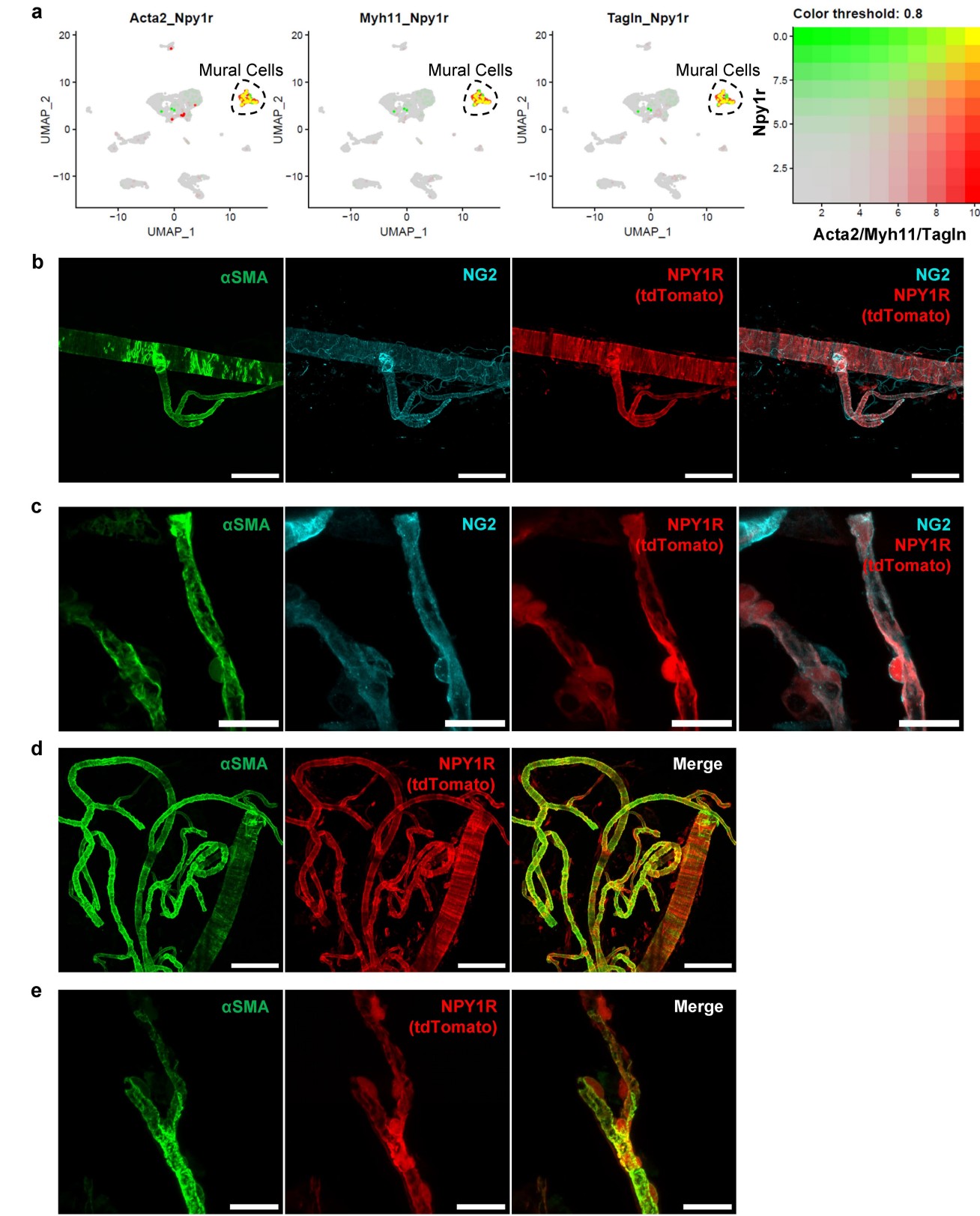

**Extended Data Fig. 2 | NPY1R-tdTomato is colocalised with NG2 and αSMA in mural cells.** (**a**) Embedding plot showing the co-expression of mural markers *Acta2* (αSMA), *Myh11*, and *Tagln* (red) with *Npy1r* (green). Co-expression is in yellow[22]. (**b**) Confocal images of vessels dissected from adipose tissue of Npy1r[Cre];Rosa26[tdTomato] mice stained with anti-αSMA (green), scale bar = 100 μm.

(**c**) Zoomed-in images of vessels as in Extended Data Fig. 2b, scale bar = 20 μm. (**d**) Confocal images of vessels dissected from Npy1r[Cre];Rosa26[tdTomato] mice stained with anti-αSMA (green) and anti-NG2 (cyan), scale bar = 100 μm. (**e**) Zoomed-in images of vessels as in Extended Data Fig. 2b, scale bar = 20 μm. For **b**-**e**, representative images were shown from 2 experiments.

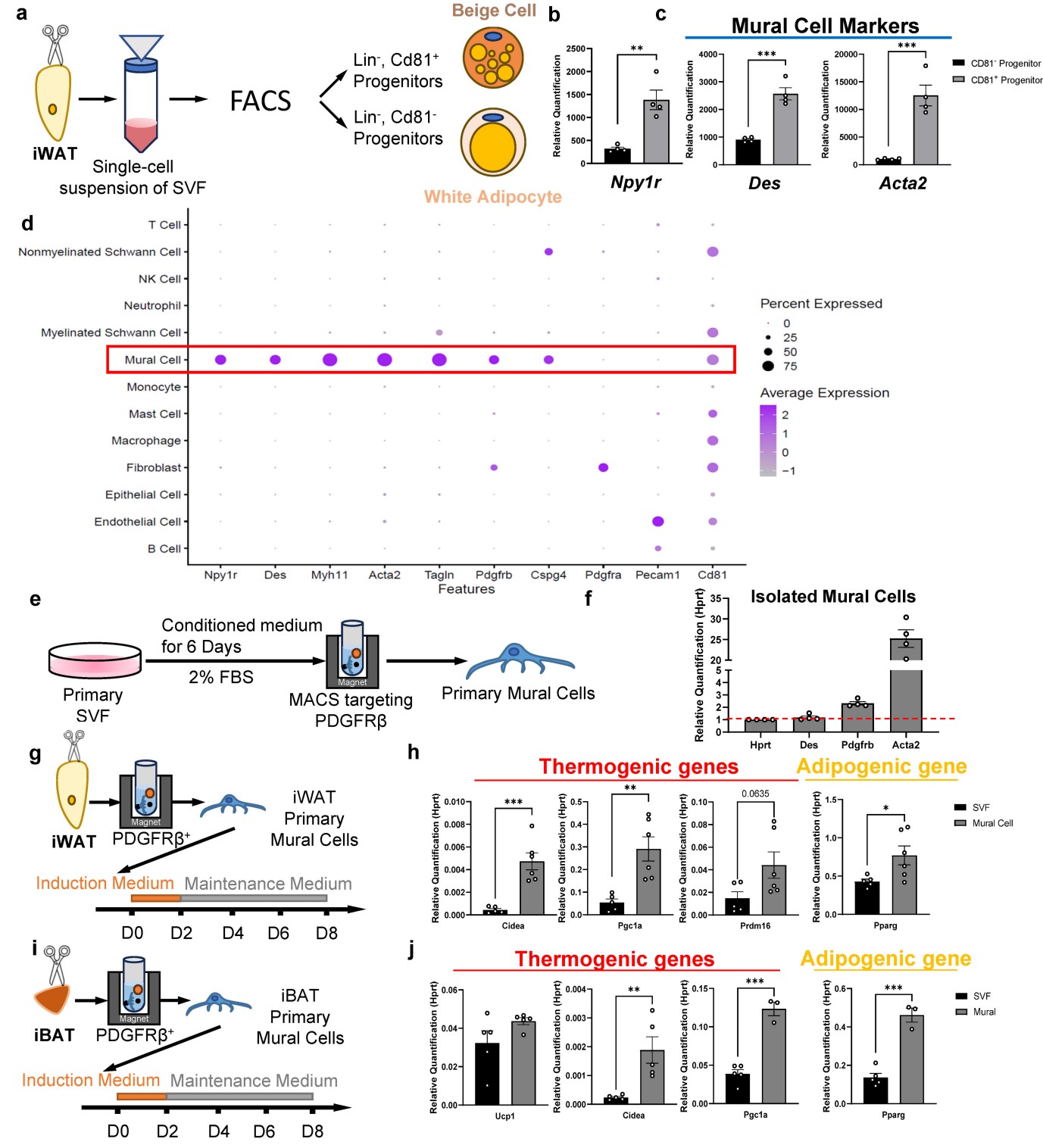

**Extended Data Fig. 3** | See next page for caption.

**Extended Data Fig. 3 | Previously identified progenitors of thermogenic adipocytes (Lin⁻/CD81⁺) in iWAT express *Npy1r* and mural cell markers, and mural cells isolated from both iWAT and iBAT can differentiate into thermogenic adipocytes.** (**a**) Schematic of isolating previously identified progenitors of thermogenic adipocytes (Lin⁻/CD81⁺) in iWAT[6]. (**b-c**) The expression level of (b) *Npy1r* (P = 0.0027), and (c) mural cell markers *Des* (P = 0.0003) and *Acta2* (P = 0.0009) in CD81⁻ vs. CD81⁺ progenitors (n = 4 biologically independent samples). (**d**) Dot plot showing the expression of *Npy1r, Des, Myh11, Acta2, Tagln, Pdgfrb, Cspg4, Pdgfra, Pecam1*, and *Cd81* in different cells in the SVF of mice iWAT. The size of the dot represents the percentage of cells expressing a certain gene, and the darkness of the dot represents the expression level. NPY1R⁺ mural cell clusters are highlighted with red rectangles. (**e**) Schematic of isolating mural cells from adipose tissue.

(**f**) The expression level of mural cell markers *Des, Pdgfrb*, and *Acta2* in isolated mural cells. The red dashed line indicates the expression level of reference gene *Hprt* (n = 4 biologically independent samples). (**g,i**) Schematic of isolating and differentiating mural cells from (g) iWAT and (i) iBAT. (**h,j**) The expression of thermogenic and adipogenic genes in adipocytes differentiated from the stromal-vascular fraction (SVF) and mural cells isolated from (h) iWAT (n = 5&6 biologically independent samples; *Cidea* P = 0.0006, *Pgc1a* P = 0.0037, *Pparg* P = 0.0.0364) and (j) iBAT (n = 5&5 biologically independent samples for *Ucp1* and *Cidea* P = 0.0071; n = 5&3 biologically independent samples for *Pgc1a* P = 0.0001 and *Pparg* P = 0.0003). All values are expressed as mean ± SEM. Statistical comparisons were made using 2-tailed Student T-tests, *p < 0.05, **p < 0.01, ***p < 0.001. The icon of scissors in **c,g,i** is from Flaticon (https://flaticon.com). *Hprt* was used as the reference gene for qPCR.

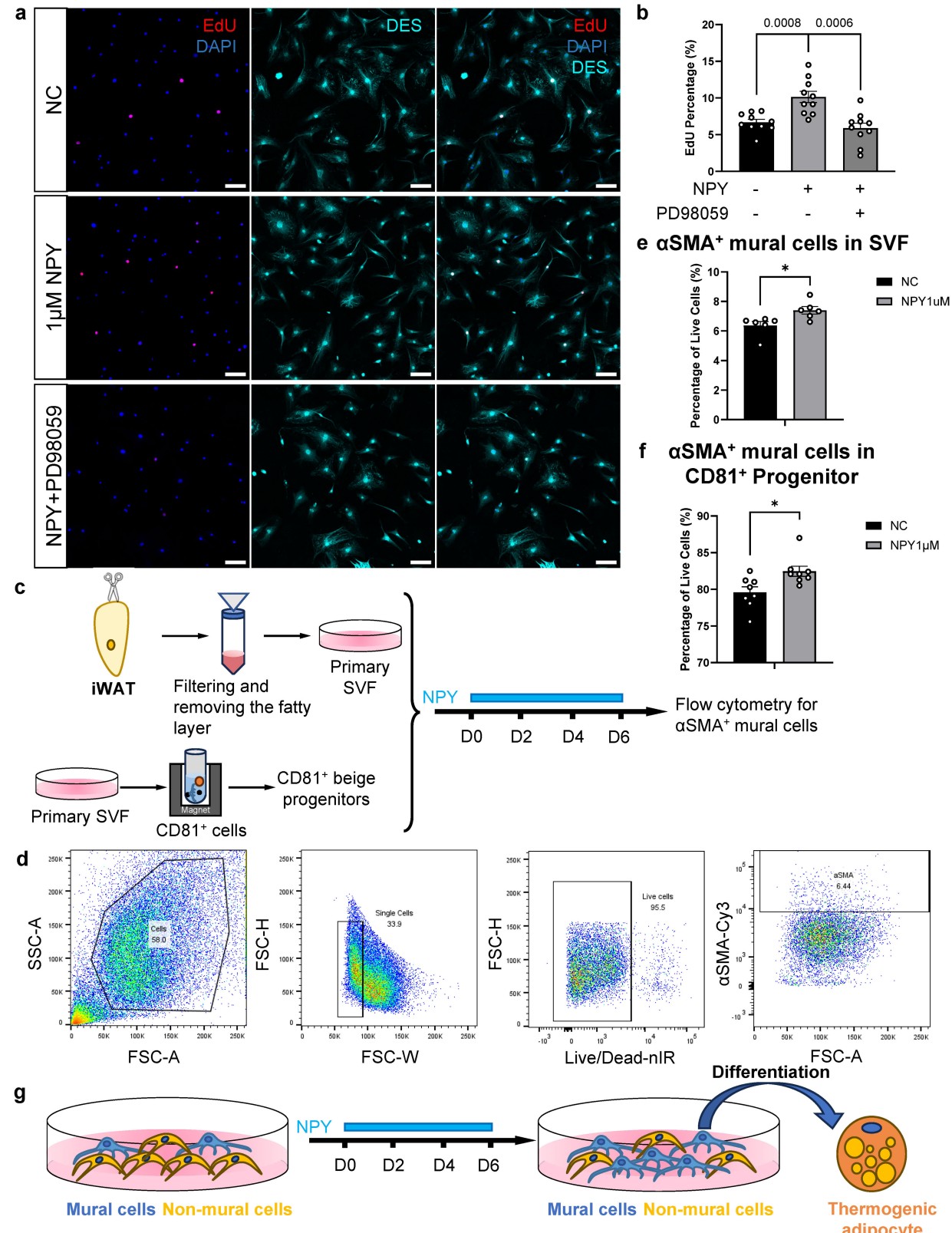

**Extended Data Fig. 4 | NPY promotes the proliferation of mural cells via ERK1/2 and increases the proportion of mural cells ex vivo.** (**a**) Confocal images of isolated mural cells treated with or without 1 μM NPY and 2 μM PD98059, an ERK inhibitor, stained with anti-DES (cyan) and DAPI (blue). EdU (red) was used to indicate proliferation. Scale bar = 80 μm. (**b**) The percentage of EdU⁺ cells quantified based on images as in Extended Data Fig. 4a (n = 6 biologically independent samples). (**c**) Schematic of isolating iWAT SVF and CD81⁺ cells for NPY treatment experiments and flow cytometry analysis. (**d**) Gating strategy for measuring the percentage of αSMA⁺ mural cells in cultured SVF and CD81⁺ cells by flow cytometry. (**e-f**) The percentage of αSMA⁺ mural cells in (e) SVF (n = 6 biologically independent samples, P = 0.0191) and (e) CD81⁺ cells (n = 8 biologically independent samples, P = 0.0141) after incubating with or without 1 μM NPY. (**g**) Schematic showing NPY treatment increases the percentage of mural cells in cultured SVF or CD81⁺ cells, and mural cells are the progenitor of thermogenic adipocytes. For **a**, representative images were shown from 3 experiments. All values are expressed as mean ± SEM. Statistical comparisons were made using 2-tailed Student T-tests, *p < 0.05. The icon of scissors in **c** is from Flaticon (https://flaticon.com).

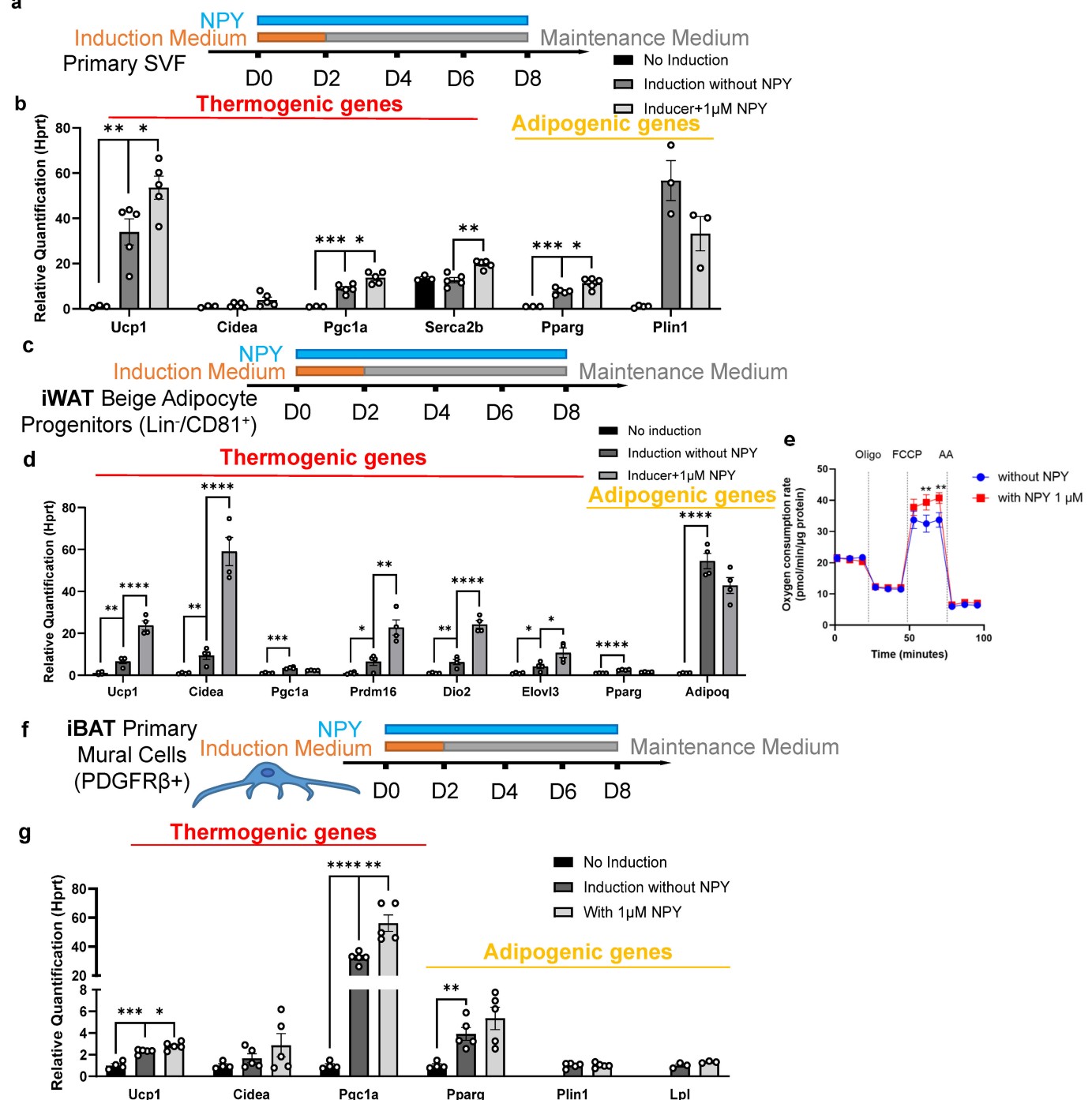

**Extended Data Fig. 5 | NPY facilitates the neogenesis of thermogenic adipocytes from iWAT SVF and mural progenitors isolated from both iWAT and iBAT.** (**a**) Schematic of differentiating primary stromal vascular fraction (SVF) of iWATs to thermogenic adipocytes. (**b**) The expression *Ucp1, Cidea, Pgc1a, Serca2b, Pparg, and Plin1* (n = 4&3&3 biologically independent samples for *Plin1* and n = 3&5&5 biologically independent samples for other genes) in thermogenic adipocytes differentiated from primary SVF with or without 1 μM NPY. No Induction vs. Induction: *Ucp1*, P = 0.0052; *Pgc1a*, P = 0.0006; *Pparg*, P = 0.0001. Induction vs. Induction+NPY: *Ucp1*, P = 0.0350; *Pgc1a*, P = 0.0113; *Serca2b*, P = 0.0012; *Pparg*, P = 0.0190. (**c**) Schematic of differentiating beige adipocyte progenitors (Lin⁻/CD81⁺) isolated from iWAT[6]. (**d**) The expression of thermogenic and adipogenic genes in adipocytes differentiated from iWAT beige adipocyte progenitors (Lin⁻/CD81⁺) with or without 1 μM NPY (n = 4 biologically independent samples). No Induction vs. Induction: *Ucp1*,

P = 0.0027; *Cidea*, P = 0.0043; *Pgc1a*, P = 0.0001; *Prdm16*, P = 0.0261; *Dio2*, P = 0.0052; *Elvol3*, P = 0.0246; *Pparg&Adipoq*, P < 0.0001. Induction vs. Induction+NPY: *Ucp1&Cidea&Dio2*, P < 0.0001; *Prdm16*, P = 0.0071; *Elvol3*, P = 0.0400. (**e**) Oxygen consumption in adipocytes differentiated from beige adipocyte progenitors (Lin⁻/CD81⁺) with or without 1 μM NPY (n = 5 biologically independent samples per group). At 60 min, P = 0.0077; at 70 min, P = 0.0056. (**f**) Schematic of differentiating iBAT mural cells (PDGFRβ⁺) to thermogenic adipocytes. (**g**) The expression of thermogenic and adipogenic genes in adipocytes differentiated from iBAT mural cells or SVF (n = 4&5&5). No Induction vs. Induction: *Ucp1*, P = 0.0002; *Pgc1a*, P < 0.0001; *Prdm16*, P = 0.0261; *Pparg*, P = 0.0040. Induction vs. Induction+NPY: *Ucp1, P = 0.0259; Pgc1a, P = 0.0072*. All values are expressed as mean ± SEM. Statistical comparisons were made using 2-tailed Student T-tests (**b,d,g**) or 2-way ANOVA (**e**), *p < 0.05, **p < 0.01, ***p < 0.001, ****p < 0.0001. *Hprt* was used as the reference gene for qPCR.

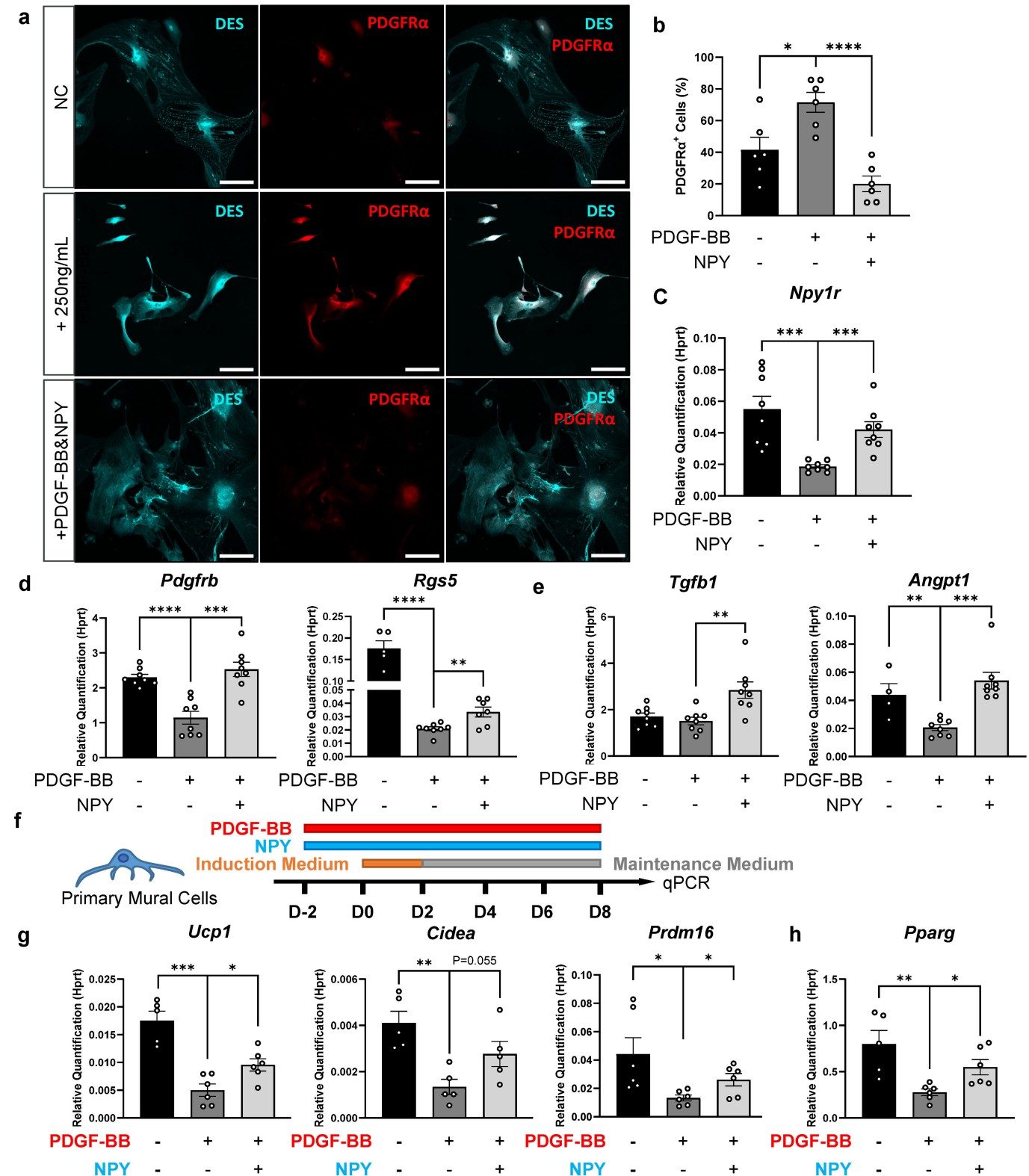

**Extended Data Fig. 6 |** See next page for caption.

**Extended Data Fig. 6 | NPY sustains mural cells and their ability to differentiate into thermogenic adipocytes against PDGF-BB.** (**a**) Confocal images of primary mural cells stained with anti-DES (cyan) and anti-PDGFRα (red), cultured under different conditions. Scale bar = 100 μm. (**b**) The percentage of PDGFRα⁺ cells under different stimulating conditions (n = 6 biologically independent samples). No drug vs. PDGF-BB, P = 0.0138; PDGF-BB vs. PDGF-BB + NPY, P < 0.0001. (**c-e**) The expression of (c) *Npy1r* (n = 8 biologically independent samples), (d) mural cell markers *Pdgfrb* (n = 8 biologically independent samples), and *Rgs5* (n = 5&8&8 biologically independent samples) and (e) secretory factors in maintaining vascular integrity including *Tgfb1* (n = 8) and *Angpt1* (n = 4&8&8 biologically independent samples) in primary mural cells cultured with or without 1 μM NPY or 250 ng/mL PDGF-BB. No drug vs. PDGF-BB: *Npy1r*, P = 0.0005, *Pdgfrb&Rgs5*, P < 0.0001; *Angpt1*, P = 0.0032. PDGF-BB vs. PDGF-BB + NPY: *Npy1r*, P = 0.0004; *Rgs5*, P = 0.0051; *Tgfb1*, P = 0.0043; *Angpt1*, P = 0.0001. (**f**) Schematic of differentiating mural cells with or without 1 μM NPY or 250 ng/mL PDGF-BB. (**g-h**) The expression of (g) thermogenic and adipogenic genes in adipocytes differentiated from mural cells with or without 1μM NPY or 250 ng/mL PDGF-BB (n = 5&6 biologically independent samples). No drug vs. PDGF-BB: *Ucp1*, P = 0.0001; *Cidea*, P = 0.0019; *Prdm16*, P = 0.0260; *Pparg*, P = 0.0045. PDGF-BB vs. PDGF-BB + NPY: *Ucp1*, P = 0.0154; *Prdm16*, P = 0.0240; *Pparg*, P = 0.0134. For **a**, representative images were shown from 3 experiments. All values are expressed as mean ± SEM. Statistical comparisons were made using 2-tailed Student T-tests, *p < 0.05. *Hprt* was used as the reference gene for qPCR.

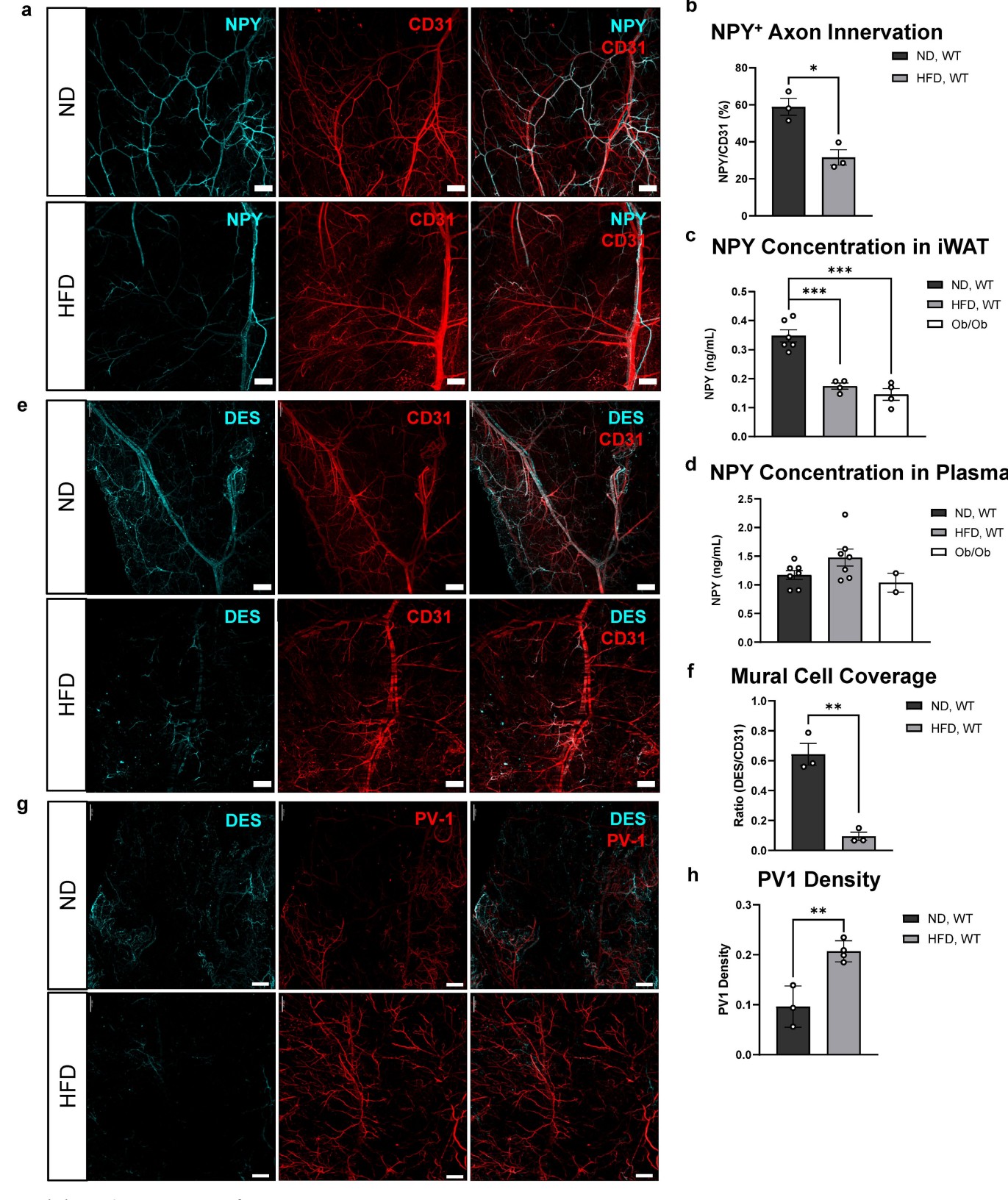

**Extended Data Fig. 7** | See next page for caption.

**Extended Data Fig. 7 | High-fat diet-induced obesity (DIO) depletes NPY⁺ innervation and mural cells and leads to vascular leakiness. (a)** Light-sheet images of cleared iWAT of normal diet (ND)-treated and high-fat diet (HFD)-treated 17-week-old male WT mice stained with anti-NPY (cyan) and anti-CD31 (red). Scale bar = 500 µm. **(b)** NPY⁺ innervation calculated as the ratio of NPY to CD31 quantified based on images as in Extended Data Fig. 7a (n = 3 mice, P = 0.0112). **(c,d)** The concentration of NPY in (c) iWAT (n = 5&4&4 mice) and (d) blood plasma of ND and HFD-treated 17-week-old male WT mice, and ND-treated Ob/Ob 17-week-old male mice (n = 6&7&2 mice). P = 0.0002 for both comparisons. **(e)** Light-sheet images of cleared iWAT of ND- and HFD-treated 17-week-old male WT stained with anti-DES (cyan) and anti-CD31 (red). Scale bar = 500 µm. **(f)** Mural cell coverage calculated as the ratio of DES⁺ cells to CD31⁺ cells quantified based on images as in Extended Data Fig. 7e (n = 3 mice, P = 0.0021). **(g)** Light-sheet images of iWATs of ND- and HFD-treated 17-week-old male mice stained with anti-DES (cyan) and anti-PV1 (red). Scale bar = 500 µm. **(h)** PV-1 density quantified based on images as in Extended Data Fig. 7g (n = 3&4 mice, P = 0.0053). For **a,e,g**, representative images are shown from 3 experiments. All values are expressed as mean ± SEM. Statistical comparisons were made using 2-tailed Student T-tests, *p < 0.05, **p < 0.01, ***p < 0.001, ****p < 0.0001. *Hprt* was used as the reference gene for qPCR.

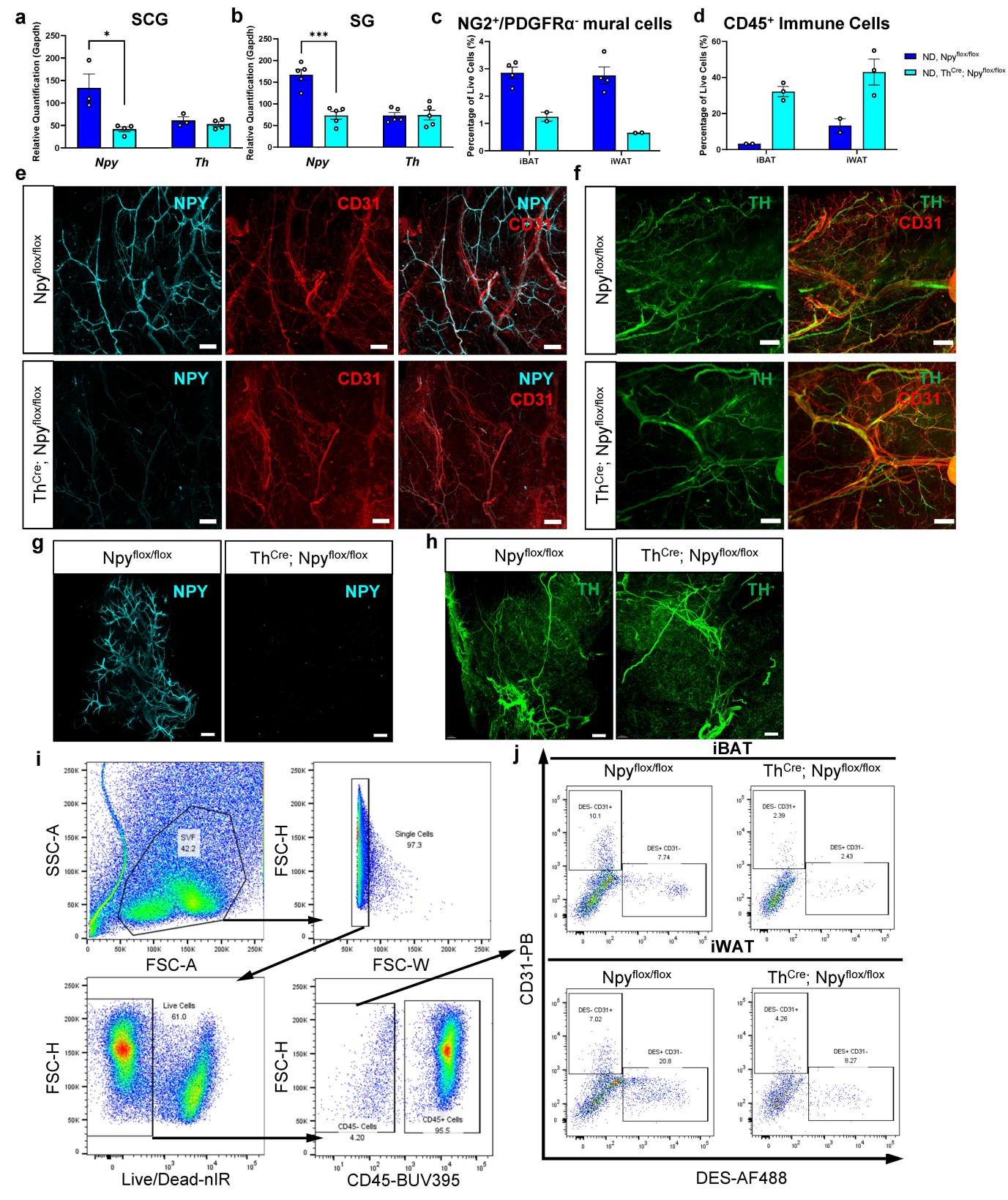

**Extended Data Fig. 8** | See next page for caption.

**Extended Data Fig. 8 | Th^Cre;Npy^flox/flox mice have depleted NPY from the sympathetic nervous system and unchanged sympathetic innervation, central or plasma NPY level, causing mural cell depletion.** (**a**,**b**) The expression of *Npy* and *Th* in (a) SCG (n = 3&4 mice; *Npy*, P = 0.0190) and (b) SG (n = 5 mice; *Npy*, P = 0.0003) of 12-week-old Th^Cre;Npy^flox/flox mice and Npy^flox/flox male mice. (**c**,**d**) The percentage of (c) PDGFRα^−/NG2^+ mural cells (n = 4&2 mice) and (d) CD45^+ immune cells (n = 2&3 mice) in iBATs and iWATs of 30-week-old, ND-treated mice measured by flow cytometry. (**e**,**f**) The image of cleared iWAT of 12-week-old Th^Cre;Npy^flox/flox mice and Npy^flox/flox male mice stained with (e) anti-NPY (cyan) or (f) anti-TH (green) and anti-CD31 (red). Scale bar = 500 µm. (**g**,**h**) Light-sheet image of cleared iBAT of 12-week-old Th^Cre;Npy^flox/flox mice and Npy^flox/flox male mice stained with (g) anti-NPY (cyan), scale bar = 500 µm, or (h) anti-TH (green), scale bar = 200 µm. (**i**) The gating strategy of flow cytometry to quantify the percentage of CD45^+ immune cells DES^+ mural cells, used for Fig. 4e. (**j**) The representative flow cytometry analysis of the percentage of mural cells (DES^+) in the iBAT (top) and iWAT (bottom) of HFD-treated male Th^Cre;Npy^flox/flox mice and Npy^flox/flox mice. For **e**-**h**, representative images are shown from 2 experiments. All values are expressed as mean ± SEM. Statistical comparisons were made using 2-tailed Student T-tests, *p < 0.05, **p < 0.01, ***p < 0.001, ****p < 0.0001. *Gapdh* was used as the reference gene for qPCR.

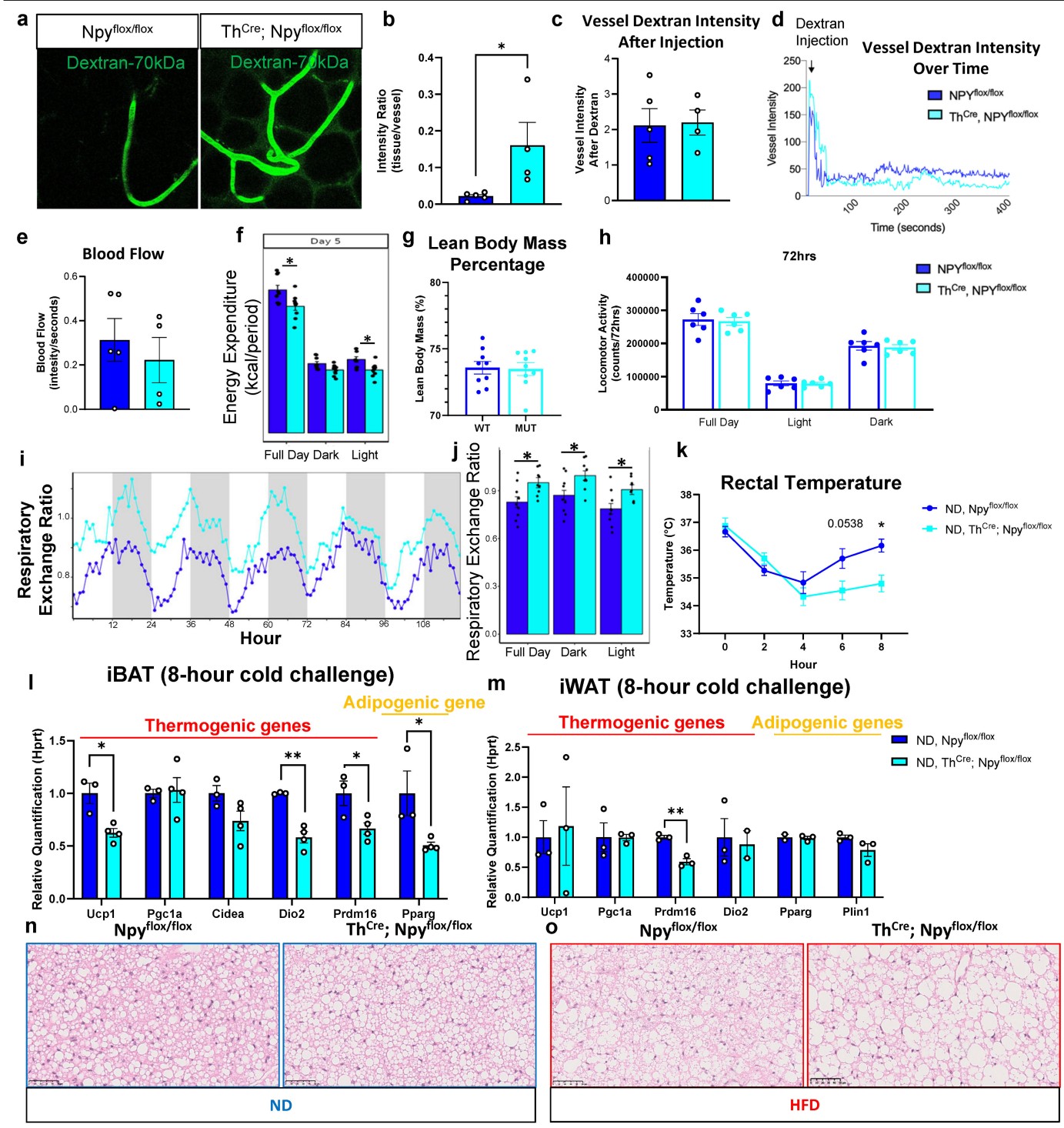

**Extended Data Fig. 9** | See next page for caption.

**Extended Data Fig. 9 | Loss of sympathetic NPY increases vascular leakiness, decreases energy expenditure (EE) and cold tolerance, inhibits thermogenesis and beiging, and increases respiratory exchange ratio (RER) and lipid droplet size.** (**a**) Intravital microscopy of epididymal WATs of 23-week-old male Npy[flox/flox] and Th[Cre];Npy[flox/flox] mice 12 min after i.v. injection of 70 kDa Dextran at the orbital plexus. (**b**) Fluorescence intensity ratio between tissue parenchyma and vessel calculated based on images as in Extended Data Fig. 9a (n = 5&4 mice). P = 0.0390. (**c**) Dextran intensity in the vessels of eWAT right after i.v. injection (n = 5&4 mice). (**d**) Dextran intensity in vessels over time measured with a 2-photon microscope. Arrow indicates the time when Dextran was injected. (**e**) Blood flow calculated using the average of the raw intensity of Dextran inside the artery over 100 s right after the stabilization of the Dextran signal following administration (n = 5&4 mice). (**f**) The quantification of daily energy expenditure (EE) recorded and quantified using an indirect calorimetry system (n = 9 mice). Full day, P = 0.0416; dark, P = 0.0288. (**g**) The percentage of lean body weight (LBM) of 8-week-old male mice measured by MiniSpec MRI (n = 9). (**h**) The locomotor activity measured using Multitake cage (n = 6 mice). (**i**) Respiratory exchange ratio (RER) of 8-week-old mice recorded using a metabolic cage. (**j**) The quantification of (i) (n = 9 mice). Full day, P = 0.0173; light, P = 0.0136; dark, P = 0.0243. (**k**) Rectal temperature of 20-week-old Npy[flox/flox] and Th[Cre];Npy[flox/flox] mice under cold exposure recorded using a rectal probe every hour (n = 3&4 mice). 8 h, P = 0.0158. (**l,m**) The expression of thermogenic and adipogenic genes in (l) iBATs (n = 3&4 mice) and (m) iWATs (n = 3 mice) of cold-challenged Npy[flox/flox] and Th[Cre];Npy[flox/flox] mice. iBAT: *Ucp1*, P = 0.0106; *Dio2*, P = 0.0012; *Prdm16*, P = 0.0385; *Pparg*, P = 0.0412. iWAT: *Prdm16*, P = 0.0024. (**n,o**) Histological staining of BATs of (n) ND- and (o) HFD-treated 17-week-old male Th[Cre];Npy[flox/flox] and Npy[flox/flox] mice. Scale bar = 50 μm. For **a**, representative images are shown from 4 experiments. All values are expressed as mean ± SEM. Statistic comparisons were made by 2-tailed Student T-tests (**b,l,m**) or 2-way ANOVA (**f,j,k**), *p < 0.05, **p < 0.01, ***p < 0.001, ****p < 0.0001. *Hprt* was used as the reference gene for qPCR.

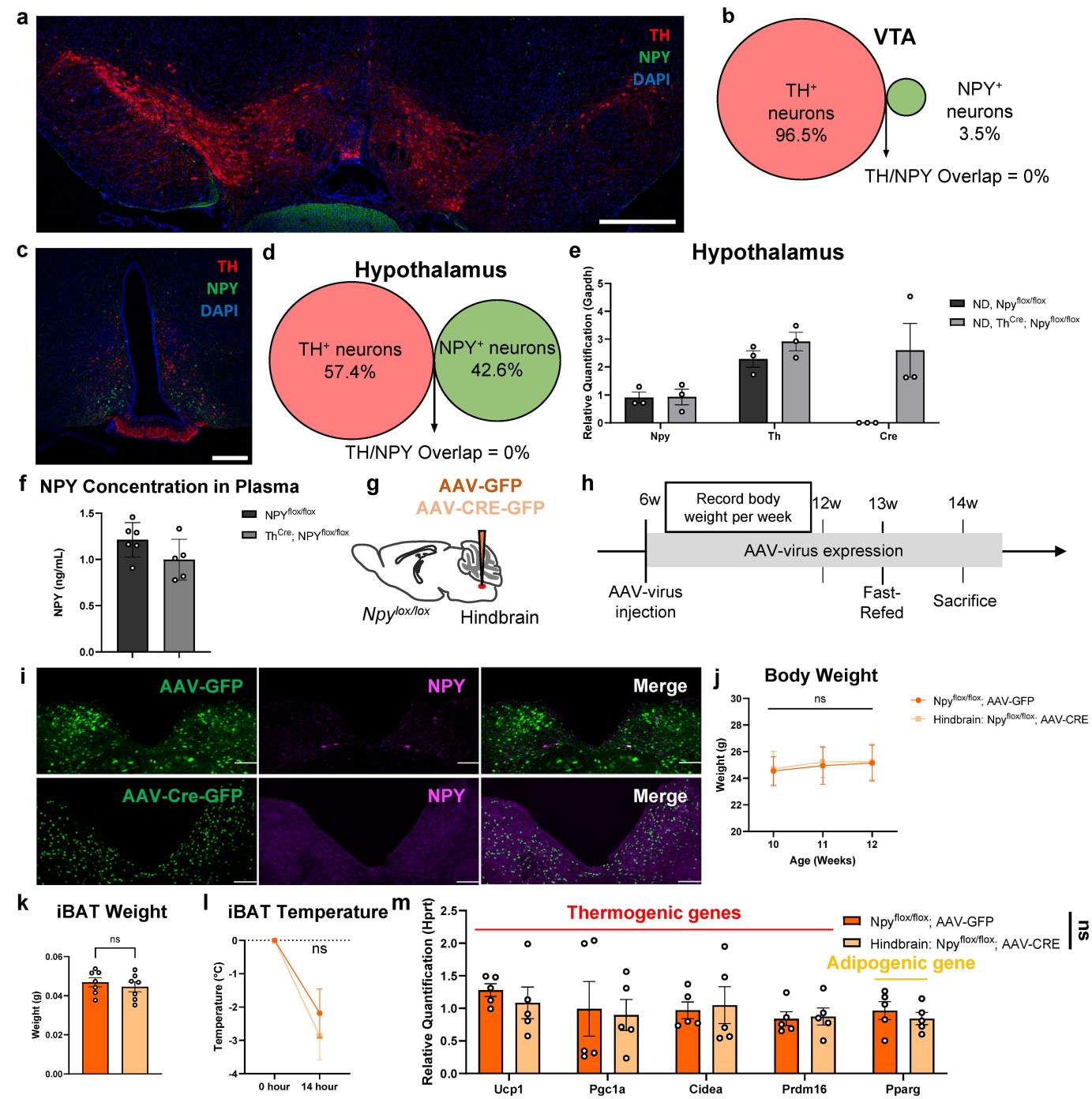

**Extended Data Fig. 10 | Ablating NPY from sympathetic neurons using Th^Cre;Npy^flox/flox mice does not affect NPY levels in VTA, hypothalamus or plasma, and ablating NPY from the hindbrain does not affect body weight, BAT weight or thermogenesis.** (a,c) Confocal image showing a slice of (a) ventral tegmental area (VTA, Scale bar = 500 μm) and (c) arcuate nucleus (Arc, Scale bar = 200 μm) of the hypothalamus of an 8-week-old NPY-GFP male mouse stained with anti-TH (red), anti-GFP (green), and DAPI (blue). (b,d) Quantification of TH/NPY-GFP overlap in (b) VTA and (d) hypothalamus (n = 3 mice). (e) The expression of *Npy, Th*, and *Cre* in the hypothalamus of 12-week-old Th^Cre;Npy^flox/flox mice and Npy^flox/flox male mice (n = 3). (f) NPY concentration in the blood plasma of 12-week-old Th^Cre;Npy^flox/flox and Npy^flox/flox male mice (n = 6&5). (g) Schematic of knocking out NPY in the hindbrain by AAV

injection. (h) Schematic of metabolic phenotyping of mice with NPY knocked out from the hindbrain (Hindbrain: Npy^flox/flox; AAV-CRE-GFP) and WT (Npy^flox/flox; AAV-GFP) mice. (i) Confocal images of the sliced hindbrain of Hindbrain: Npy^flox/flox; AAV-CRE-GFP and WT mice stained with anti-NPY (magenta). Scale bar = 100 μm. (j) Weekly body weight of ND-treated male Hindbrain: Npy^flox/flox; AAV-CRE-GFP and WT mice (n = 7). (k) iBAT weight of 14-week-old ND-treated male Hindbrain: Npy^flox/flox; AAV-CRE-GFP and WT mice (n = 7). (l) iBAT temperature change after a 14-hour fast (n = 7). (m) The expression levels of thermogenic and adipogenic genes in the iBAT of RT-housed ND-treated male Hindbrain: Npy^flox/flox; AAV-CRE-GFP and WT mice (n = 5). For **a,c,i**, representative images are shown from 3 experiments. All values are expressed as mean ± SEM. *Hprt* was used as the reference gene for qPCR.

# Reporting Summary

## Statistics

For all statistical analyses, confirm that the following items are present in the figure legend, table legend, main text, or Methods section.

| n/a | Confirmed | |
|---|---|---|
| ☐ | ☒ | The exact sample size (*n*) for each experimental group/condition, given as a discrete number and unit of measurement |
| ☐ | ☒ | A statement on whether measurements were taken from distinct samples or whether the same sample was measured repeatedly |
| ☐ | ☒ | The statistical test(s) used AND whether they are one- or two-sided<br>*Only common tests should be described solely by name; describe more complex techniques in the Methods section.* |
| ☒ | ☐ | A description of all covariates tested |
| ☐ | ☒ | A description of any assumptions or corrections, such as tests of normality and adjustment for multiple comparisons |
| ☐ | ☒ | A full description of the statistical parameters including central tendency (e.g. means) or other basic estimates (e.g. regression coefficient) AND variation (e.g. standard deviation) or associated estimates of uncertainty (e.g. confidence intervals) |
| ☐ | ☒ | For null hypothesis testing, the test statistic (e.g. *F*, *t*, *r*) with confidence intervals, effect sizes, degrees of freedom and *P* value noted<br>*Give P values as exact values whenever suitable.* |
| ☒ | ☐ | For Bayesian analysis, information on the choice of priors and Markov chain Monte Carlo settings |
| ☒ | ☐ | For hierarchical and complex designs, identification of the appropriate level for tests and full reporting of outcomes |
| ☒ | ☐ | Estimates of effect sizes (e.g. Cohen's *d*, Pearson's *r*), indicating how they were calculated |

*Our web collection on statistics for biologists contains articles on many of the points above.*

## Software and code

Policy information about availability of computer code

| Data collection | EE and RER were measured using an indirect calorimetry system (Panlab; Harvard Apparatus; LE405 Gas Analyzer and Air Supply & Switching). Locomotor activity was measured using an LE001 PH Multitake Cage(Panlab; Harvard Apparatus) with a COMPULSE v1.0 Software (PanLab). Thermo-imaging data were recorded using Optris thermo-cameras (Optris PI 160 with standard 61 lens, Optris GmbH, Berlin, Germany) with an Optris PIX Connect software (rel. 2.0.6, Optris GmbH)  or an FOTRIC 225s infrared camera (ED Fig.10l). Intravital 2 photon-microscopy was conducted with Optical microscope CARS (Coherent Anti-stokes Raman Scattering), TPEF (Two-photon excitation fluorescence) and Bright-field microscope, simultaneously, using confocal LSM 780-NLO Zeiss in the inverted microscope Axio Observer Z.1 (Carl Zeiss AG, Alemanha). Images were acquired using Zen Black v2.1 (Zeiss LSM880)  and Inspector Pro (Miltenyi Biotec UltraMicroscope II Lightsheet Microscope). ELISA data were recorded using a FLUOstar Omega microplate reader. Flow cytometry data were collected using BD FACSDiva v6.0 (BD x20 Fortessa). qPCR data were acquired using StepOneTM qPCR system or a CFX96 Real-Time PCR Detection System (Bio-Rad) (ED Fig. 10m). Oxygen consumption of cultured adipocytes was measured using a Seahorse XF Analyzer. |
|---|---|
| Data analysis | EE and RER were analyzed using CalR Web-based Analysis Tool for Indirect Calorimetry Experiments (v1.3). Mouse body composition was recorded and analyzed using Minispec LF50 (Brucker). Thermo-image data in ED Fig. 10l were anlysed using FOTRIC software (v5.0.8.214). Images were analysed using Imaris v9.9.1, Fiji v1.53t and JaCoP v2.1.1. qPCR data were analysed using a StepOne Software v2.3. scRNA-seq datasets were analysed using Seurat v4.2.0 in an R v4.2.2 environment with the code deposited at Github (https://github.com/PLAVRVSO/scRNA-Seq-analysis-of-iBAT-SVF).  Flow cytometry data were analysed using FlowJo v10.8.1. Statistical analysis was performed using GraphPad Prism v9.5.0. |

For manuscripts utilizing custom algorithms or software that are central to the research but not yet described in published literature, software must be made available to editors and reviewers. We strongly encourage code deposition in a community repository (e.g. GitHub). See the Nature Portfolio guidelines for submitting code & software for further information.

## Data

Policy information about availability of data

All manuscripts must include a data availability statement. This statement should provide the following information, where applicable:
- Accession codes, unique identifiers, or web links for publicly available datasets
- A description of any restrictions on data availability
- For clinical datasets or third party data, please ensure that the statement adheres to our policy

The public scRNA-Seq dataset of sympathetic neurons can be browsed on the Linnarson Lab website (https://linnarssonlab.org/sympathetic/), with raw data deposited on the Gene-Expression Omnibus (GEO) under the accession number GSE78845. The scRNA-Seq datasets of iWAT and iBAT were deposited on GEO under the accession numbers GSE154047 and GSE160585. The GWAS data of the association between genomic variants and metabolic traits is publicly available at Common Metabolic Diseases Knowledge Portal (https://hugeamp.org/). All the numeric data for supporting this research are available within the paper and the Supplementary information.

## Human research participants

Policy information about studies involving human research participants and Sex and Gender in Research.

| | |
|---|---|
| Reporting on sex and gender | N/A |
| Population characteristics | N/A |
| Recruitment | N/A |
| Ethics oversight | N/A |

Note that full information on the approval of the study protocol must also be provided in the manuscript.

## Field-specific reporting

Please select the one below that is the best fit for your research. If you are not sure, read the appropriate sections before making your selection.

☒ Life sciences ☐ Behavioural & social sciences ☐ Ecological, evolutionary & environmental sciences

For a reference copy of the document with all sections, see nature.com/documents/nr-reporting-summary-flat.pdf

## Life sciences study design

All studies must disclose on these points even when the disclosure is negative.

| | |
|---|---|
| Sample size | No statistical methods were used to pre-determine the sample sizes. The sample sizes were determined empirically based on previous studies and literature using the same experimental paradigm. (W Zeng et al. Cell 2015, Y Wang et al. Nature 2022) |
| Data exclusions | No data were excluded, except for mice with deteriorating health issues during the experiments or tissue samples contaminated. |
| Replication | All experiments are replicated at least twice with the same conclusion. All in vivo experiments were replicated in at least 2 independent cohorts. The exact numbers of replicated experiments are stated in each figure legend. |
| Randomization | Samples/animals were randomly allocated to experimental groups and proceeded in experiments. |
| Blinding | Data were collected and analysed blind, and post hoc registered to treatments and genotypes. |

## Reporting for specific materials, systems and methods

We require information from authors about some types of materials, experimental systems and methods used in many studies. Here, indicate whether each material, system or method listed is relevant to your study. If you are not sure if a list item applies to your research, read the appropriate section before selecting a response.

## Materials & experimental systems

| n/a | Involved in the study |
|---|---|
| ☐ | ☒ Antibodies |
| ☐ | ☒ Eukaryotic cell lines |
| ☒ | ☐ Palaeontology and archaeology |
| ☐ | ☒ Animals and other organisms |
| ☒ | ☐ Clinical data |
| ☒ | ☐ Dual use research of concern |

## Methods

| n/a | Involved in the study |
|---|---|
| ☒ | ☐ ChIP-seq |
| ☐ | ☒ Flow cytometry |
| ☒ | ☐ MRI-based neuroimaging |

# Antibodies

| | |
|---|---|
| Antibodies used | The following primary antibodies were used for immunofluorescent staining: rat anti-CD31 (BioLegend, 102501 ,MEC13.3,1:100 dilution), rat anti-PLVAP (BioLegend, 120503, MECA32, 1:100 dilution), rabbit anti-DES (Abcam, AB15200, 1:500 dilution), rabbit anti-NPY (Cell Signalling, D7Y5A, 1:500 dilution), rabbit anti-NPY (Abcam, AB30914, 1:500 dilution), chicken anti-TH (Aves Labs, TYH73787982, 1:500 dilution), rabbit anti-TH (Sigma, Ab152, 1:500 dilution), mouse anti-NPY1R (Santa Cruz, sc-393192, 1:200 dilution), rat anti-PDGFRα (BioLegend, 135902, APA5, 1:200 dilution), rabbit anti-TAGLN (Abcam, AB14106, 1:250 dilution), Cy3 anti-αSMA (Sigma, C6198, 1A4, 1:250 dilution), goat anti-SOX17 (R&D, AF1924, 1:250), goat anti-EPHB4 (R&D, AF3034, 1:250), rabbit anti-UCP1 (Abcam, Ab10983, 1:500).<br>The following antibodies were used for FACS and flow cytometry: AF700 anti-CD45 (BioLegend, 103128), BUV395 anti-CD45 (BD Horizon, 564279), Pacific Blue anti-CD31 (BioLegend, 102421), APC anti-PDGFRa (BioLegend 135907), AF488 anti-NG2 (Sigma, MAB5384A4), and AF488 anti-DES (Abcam, AB185033, Y66). All antibodies for FACS and flow cytometry were diluted at 1:500. LIVE/DEAD™ Fixable Near-IR Dead Cell Stain Kit (1:1000, ThermoFisher, L10119) was used for live/dead staining.<br>APC anti-CD104b antibody (1:100, BioLegend, 136007) was used for magnetic sorting. |
| Validation | Rat anti-CD31 (BioLegend, 102501 ,MEC13.3): validated in mouse tissues by K Cheung et al. PNAS 2015, E5815–E5824. https://doi.org/10.1073/pnas.1509627112<br>Rat anti-PLVAP (BioLegend, 120503, MECA32): validated in mouse tissues by S Carloni et al. Science 2021, 374(6566), 439–448. https://doi.org/10.1126/science.abc6108<br>Rabbit anti-DES (Abcam, AB15200): validated in mouse tissues by W Yang et al. Hepatology 2021 and J Chang et al. Nature Medicine 2017, 23(4), 450–460. https://doi.org/10.1038/nm.4309<br>Rabbit anti-NPY (Cell Signalling, D7Y5A): validated in mouse tissues by Rahman T. U. et al. Oncotarget, 8(32), 53450–53464. https://doi.org/10.18632/oncotarget.18519<br>Chicken anti-TH (Aves Labs, TYH73787982): validated in mouse tissues by B. Sofia Beas et al. Nature communications, 11(1), 6218. https://doi.org/10.1038/s41467-020-19980-7<br>Rabbit anti-TH (Sigma, Ab152): validated in mouse tissues by D. Rycko et al. PNAS 2013, 110(34), E3235–E3242. https://doi.org/10.1073/pnas.1301125110<br>Mouse anti-NPY1R (Santa Cruz, sc-393192): validated in mouse tissues by K Xu et al. Wound Repair and Regeneration 2018 and X. Kang et al. American Society for Bone and Mineral Research, 35(7), 1375–1384. https://doi.org/10.1002/jbmr.3991<br>Rat anti-PDGFRα (BioLegend, 135902, APA5): validated in mouse tissues by MW Hogarth et al. Nature Communication 2019, 10(1), 2430. https://doi.org/10.1038/s41467-019-10438-z<br>Rabbit anti-TAGLN (Abcam, AB14106): validated in mouse tissues by C. De Bono et al. Nature communications, 14(1), 1551. https://doi.org/10.1038/s41467-023-37015-9<br>Cy3-conjugated anti-αSMA (Sigma, C6198): validated in mouse tissues by S. Ock, Cell death & disease, 12(7), 688. https://doi.org/10.1038/s41419-021-03965-5<br>Goat anti-SOX17 (R&D, AF1924) and Goat anti-EPHB4 (R&D, AF3034, 1:250) are validated in mouse tissues by M. Corada et al. Nature communications, 4, 2609. https://doi.org/10.1038/ncomms3609<br>Rabbit anti-UCP1 (Abcam, Ab10983): validated in mouse tissues by Y. Oguri et al. Cell, 182(3), 563–577.e20. https://doi.org/10.1016/j.cell.2020.06.021<br>AF700 anti-CD45 (BioLegend, 103128), Pacific Blue anti-CD31 (BioLegend, 102421),  and APC anti-PDGFRa (BioLegend 135907) are validated in our previous paper by E. Habermann et al. Immunity, 57(1), 141–152.e5. https://doi.org/10.1016/j.immuni.2023.11.006<br>Other conjugated antibodies have been validated by the commercial vendors with information demonstrated on their websites:<br>BUV395 anti-CD45 (BD Horizon, 564279): https://www.bdbiosciences.com/en-gb/products/reagents/flow-cytometry-reagents/research-reagents/single-color-antibodies-ruo/buv395-rat-anti-mouse-cd45.564279<br>AF488 anti-NG2 (Sigma, MAB5384A4): https://www.sigmaaldrich.com/GB/en/product/mm/mab5384a4<br>AF488 anti-DES (Abcam, AB185033, Y66): https://www.abcam.com/en-gb/products/primary-antibodies/alexa-fluor-488-desmin-antibody-y66-cytoskeleton-marker-ab185033<br>APC anti-CD104b antibody (1:100, BioLegend): https://www.biolegend.com/en-gb/products/apc-anti-mouse-cd140b-antibody-6441 |

# Eukaryotic cell lines

Policy information about cell lines and Sex and Gender in Research

| | |
|---|---|
| Cell line source(s) | 3T3-L1 cell line was established and described in a previously published report (H Green et al. Cell 1975) and was a gift from Robin Klemm (CAC Freyre et al. Molecular Cell 2019, from American Type Culture Collection). RAW264.7 was purchased from Sigma (Sigma 91062702-1VL). |
| Authentication | No further authentication was performed for 3T3-L1 and RAW264.7. |
| Mycoplasma contamination | No further test for Mycoplasma was performed for 3T3-L1 and RAW264.7. |

| Commonly misidentified lines (See ICLAC register) | No commonly misidentified lines were used. |
|---|---|

## Animals and other research organisms

Policy information about studies involving animals; ARRIVE guidelines recommended for reporting animal research, and Sex and Gender in Research

| Laboratory animals | ThCre mice (B6.Cg-7630403G23RikTg(Th-cre)1Tmd/J; stock no. 008601), Cx3cr1GFP/+ mice (Cx3cr1tm1Litt/LittJ; stock no. 008451) were purchased from the Jackson Lab (JAX). Npyflox/flox mice were a donation from Ivo Kalajzic Lab at the Department of Reconstructive Science, University of Connecticut45 under MTA. Tissues of NPY-GFP mice (B6.FVB-Tg(Npy-hrGFP)1Lowl/J) are from Tamas Horvath Lab at Brandy Memorial Laboratory, Yale University. Tissue of Npy1rCre; Rosa26tdTomato mice were from Professor Michael Roberts at Department of Otolaryngology-Head and Neck Surgery, University of Michigan. Sympathetic neuron-specific NPY-cKO mice were generated by crossing ThCre mice with Npyflox/flox mice. Diet-induced obesity (DIO) was achieved by feeding mice an HFD (Diet Research, D12492) when they were 7 weeks old, and lasted for 10 weeks. The body weight of each mouse and the food consumption in each cage were recorded weekly. All the mice were group housed in standard housing at controlled room temperature (21-23ºC) and 50% humidity under a 12-12-hour light-dark cycle and given access to diet and water ad libitum, and. |
|---|---|
| Wild animals | No wild animals were used in this study. |
| Reporting on sex | Both sexes were used for in vivo studies. The sexes of mice were indicated in each figure legend. |
| Field-collected samples | No field-collected samples were used in this study. |
| Ethics oversight | All experimental procedures were performed on living animals in accordance with the United Kingdom ANIMALS ACTS 1986 under the project license (PPL number: P80EDA9F7) and personal licenses granted by the United Kingdom Home Office. Ethical Approval was provided by the Ethical Review Panel at the University of Oxford. |

Note that full information on the approval of the study protocol must also be provided in the manuscript.

## Flow Cytometry

### Plots

Confirm that:

☒ The axis labels state the marker and fluorochrome used (e.g. CD4-FITC).

☒ The axis scales are clearly visible. Include numbers along axes only for bottom left plot of group (a 'group' is an analysis of identical markers).

☒ All plots are contour plots with outliers or pseudocolor plots.

☒ A numerical value for number of cells or percentage (with statistics) is provided.

### Methodology

| Sample preparation | Dissected iWATs and BATs were minced and digested in the enzyme mixture (for each sample, 500μL Collagenase II (4 mg/mL, C6885), 500μL Hyaluronidase (5.3 mg/mL=40000 U/mL, H3884), and 5 μL Dnase I (BioLabs, M0303L)) in a 37 ℃-water bath shaking for 45 minutes, and samples were pipetted every 10 minutes. Digestion was stopped by adding FACS buffer (PBS containing 2% FBS), and single-cell suspension was collected by filtering the digested sample using EASYStrainer cell sieves with 70μm mesh (Greiner, 542070). To prepare the samples for sorting or flow cytometry, cells were first treated with red-blood-cell lysis buffer (BioLegend, 420301) to remove red-blood cells and Fc block (ThermoFisher, 14-9161-73) before staining with antibodies. Before immunolabelling for intracellular markers, cells were fixed and permeabilised using the eBioscience Intracellular Fixation and Permeabilization Buffer Set (ThermoFisher, 88-8824-00). |
|---|---|
| Instrument | BD FACSAria III sorter or LSRFortessa X20 cytometer. |
| Software | Flow cytometry data were collected using BD FACSDiva v6.0 and analysed using FlowJo v10.8.1. |
| Cell population abundance | At least 10000 singlet cells were harvested. |
| Gating strategy | For all experiments, the SSC-A/FSC-A gating was used to find viable cells in the starting cell population, and singlet cells were identified using the FSC-H/FSC-W gating. Isotype controls were used to distinguish marker-positive events from background events. |

☒ Tick this box to confirm that a figure exemplifying the gating strategy is provided in the Supplementary Information.

