## [Peer Review File · Nature]

Manuscript Title: Sympathetic NPY protects from obesity by sustaining thermogenic fat

Reviewer Comments & Author Rebuttals

Reviewer Reports on the Initial Version:

Referees' comments:

Referee #1 (Remarks to the Author):

This study investigates the role of NPY-expressing sympathetic neurons in regulating mural/smooth muscle cells and thermogenic programs in brown and beige adipose tissue. Imaging studies show that NPY+ fibers innervate vessels in brown and white adipose tissue and that the density of these nerves is reduced by obesity. Interestingly, obesity or ablation of NPY in sympathetic neurons leads to reductions in mural cell density. The NPY-deleted mice also exhibit whitened brown fat depots and increased weight gain upon high fat diet feeding. The paper introduces novel concepts regarding the role of NPY in regulating mural cells. However, the effects of this pathway in regulating vessel and adipose tissue function (including browning) are not sufficiently delineated. The connections between the loss of NPY, decrease in mural cells, reductions in thermogenic genes and obesity are unclear. Overall, the studies are interesting but too preliminary for publication, especially at this level. Many of the stated conclusions in the paper are not well supported. Additional more specific comments follow:

1. How does NPY signaling in mural cells regulate the maintenance of these cells? Does the pathway regulate proliferation, turnover, survival of mural cells?
2. (Related to #1) Have the authors considered whether the loss of mural/SMC cells affect vessel function, blood flow? It seems possible (and in my opinion, more likely) that deficits/changes in vessel function could modulate adipose tissue metabolic programs.
3. Fig. 3. The cell culture experiments with SVF are not convincing. First, there is a very modest effect of NPY on adipocyte differentiation. Also, the effects seem to be mainly on adipocyte differentiation rather than browning (ie. the increase in UCP1 follows an increase in PPAR γ , a general adipogenesis gene).
4. (Related to #3) The effect of NPY on mural/SMC adipogenesis/browning should be assessed using purified cell populations. The SVF is likely to contain mostly fibroblasts and the responses observed may not be relevant.
5. There is no direct evidence that NPY deletion reduces the beige/brown fat differentiation of mural/SMCs, which is a key conclusion in the paper. There is also uncertainty about whether these cells contribute to brown/beige fat cell differentiation, especially under homeostatic conditions.
6. Are smooth muscle cells and pericytes similarly regulated in obesity and NPY KO mice? There are

statements in the paper to the effect that SMCs are a type of mural cell. However, different types of mural cells (ie. pericytes and SMCs) have specific gene signatures, and these populations should be independently evaluated. Eg. Do SMCs (Myh11+) express NPY receptors and also decline in the KO mice?

7. Fig. 5- Are there effects of the NPY KO in skeletal muscle? Is the vasculature also altered in skeletal muscle, which could affect whole body metabolism?

8. Fig. 5. Thermogenic activity/energy expenditure should be assessed in the NPY KO mice. It is unclear if thermogenesis is actually reduced.

Referee #2 (Remarks to the Author):

The manuscript from Zhu and colleagues describe the results from a series of studies investigating the role of the sympathetic nervous system and neuropeptide Y(NPY) in regulating adipocyte precursors from progenitor cells. The studies provide evidence of a previously unidentified role of NPY secretion from sympathetic terminals in the “being” and browning of adipose tissue. The studies are novel and will be of wide interest. The only major concerns are technical and need to be addressed.

The biggest concern is the characterization of the genetic models. In particular, the authors note the overlap of TH and NPY was exclusively in “sympathetic” neurons. While they did assess overlap in VTA and hypothalamus, there are other populations of NPY expressing cells including brainstem neurons, many of which are considered “pre-sympathetic”. i.e., these neurons are known to project to the spinal cord and to pre-autonomic regulatory sites in the hypothalamus.

Some TH-cre models have been reported to have a baseline phenotype do to altered catecholamine levels. More detailed about the Cre alone controls would be welcome..

In addition, a bit more information about the floxed NPY mouse could be included in the supplemental data.

Finally, I prefer that each panel in a figure have a letter label.

Referee #3 (Remarks to the Author):

The manuscript # 2023-02-01744A seeks to address the putative mechanisms by which Neuropeptide Y (NPY) regulates thermogenic adipocytes and obesity. The authors perform a variety of approaches including whole adipose tissue staining and clearing followed by IF staining together with scRNAseq analysis and identify NYP-positive axons to be more closely associated with the vasculature. The scRNAseq data show that mural cells express *Nypr1* and that at the protein levels *Nypr1* is expressed by capillary pericytes. The authors show that this association is reduced in high fat diet (HFD) adipose tissue. In vitro, NYP seems to affect mural cell differentiation or maturation. Finally, genetic ablation of NYP in sympathetic neurons, whitens the adipose tissue and makes mice more susceptible to high fat diet. Although the authors provide some compelling data by which NPY operating on mural cells may regulate the adipose tissue differentiation, there are several concerns with the paper that need to be addressed to strengthen the mechanism and the premise for the study.

Major concerns -

1. While the authors show that NPY is required in sympathetic neurons for brown adipose tissue development, mural cell coverage and diet-induced obesity, the mechanisms are not explored. Does NYP-NYPR1 signaling in mural cells triggers a secondary signaling cascade that leads to changes in adipose tissue? What are these signals? The authors show that some of the markers of pericytes are upregulated in vitro with the addition of PDGF-BB and NYP including *Angpt1*. Is *Angpt1-Tie2* signaling driving this process, or are there additional pathways driving this process? This mechanism needs to be explored in depth and it is currently missing from the paper.
2. *Nypr1* receptor is expressed by all mural cells, including arterial SMCs, venous SMCs, capillary pericytes. Throughout the paper, the authors use only one marker (Desmin) to stain or sort these cells. This is NOT sufficient. The authors need to use additional mural cell markers including PDGFR-beta, NG2, and α -SMA (SMCs). This is critical throughout the manuscript since the focus of the paper is on the mural cells. The current IF images are very low magnification and therefore it is difficult to distinguish the mural cells. There are several mouse strains where mural cells are labelled with eGFP, DsRed (See <https://pubmed.ncbi.nlm.nih.gov/26119027/>). The authors should use these strains to strengthen their findings.
3. In Figure 1, the image shown in panel C do not reflect the quantification shown in panel F. Can the authors provide more details on how the analysis of vascular innervation is performed using a quantitative unbiased methodology? In that figure, it looks like a lot of NYP sympathetic innervation is around larger caliber vessels (arteries or veins). The authors need to distinguish arteries from capillaries and veins using established markers and analyze whether NPY axons are associated more with arteries, veins or capillaries.
4. In Figure 2, the authors show nicely by scRNAseq data that NYPR1 is in mural cells. The authors need to validate these data by showing NPY1R staining with a mural marker such as Desmin, NG2, PDGR-beta or α -SMA (vSMCs). What is the fraction of mural cells expressing the receptor by IF?
5. In Figure 5A-C, the authors should use an independent marker from NYP to label sympathetic

axons to ensure that the differences between ND and HFD are not due to differences in general innervation. How is the axonal - vessel association quantified in this figure? What are the differences in mural cell numbers between ND and HFD?

6. For the in vitro studies with mural cells Fig 5G-K, mural cells change dramatically their transcriptional identity when they are cultured with 10% FBS. Have the authors tried to culture cells with lower serum (2% FBS)? For the in vitro analysis, the authors should perform bulk RNAseq analysis in order to properly analyze the effects of addition of PDGF-BB and NYP on mural cell identity, survival and phenotypic switching. This analysis should be critical for the paper.

7. Along those lines, it is important to perform single cell RNA seq of mural cells in NYP genetic ablation in sympathetic neurons in order to understand the mechanisms by which this signaling pathway may impact brown adipose tissue formation.

Minor:

1. Please provide tables for the single cell RNAseq analysis.
2. In Fig. S1, what proportion of Th neurons express NYP?
3. In Fig. S5, please increase the number of mice to N=4-5 for the analysis shown in panel A.
4. The authors show stain for an axonal marker in mice where NYP is deleted in sympathetic neurons to show that innervation of the adipose tissue is not affected (Fig. S7).
5. In Fig. S8 the authors show that NYP depletion causes reduction of CD31. How do the authors explain these data?

Author Rebuttals to Initial Comments:

All the changes in the text of the revised manuscript are marked in pink font.

Referees' comments and Replies:

Referee #1 (Remarks to the Author):

This study investigates the role of NPY-expressing sympathetic neurons in regulating mural/smooth muscle cells and thermogenic programs in brown and beige adipose tissue. Imaging studies show that NPY+ fibres innervate vessels in brown and white adipose tissue and that the density of these nerves is reduced by obesity. Interestingly, obesity or ablation of NPY in sympathetic neurons leads to reductions in mural cell density. The NPY-deleted mice also exhibit whitened brown fat depots and increased weight gain upon high fat diet feeding. The paper introduces novel concepts regarding the role of NPY in regulating mural cells. However, the effects of this pathway in regulating vessel and adipose tissue function (including browning) are not sufficiently delineated. The connections between the loss of NPY, decrease in mural cells, reductions in thermogenic genes and obesity are unclear. Overall, the studies are interesting but too preliminary for publication, especially at this level. Many of the stated conclusions in the paper are not well supported. Additional more specific comments follow:

R1.1 How does NPY signalling in mural cells regulate the maintenance of these cells? Does the pathway regulate proliferation, turnover, survival of mural cells?

A: *We appreciate the reviewer's input, which encouraged us to delve deeper into the regulation of mural cell maintenance by NPY signalling. Through our investigation using EdU as a proliferation marker, we were able to illustrate that NPY has the capacity to stimulate mural cell proliferation (Figure 3A-B). This effect is expected because, on the one hand, previous studies have demonstrated that the activation of Pi3K and MAPK is downstream of NPY1R (PMID:24917132, PMID:25817573); on the other hand, it is also known that the activation of Pi3K and MAPK can promote the proliferation of mural cells (PMID:32466671, PMID:11942415).*

Figure 3A-B. NPY promotes the proliferation of mural cells in culture.
 (A) Confocal images of isolated mural cells treated with or without NPY stained with anti-DES (cyan) and DAPI (blue). EdU (red) was used to indicated proliferation. Scale bar=80 µm. (B) The percentage of EdU⁺ cells quantified based on images as in A (n=6).

R1.2 (Related to #1) Have the authors considered whether the loss of mural/SMC cells affect vessel function, blood flow? It seems possible (and in my opinion, more likely) that deficits/changes in vessel function could modulate adipose tissue metabolic programs.

A: We followed up on the reviewer's suggestion to assess vessel function; Our images revealed increased vascular permeability in DIO mice (**Supplementary Figure 9G&H**), as indicated by PV1 staining, a marker for vascular leakage. (PMID 26564856).

Supplementary Figure 9G-H. High-fat diet-induced obesity (DIO) leads to vascular leakiness.

(G) Light-sheet images of iWATs of ND- and HFD-treated 17-week-old male mice stained with anti-DES (cyan) and anti-PV1 (red). Scale bars=500 μ m. (H) PV-1 density quantified based on images as in G (n=3&4).

The reviewer's observation is accurate, as supported by our new findings in **Figure 4** below. It is evident that eliminating NPY from sympathetic neurons not only leads to mural cell depletion but also results in the increased vascular permeability that has been previously reported in the brain (**PMID:20944625**, **PMID:33375813**), and which we validate in adipose tissues by staining against PV1, an endothelial marker of vascular leakiness (**PMID 26564856**) - as depicted in **Figure 4C-D**. Our data using $Th^{Cre}; Npy^{flox/flox}$ mice further shows that sympathetic-derived NPY is required in preventing inflammatory infiltration within the adipose tissues (**Figure 4E**).

To summarise, NPY maintains the number and function of mural cells, protecting vascular integrity and preventing immune infiltration into adipose tissue (**Schematic Figure 1**).

D NG2⁺/PDGFR^α⁻ mural cells

E CD45⁺ Immune Cells

Figure 4C-E. Loss of NPY from sympathetic neuron decreases mural cells and increases vascular leakiness and immune infiltration

(C) Light-sheet image of cleared iWAT of 30-week-old, ND-treated mice stained against PV1, a marker of endothelial leakiness (red) and the mural marker TAGLN (cyan). Scale bar=500 μ m. Scale bars of the zoomed-in images are 200 μ m. (D-E) The percentage of (D) NG2⁺/PDGFR^α⁻ mural cells and (E) CD45⁺ immune cells in BATs, iWATs, and vWATs of 30-week-old, ND-treated mice measured by FACS.

Rebuttal Only Schematic figure 1. Loss of sympathetic axon-derived NPY promotes loss of mural cells, which is known to drive of vascular leakiness (PMID 26564856), and concomitant immune infiltration onto adipose tissue

As for blood flow/pressure, a recent preprint showed that deletion of NPY from sympathetic neurons results in no changes in blood pressure (PMID:37546870), possibly owing to by various redundant neuroendocrine mechanisms acting in tandem (sympathetic vs RAA axis).

R1.3 Fig. 3. The cell culture experiments with SVF are not convincing. First, there is a very modest effect of NPY on adipocyte differentiation. Also, the effects seem to be mainly on adipocyte differentiation rather than browning (i.e., the increase in UCP1 follows an increase in PPAR γ , a general adipogenesis gene).

(Related to #3) The effect of NPY on mural/SMC adipogenesis/browning should be assessed using purified cell populations. The SVF is likely to contain mostly fibroblasts and the responses observed may not be relevant.

A: We followed the reviewer's suggestion of testing the effect of NPY on purified mural cells, instead of SVF. Using the purification pipeline shown in **Supplementary Figure 6A**, we confirmed that isolated mural cells are more potent in differentiating into thermogenic adipocytes than SVF (**Supplementary Figure 6C-D**).

Supplementary Figure 6A-D. Purified mural cells are progenitors of thermogenic adipocytes

(A) Schematic of isolating mural cells from adipose tissue. (B) The expression level of mural cell markers *Des*, *Pdgfrb*, and *Acta2*. Red dashed line indicates the expression level of reference gene *Hprt*. (C-D) The expression level of (C) thermogenesis genes *Cidea*, *Pgc1a*, and *Prdm16*, and (D) beiging gene *Pparg* in adipocytes differentiated from the stromal-vascular fraction of adipose tissue and isolated mural cells. All values are expressed as mean \pm SEM, * p <0.05, ** p <0.01, *** p <0.001, **** p <0.0001, Student T-tests.

To address the reviewer's concern that the effects were attributed to adipocyte differentiation, we show in **Supplementary Figure 6E-H** that only beiging genes and thermogenesis genes are affected by NPY, while adipogenesis marker *Plin1* is not affected by NPY in thermogenesis genes differentiated from SVF. We further demonstrate in **Supplementary Figure 7A-C** that NPY does not affect adipogenesis using NPY1R⁺ 3T3-L1 preadipocytes. Since NPY can promote the proliferation of mural cells (**Figure 3A-B**), the effect of NPY on SVF is due an enrichment of the mural cell fraction, which gives rise to more thermogenic adipocytes.

Supplementary Figure 6E-H. NPY facilitates the differentiation of SVF to thermogenic adipocytes.

(E) Schematic figure showing in vitro method to differentiate primary stromal vascular fraction (SVF) of iWATs to thermogenic adipocytes. (F-G) The expression of (F) being gene *Pparg* ($n=3&5&5$), and (G) thermogenesis genes *Pgc1a*, *Ucp1*, *Serca2b* ($n=3&5&5$), and (H) adipogenesis gene *Plin1* ($n=4$), in thermogenic adipocytes differentiated from primary SVF. All values are expressed as mean \pm SEM, * $p<0.05$, ** $p<0.01$, *** $p<0.001$, **** $p<0.0001$, Student T-tests.

Supplementary Figure 7A-C. NPY does not affect general adipogenesis. (A) The expression of *Npy1r*, *Npy2r*, and *Npy5r* in 3T3-L1 preadipocyte cell line ($n=3$). (B-C) The expression of adipogenesis markers *Pparg* and *Plin1* in differentiated 3T3-L1 preadipocytes induced by induction medium (B) with or (C) without 1 $\mu\text{g}/\text{mL}$ insulin ($n=9, 8, 8, 9$ & 7 for *Pparg*, $n=9, 9, 9, 9$ & 7 for *Plin1*). The concentration of NPY is indicated in the plot. All values are expressed as mean \pm SEM, * $p<0.05$, ** $p<0.01$, *** $p<0.001$, **** $p<0.0001$, Student T-tests.

To further mitigate the concerns of the reviewer we sought the expertise of our collaborator, Professor Shingo Kajimura, a specialist in thermogenic progenitors. He isolated and purified thermogenic adipocyte progenitors and confirmed their expression of *Npy1r* along with mural cell markers (Figure 3C-E). Through his experiments, it was established that the addition of NPY during the differentiation process results in an enhanced maximal respiratory capacity

(Figure 3F-G). This increase is attributed to NPY's capacity to augment the population of thermogenic adipocyte progenitors.

Figure 3C-E. Progenitors of beige cells express *Npy1r* and mural cell markers.

(C) Schematic figure showing the isolation of CD81⁺ and CD81⁻ progenitors from the SVF of iWAT. (D-E) Expression of (D) *Npy1r*, and (E) mural cell markers *Des*, and *Acta2* in the CD81⁺ or CD81⁻ progenitor fractions. All values are expressed as mean ± SEM, **p*<0.05, ***p*<0.01, ****p*<0.001, *****p*<0.0001, Student T-tests.

Figure 3F-G. NPY facilitates beiging in vitro by increasing the maximal respiratory capacity.

(F) Schematic of Kajimura's protocol for differentiating isolated Lin^{-} , $CD81^{+}$ progenitors. (G) Oxygen consumption in adipocytes differentiated from $CD81^{+}$ progenitors with or without NPY ($n=5$). All values are expressed as mean \pm SEM, * $p<0.05$, ** $p<0.01$, *** $p<0.001$, **** $p<0.0001$, Student T-tests.

R1.5 There is no direct evidence that NPY deletion reduces the beige/brown fat differentiation of mural/SMCs, which is a key conclusion in the paper. There is also uncertainty about whether these cells contribute to brown/beige fat cell differentiation, especially under homeostatic conditions.

A: Several papers demonstrate that cells expressing mural markers including *Des*, *Myh11*, and *Acta2* are progenitors of thermogenic adipocytes, even under homeostatic conditions (See **Figure 2** in **PMID:32615086**, **PMID:33846638**). Furthermore, our collaborator Shingo Kajimura confirmed that the $CD81^{+}$ progenitors of thermogenic adipocytes, which he identified and published in **PMID:32615086**, are $NPY1R^{+}$ mural cells as they highly express *Npy1r* and mural cells markers *Des* and *Acta2* (aka aSMA), while the $CD81^{-}$ progenitors of white adipocytes are not mural cells (**Figure 3C-E**).

To further address the reviewer's concern, we dissected vessels from the BAT of ND-treated RT-housed $Npy1r^{Cre}; Rosa26^{tdTomato}$ lineage-tracing mice and did IF staining. The image demonstrates that the mural cells in BAT are labelled with *tdTomato*, which is consistent with our finding that mural cells express *Npy1r* (**Figure 2H**), and that a subgroup of $UCP1^{+}$ thermogenic adipocytes are also labelled with *tdTomato* (**Figure 2I**). Since *Npy1r* is only expressed by mural cells in BAT (**Figure 2C-D**), we reason that the *tdTomato*-labelled adipocytes are differentiated from $NPY1R^{+}$ mural cells.

We hope that these additional experiments and references to literature address the reviewer's concerns, and are sufficient to demonstrate $NPY1R^{+}$ mural cells give rise to beige cells, even at homeostatic conditions.

Figure 2H-I. Thermogenic adipocytes can be lineage-traced to NPY1R⁺ cells.

(H-I) Confocal images of vessels dissected from the BAT of ND-treated, RT-housed 12-week-old *Npy1^{Cre}; Rosa26^{tdTomato}* mice stained with anti-UCP1 (cyan), Scale bar = 100 μ m. (I) Zoomed-in images of H, scale bar = 50 μ m.

Schematic Figure 2. Schematic figure of our findings demonstrating that NPY1R⁺ mural cells are progenitors of beige cells, and that NPY promotes the proliferation of NPY1R⁺ mural cells.

Lastly, the use of a conditional KO is a means to ascertain direct causality *in vivo*: we have demonstrated that the deletion of NPY from sympathetic axons results in the whitening of BAT (**Figure 5A**), the downregulation of thermogenesis and beiging genes (**Figure 5B-C**), and a depletion of mural cells from adipose tissue (**Figure 4A-F**). We considered knocking out NPY1R from mural cells, which perhaps the reviewer would consider as more direct evidence, but we were worried about off-target effects linked to feeding behaviour since NPY1R is also expressed by mural cells in the brain. At least, food intake is unaltered in *Th^{Cre}; Npy^{flox/flox}* (**Figure 5L**) and we predicted that the cost of opportunity of creating another conditional knockout would very high - considering the already very dense body of work and the limited space.

R1.6 Are smooth muscle cells and pericytes similarly regulated in obesity and NPY KO mice? There are statements in the paper to the effect that SMCs are a type of mural cell. However, different types of mural cells (i.e. pericytes and SMCs) have specific gene signatures, and these populations should be independently evaluated. E.g. Do SMCs (Myh11⁺) express NPY receptors and also decline in the KO mice?

A: The reviewer has astutely drawn attention to a distinction that we had also contemplated. This prompted us to conduct an exhaustive review of the literature and undertake additional experiments aimed at addressing the reviewer's concerns. Initially, we encountered conflicting literature regarding the identification of specific markers capable of reliably distinguishing pericytes from smooth muscle cells (SMC), and vice versa. Notably, genes such as PDGFR β , NG2 (also known as Cspg4), CD13 (also known as Anpep), Des, and α SMA (also known as Acta2) have commonly been utilized as markers for pericytes. However, it is worth noting that these same genes are also expressed in smooth muscle cells, as evidenced in an excerpt from Table 1 in PMID:21839917. Secondly, a review on mural cells (PMID:21839917) has indicated that there exists no single molecular marker that unequivocally distinguishes pericytes from vascular smooth muscle cells (vSMC), as mentioned in **Rebuttal Only Sentence 1**.

Table 1. Murine Pericyte Markers

Pericyte Marker	Gene Symbol	Examples of Other Cell Types Expressing the Marker	Comments	References
Validated Markers				
PDGFR- β (platelet-derived growth factor receptor-beta)	Pdgfrb	Interstitial mesenchymal cells during development; smooth muscle in the CNS; certain neurons and neuronal progenitors; myofibroblasts; mesenchymal stem cells	Receptor tyrosine kinase; functionally involved in pericyte recruitment during angiogenesis; useful marker for brain pericytes	Lindahl et al., 1997; Winkler et al., 2010
NG2 (chondroitin sulfate proteoglycan 4)	Cspg4	Developing cartilage, bone, muscle; early postnatal skin; adult skin stem cells; adipocytes; vSMCs ; neuronal progenitors; oligodendrocyte progenitors	Integral membrane chondroitin sulfate proteoglycan; involved in pericyte recruitment to tumor vasculature	Ozerdem et al., 2001; Rüter et al., 1993; Huang et al., 2010
CD13 (alanyl (membrane) aminopeptidase)	Anpep	vSMCs ; inflamed and tumor endothelium; myeloid cells; epithelial cells in the kidney, gut	Type II membrane zinc-dependent metalloprotease; useful marker for brain pericytes	Dermietzel and Krause, 1991; Kunz et al., 1994
α SMA (alpha-smooth muscle actin)	Acta2	Smooth muscle ; myofibroblasts; myoepithelium	Structural protein; quiescent pericytes do not express α SMA (e.g., CNS); expression in pericytes is commonly upregulated in tumors and in inflammation	Nehls and Drenckhahn, 1993
Desmin	Des	Skeletal, cardiac, smooth muscle	Structural protein; useful pericyte marker outside skeletal muscle and heart	Nehls et al., 1992

Rebuttal Only Table 1 (excepted from PMID:21839917). Common pericyte markers also mark SMCs in various tissues.

The periendothelial location of pericytes is frequently confused with the periendothelial location of vascular smooth muscle cells (vSMCs), fibroblasts, macrophages, and even epithelial cells. Although the field has generally adopted the view that pericytes belong to the same lineage and category of cells as vSMCs, it should be remembered that there is no single molecular marker known that can be used to unequivocally identify pericytes and distinguish them from vSMCs or other mesenchymal cells. The multiple markers that are commonly applied are neither specific nor stable in their expression.

Rebuttal Only Sentence 1. Excerpt from PMID:21839917 stating that no single marker distinguishes pericytes from SMCs.

Lastly, and for the reviewer's convenience, in **Rebuttal Only Figure 1**, we excerpted images from additional publications showing that SMC markers can label pericytes and vice versa.

Rebuttal Only Figure 1. SMC and pericytes share molecular markers.

(A) Images excerpted from PMID:29561727 showing that α SMA exists in pericytes in the retina. (B) Images excerpted from PMID:29298435 showing that pericytes express Myh11 in the brain. (C) Images excerpted from PMID:33093635 showing that NG2^{Cre}-tdTomato labels SMC in the skeletal muscle.

Despite the challenge of distinguishing between the two subtypes of mural cells based on gene signatures, and in response to the reviewer's concerns, we conducted a reanalysis of our single-cell RNA sequencing datasets, as presented in Figure 2C-D. Our findings revealed a significant co-expression of *Npy1r* with *Acta2* (α SMA), *Tagln*, and *Myh11* in mural cells, demonstrated in Supplementary Figure 4A. To validate these results, we performed immunolabeling experiments on vessels extracted from adipose tissue in *Npy1r^{Cre}; Rosa26^{tdTomato}* mice, using the SMC markers α SMA and TAGLN. These experiments showed that α SMA⁺ cells and TAGLN⁺ cells are all positive for NPY1R, whether they are SMCs surrounding large vessels or pericytes encircling small capillaries, as depicted in Supplementary Figure 4B-C and Rebuttal Only Figure 2. Furthermore, our images in Figure 4C (above) provide additional confirmation of the reduction in TAGLN⁺ mural cells within the adipose tissue of *Th^{Cre}; Npy^{flox/flox} cKO* mice. Due to space constraints, we have opted not to include Rebuttal Only Figure 2 in our revised manuscript. It's worth noting that we utilized TAGLN and α SMA as markers for SMCs due

to the limited availability of reliable anti-MYH11 antibodies—previous research mostly used MYH11-reporter mice to detect MYH11, and the only published good anti-mouse MYH11 antibody that works for IF (PMID:32978500, Clonal ID km3669) has been discontinued.

Supplementary Figure 4. NPY1R-tdTomato is colocalised with NG2 and aSMA in mural cells.

(A) Embedding plot of single nuclear sequencing showing the co-expression of mural markers *Acta2* (aSMA), *Myh11*, and *Tagln* (red) with *Npy1r* (green) (PMID:32755590). Co-expression is in yellow. (B) Confocal images of vessels dissected from adipose tissue of *Npy1r^{Cre}; Rosa26^{tdTomato}* mice stained with anti-aSMA (green), scale bar=100 μ m. (C) Zoomed-in images of vessels as in B, scale bar=20 μ m. (D) Confocal images of vessels dissected from *Npy1r^{Cre}; Rosa26^{tdTomato}* mice stained with anti-aSMA (green) and anti-NG2 (cyan), scale bar=100 μ m. (E) Zoomed-in images of vessels as in D, scale bar=20 μ m.

Rebuttal Only Figure 2 NPY1R-tdTomato is colocalised with TAGLN
Confocal images of vessels dissected from the adipose tissue of *Npy1r^{Cre}; Rosa26^{tdTomato}* immunolabelled with anti-TAGLN (cyan). Scale bars=20 μ m.

R1.7 Fig. 5 Are there any effects of the NPY KO in skeletal muscle? Is the vasculature also altered in skeletal muscle, which could affect whole body metabolism?

*A: The reviewer raises an important question because skeletal muscles control metabolism. In contrast to brown adipose tissue (BAT), we did not observe any macroscopic or functional disparities in the skeletal muscle of *Th^{Cre}; Npy^{flox/flox}* cKO mice. To be precise, we employed MRI to assess the lean mass of these mice and utilized beam break sensors to quantify locomotor activity. Our assessments did not reveal any discernible distinctions in the *Th^{Cre}; Npy^{flox/flox}* cKO mice, as illustrated in **Supplementary Figure 13B-C**.*

Supplementary Figure 13B-C. $Th^{Cre}; Npy^{flox/flox}$ mice have unchanged muscle output.

(B) The percentage of lean body weight (LBM) measured by MiniSpec MRI (n=9). (C) The locomotor activity measured using Multitake cage (n=6).

The main phenotype that we identified was the whitened and dysfunctional BAT, and previous research indicates that blunting beige cell differentiation is sufficient to decrease energy expenditure and promote obesity (PMID:32615086). Moreover, the protocols for clearing muscle (MyoClear, PMID:30873005) are hydrophilic, whereas those that we use to clear fat (AdipoClear) are hydrophobic. This technical contrast may introduce challenges when interpreting our results. Specifically, it may hinder our ability to determine whether alterations in the vasculature of skeletal muscle mirror those observed in adipose tissue to the same degree.

R1.8 Fig. 5. Thermogenic activity/energy expenditure should be assessed in the NPY KO mice. It is unclear if thermogenesis is actually reduced.

A: This is an excellent point, and we have taken steps to address it in collaboration with the laboratory of Professor Licio Velloso. Their research involved measuring the energy expenditure of $Th^{Cre}; Npy^{flox/flox}$ KO mice. As depicted in **Figure 5D**, the data reveals that $Th^{Cre}; Npy^{flox/flox}$ KO mice exhibit lower energy expenditure compared to WT mice during the light phase. Furthermore, we assessed thermogenesis by monitoring BAT temperature using thermo-cameras. After a 14-hour fast, **Figure 5E** demonstrates that the BAT temperature in $Th^{Cre}; Npy^{flox/flox}$ KO mice is notably lower than in WT Th^{Cre} or $Npy^{flox/flox}$ mice. This reduction in thermogenesis translates into a difference in body weight loss following a 14-hour fast, as shown in **Figure 5F**. In summary, these findings collectively indicate that $Th^{Cre}; Npy^{flox/flox}$ KO mice exhibit inefficient thermogenesis. We believe these results effectively address the concerns raised by the reviewer.

Figure 5D-F. $Th^{Cre}; Npy^{flox/flox}$ mice have inefficient thermogenesis and lose less weight during fasting because they burn less.

(D) Daily energy expenditure of 7-week-old male mice measured using an indirect calorimetry system ($n=7$). (E-F) The change in (E) BAT temperature and (F) body weights of 12-week-old mice after a 14-hour fast ($n=6$ for $Th^{Cre}; Npy^{flox/flox}$ mice and $n=3$ for both Th^{Cre} and $Npy^{flox/flox}$ WT mice).

Referee #2 (Remarks to the Author):

R2.1 The manuscript from Zhu and colleagues describes the results from a series of studies investigating the role of the sympathetic nervous system and neuropeptide Y (NPY) in regulating adipocyte precursors from progenitor cells. The studies provide evidence of a previously unidentified role of NPY secretion from sympathetic terminals in the “being” and browning of adipose tissue. The studies are novel and will be of wide interest. The only major concerns are technical and need to be addressed.

A: We thank the reviewer for recognising that our study is novel and of wide interest.

R2.2 The biggest concern is the characterization of the genetic models. In particular, the authors note the overlap of TH and NPY was exclusively in “sympathetic” neurons. While they did assess overlap in VTA and hypothalamus, there are other populations of NPY expressing cells including brainstem neurons, many of which are considered “pre-sympathetic”. i.e., these

neurons are known to project to the spinal cord and to pre-autonomic regulatory sites in the hypothalamus.

A: *We acknowledge the reviewer's important observation and have collaborated with Professor Cheng Zhan, whose expertise lies in the study of NPY within brainstem pre-autonomic TH⁺ neurons (PMID:32822608). Professor Cheng Zhan's research has provided data showing that the removal of NPY from neurons in the hindbrain does not affect body weight in mice on a normal diet (ND). Furthermore, his findings have indicated that the elimination of NPY from the hindbrain using AAV-CRE (Hindbrain: Npy^{flox/flox}; AAV-CRE) does not alter BAT weight or thermogenesis, as demonstrated in **Supplementary Figure 15**. This data supports that the phenotypes we have observed are not a consequence of NPY loss in the hindbrain.*

Supplementary Figure 15. Ablating NPY from the hindbrain does not affect body weight, BAT weight or thermogenesis.

(A) Schematic of AAV-mediated knockout NPY in the hindbrain via injection. (B) Schematic of the experiments on mice with NPY knocked out in the hindbrain (Hindbrain: *Npy*^{lox/lox}; AAV-CRE-GFP) and control (*Npy*^{lox/lox}; AAV-GFP) mice. (C) Confocal images of the sliced hindbrain of Hindbrain: *Npy*^{lox/lox}; AAV-CRE-GFP and WT mice stained with anti-NPY (magenta). Scale bar=100 μ m. (D) Weekly body weight of ND-treated male Hindbrain: *Npy*^{lox/lox}; AAV-CRE-GFP and WT mice (n=7). (E) BAT weight of 14-week-old ND-treated male

Hindbrain: $Npy^{flox/flox}$; AAV-CRE-GFP and WT mice (n=7). (F) BAT temperature change after a 14-hour fast (n=7). (G) The expression levels of thermogenesis genes in the BAT of RT-housed ND-treated male Hindbrain: $Npy^{flox/flox}$; AAV-CRE-GFP and WT mice (n=5).

R2.3 Some TH-cre models have been reported to have a baseline phenotype due to altered catecholamine levels. More detailed about the Cre alone controls would be welcome.

A: We appreciate the reviewer for highlighting this important aspect of our study. We have thoroughly examined the characteristics of BAT in the Th^{Cre} mouse and found that it does not exhibit whitening, as observed when compared to the $Th^{Cre}; Npy^{flox/flox}$ mouse (**Rebuttal Only Figure 3A**). Additionally, it shares the same BAT weight as $Npy^{flox/flox}$ control mice, whereas $Th^{Cre}; Npy^{flox/flo}$ mice possess larger BATs (as depicted in **Rebuttal Only Figure 3B**).

Moreover, our experiments in **Figure 5E-F** were designed to address this concern effectively. These experiments demonstrate that both Th^{Cre} and $Npy^{flox/flox}$ WT mice display superior thermogenic capabilities compared to $Th^{Cre}; Npy^{flox/flox}$ mice. Importantly, there is no statistically significant difference in thermogenic ability between Th^{Cre} and $Npy^{flox/flox}$ WT mice. We trust that this new data alleviates the concerns raised by the reviewer.

Rebuttal Only Figure 3. Th^{Cre} mice don't have a baseline phenotype.
 (A) The picture of one lobe of the BAT of Th^{Cre} mice. (B) Weight of the BATs of Th^{Cre} and $Npy^{flox/flox}$ WT mice and $Th^{Cre}; Npy^{flox/flox}$ KO mice.

E Temperature Change F Body Weight Change

Figure 5E-F. Th^{Cre}; Npy^{flox/flox} mice have inefficient thermogenesis and lose less weight during fasting because they burn less.

(E-F) The change in (E) BAT temperature and (F) body weights of 12-week-old mice after a 14-hour fast ($n=6$ for Th^{Cre}; Npy^{flox/flox} mice and $n=3$ for both Th^{Cre} and Npy^{flox/flox} WT mice).

R2.4. In addition, a bit more information about the floxed NPY mouse could be included in the supplemental data.

A: The NPY-flox mice employed in our experiments were generated following the protocol outlined in **PMID:30522780** and served as our consistent negative control throughout the study. We have accordingly revised the methods section and figure legends to emphasize this point. Unless the reviewer specifies any specific additional information needed, we believe that this adjustment addresses the reviewer's concern effectively.

R2.5. Finally, I prefer that each panel in a figure have a letter label.

A: We thank the reviewer for this suggestion that we followed upon.

Referee #3 (Remarks to the Author):

The manuscript # 2023-02-01744A seeks to address the putative mechanisms by which Neuropeptide Y (NPY) regulates thermogenic adipocytes and obesity. The authors perform a variety of approaches including whole adipose tissue staining and clearing followed by IF staining together with scRNAseq analysis and identify NPY-positive axons to be more closely associated with the vasculature. The scRNAseq data show that mural cells express NPYr1 and that at the protein levels NPYr1 is expressed by capillary pericytes. The authors show that this association is reduced in high fat diet (HFD) adipose tissue. In vitro, NPY seems to affect mural cell differentiation or maturation. Finally, genetic ablation of NPY in sympathetic neurons, whitens the adipose tissue and makes mice more susceptible to high fat diet. Although the authors provide some compelling data by which NPY operating on mural cells may regulate the adipose tissue differentiation, there are several concerns with the paper that need to be addressed to strengthen the mechanism and the premise for the study.

A: We thank the reviewer for recognising that our data is compelling.

Major concerns:

R3.1. While the authors show that NPY is required in sympathetic neurons for brown adipose tissue development, mural cell coverage and diet-induced obesity, the mechanisms are not explored. Does NPY-NPYR1 signaling in mural cells triggers a secondary signaling cascade that leads to changes in adipose tissue? What are these signals? The authors show that some of the markers of pericytes are upregulated in vitro with the addition of PDGF-BB and NPY including Angpt1. Is Angpt1-Tie2 signaling driving this process, or are there additional pathways driving this process? This mechanism needs to be explored in depth and it is currently missing from the paper.

A: We appreciate the reviewer's encouragement to delve deeper into the intracellular mechanism through which NPY influences mural cells. Based on prior research and our in vitro experiments, the primary pathway driving this process appears to be the NPY1R -> Pi3K/MAPK pathway. Existing literature has reported that PI3K and MAPK act downstream of NPY1R activation (PMID:24917132, PMID:25817573), and both Pi3K and MAPK are recognized for their role in promoting cellular proliferation. Our experiments involved treating isolated mural cells with NPY, and the results, as shown in **Figure 3A**, demonstrate that NPY indeed stimulates the proliferation of mural cells. Based on this compelling evidence, we can conclude that NPY sustains mural cells by promoting their proliferation

Figure 3A-B. NPY promotes the proliferation of mural cells in culture.

(A) Confocal images of isolated mural cells treated with or without NPY stained with anti-DES (cyan) and DAPI (blue). EdU (red) was used to indicated proliferation. Scale bar=80 μ m. (B) The percentage of EdU⁺ cells quantified based on images as in A (n=6).

The experiment presented in Figure 4G-K (**Supplementary Figure 8A-E** in the revised manuscript) does not suggest that the combination of PDGF-BB and NPY upregulates pericyte markers. Rather, it indicates that PDGF-BB down-

regulates pericyte markers, while NPY restores the expression of these markers within the cell mixture by increasing the mural cell fraction. We have made revisions to the main text to clarify this point.

In the same figure, *Angpt1* was evaluated as a marker for mural cells. According to the single-cell RNA sequencing dataset, *Tek* (*TIE2*) is only expressed in endothelial cells and not in mural cells, as illustrated in **Rebuttal Only Figure 4** below. Consequently, it is unlikely that the impact of NPY on mural cells is mediated through *ANGPT1-TIE2* signalling.

Rebuttal Only Figure 4. Tek (TIE) is not expressed in mural cells.

The expression of *Tek*, *Angpt1*, and *Angpt2* in different clusters of cells in the SVF of BAT (PMID:33846638).

R3.2. NPYr1 receptor is expressed by all mural cells, including arterial SMCs, venous SMCs, capillary pericytes. Throughout the paper, the authors use only one marker (Desmin) to stain or sort these cells. This is NOT sufficient. The authors need to use additional mural cell markers including PDGFR-beta, NG2, and α -SMA (SMCs). This is critical throughout the manuscript since the focus of the paper is on the mural cells. The current IF images are very low magnification and therefore it is difficult to distinguish the mural cells. There are several mouse strains where mural cells are labelled with eGFP, DsRed (See <https://pubmed.ncbi.nlm.nih.gov/26119027/>). The authors should use these strains to strengthen their findings.

A: We agree with the reviewer that the use of DES isn't sufficient and hence we have used throughout the manuscript multiple markers to identify or isolate mural cells. For instance, in **Figure 2A-D**, we identify the mural cell cluster in scRNA-seq datasets based on the expression of *Des*, *Acta2*, *Myh11*, *Tagln*, *Cspg4* (NG2), and *Pdgfrb*, and in **Figure 2E**, we have used anti-NG2 to sort mural cells and confirmed that *Npy1r* and *Des* are co-expressed in the NG2⁺PDGFR α cells. In addition, in **Supplementary Figure 6A-B**, we have used anti-PDGFR β for magnetically sorting mural cells and proved that these cells highly express *Des*, *Acta2* (aka aSMA) and *Pdgfrb* using qPCR, and they also express *Npy1r* (**Supplementary Figure 8C**) Therefore, we have already confirmed that the NPY1R⁺ mural cells we identified are not only DES⁺, but also aSMA⁺, NG2⁺, and PDGFR β ⁺.

We used DES in IF to label mural cells because the scRNA-Seq datasets demonstrate that DES is a bona fide marker for mural cells and is more specific to mural cells than NG2, which also labels non-mural cells (**Figure 2A-D**).

Figure 2E. PDGFR α NG2⁺ mural cells express NPY1R and DES.

A

B
**Supplementary Figure 6A-B.
Markers expressed by sorted mural cells.**

(A) Schematic of isolating mural cells from adipose tissue. (B) The expression level of mural cell markers Des, Pdgfrb, and Acta2. Red dashed line indicates the expression level of reference gene Hprt.

Furthermore, we stained the iWAT of *Npy1r^{Cre}; R26^{dTomato}* mice for anti- α SMA and NG2 antibodies to demonstrate that both α SMA⁺ and NG2⁺ mural cells express *Npy1r* (shown below in **Supplementary Figure 4**), instead of importing, rederiving and breeding the *NG2^{CreER}R26^{mTomato/mGFP}*, *SMA^{mCherry}* or *SMA^{CreER}R26^{mTomato/mGFP}* reported in **PMID 26119027**.

As per the reviewer's suggestion, we imaged iWAT of *Npy1r^{Cre}; R26^{dTomato}* mice at a higher the magnification for visualizing the typical morphology of the DES⁺ mural cells which is colocalized with NPY1R-tdTomato reporter (*Npy1r^{Cre}; R26^{dTomato}*) (this data is now in **Figure 2G**).

G
Figure 2G. NPY1R marks DES⁺ mural cells.

(G) Confocal images of blood vessels dissected from *Npy1r^{Cre}; Rosa26^{tdTomato}* mice stained against-DES (green), Scale bar=20 μ m.

R3.3. In Figure 1, the image shown in panel C do not reflect the quantification shown in panel F. Can the authors provide more details on how the analysis of vascular innervation is performed using a quantitative unbiased methodology? In that figure, it looks like a lot of NPY sympathetic innervation is around larger caliber vessels (arteries or veins). The authors need to distinguish arteries from

capillaries and veins using established markers and analyse whether NPY axons are associated more with arteries, veins or capillaries.

A: We are pleased to provide additional details regarding the quantification in Figure 1. To clarify, the quantification in panel F of Figure 1 is based on Figure 1D, and this is explained explicitly in the Results section. In Figure 1C, the right-hand panels illustrate that NPY⁺ axons are primarily associated with CD31, whereas TH signals are more pronounced in the axon bundles innervating the parenchyma. To enhance clarity, we have included zoomed-in confocal images in Figure 1D, which offer a more detailed demonstration of this association. For the quantification of images, we followed the methodology outlined under the "Figure Quantification" section in the Methods. Specifically, we utilized a plug-in called JaCoP in Fiji for colocalization analysis, which is an unbiased quantitative approach.

In response to the reviewer's suggestion, we have expanded our analysis to demonstrate that NPY innervation extends beyond large vessels. The updated images in **Supplementary Figure 2A-D** below reveal that NPY exhibits substantial innervation not only around major vessels but also around smaller capillaries. To provide quantitative insight, we measured both vessel diameters and the density of NPY⁺ axons innervating each vascular branch. Our findings indicate that the density of NPY⁺ axons is most prominent around the 5th order of vascular branches (**Supplementary Figure 2D**), while the 8th order of vascular branches do not exhibit NPY innervation (**Supplementary Figure 2B&D**).

The reviewer brought up a crucial question regarding the preference of NPY⁺ axons for either arteries or veins. To differentiate between these two types of blood vessels, we utilized immunolabeling for EPHB4, a distinctive marker for veins, and SOX17, a specific marker for arteries. The comprehensive images of the adipose tissue reveal that there is an absence of NPY⁺ innervation on EPHB4⁺ veins, as demonstrated in **Supplementary Figure 2E**. In contrast, NPY⁺ axons exhibit a high degree of innervation around the arterioles, as illustrated in **Supplementary Figure 2F**.

Supplementary Figure 2. NPY⁺ axons preferentially innervate arteries and arterioles.

(A) Confocal images of cleared adipose tissue stained with anti-CD31 (red) and anti-NPY (cyan), scale bar = 150 μm , and (B) zoomed-in image showing 7th-8th order of vascular branches in A, scale bar = 50 μm . (C-D) The quantifications

of the diameters of vessels and the density of NPY⁺ axons. (E) Light-sheet images of cleared adipose tissue stained with anti-NPY (red) and anti-EPHB4, scale bar = 500 μ m. (F) Confocal images cleared adipose tissue stained with anti-NPY (red) and anti-SOX17 (cyan), scale bar = 100 μ m.

R3.4. In Figure 2, the authors show nicely by scRNAseq data that NPYR1 is in mural cells. The authors need to validate these data by showing NPY1R staining with a mural marker such as Desmin, NG2, PDGR-beta or α -SMA (vSMCs). What is the fraction of mural cells expressing the receptor by IF?

A: We appreciate the reviewer's recognition of our data in Figure 2, which highlights the presence of NPY1R in mural cells. In response to the reviewer's query, we conducted immunofluorescence staining on vessels extracted from the adipose tissue of *Npy1r^{Cre}; Rosa26^{tdTomato}* mice. We specifically targeted DES, α SMA, and NG2 for immunolabeling. As shown in Figure 2G, there is clear colocalization between DES and NPY1R-tdTomato.

Furthermore, the confocal images provided in Supplementary Figure 4B-E offer additional evidence. They demonstrate that all α SMA⁺ NG2⁺ mural cells, including not only smooth muscle cells (SMCs) around large vessels (**Supplementary Figure 4B&D**) but also pericytes around small capillaries (**Supplementary Figure 4C&E**), are indeed labelled by NPY1R-tdTomato. Conversely, NG2⁺ cells that are not mural cells do not exhibit labelling, consistent with our single-cell RNA sequencing data, which indicates that *Cspg4* (NG2) is expressed in NPY1R⁻ cells, as demonstrated by the scRNA-Seq data (**Figure 2A-D**).

Supplementary Figure 4. NPY1R-tdTomato is colocalised with NG2 and aSMA in mural cells.

(A) Embedding plot showing the co-expression of mural markers *Acta2* (aSMA), *Myh11*, and *Tagln* (red) with *Npy1r* (green). Co-expression is in yellow (PMID:32755590). (B) Confocal images of vessels dissected from adipose tissue of *Npy1r^{Cre}; Rosa26^{tdTomato}* mice stained with anti-aSMA (green), scale bar=100 μ m. (C) Zoomed-in images of vessels as in B, scale bar=20 μ m. (D) Confocal images of vessels dissected from *Npy1r^{Cre}; Rosa26^{tdTomato}* mice stained with anti-aSMA (green) and anti-NG2 (cyan), scale bar=100 μ m. (E) Zoomed-in images of vessels as in D, scale bar=20 μ m.

R3.5. In Figure 5A-C, the authors should use an independent marker from NPY to label sympathetic axons to ensure that the differences between ND and HFD are not due to differences in general innervation. How is the axonal - vessel association quantified in this figure? What are the differences in mural cell numbers between ND and HFD?

A: The reviewer's point is indeed valid, and we have taken steps to provide clarification. It's worth noting that a general sympathetic neuropathy induced by a high-fat diet (HFD) has been documented (PMID:32699414), and the loss of NPY is indeed a secondary outcome of this phenomenon.

To assess NPY⁺ innervation of the vasculature, we used the NPY/CD31 ratio, which was automatically calculated using the "Surface" tool of Imaris. Detailed information on this methodology can be found in the "Figure Quantification" section of the Methods.

As for quantifying the actual number of mural cells in adipose tissue, it is technically challenging with IF. To address this concern, we employed flow cytometry to measure the proportion of mural cells in adipose tissue. The results confirmed a reduction in the proportion of mural cells in BAT, iWAT, and vWAT of HFD-treated mice, as shown in **Rebuttal Only Figure 5**.

Rebuttal Only Figure 5. Depletion of adipose mural cells upon diet - induced obesity. The proportion of mural cells (CD45⁻/CD31⁺/PDGFRa⁻/DES⁺) in the SVF of BAT, iWAT, and vWAT measured by flow cytometry.

R3.6. For the in vitro studies with mural cells Fig 5G-K, mural cells change dramatically their transcriptional identity when they are cultured with 10% FBS. Have the authors tried to culture cells with lower serum (2% FBS)? For the in vitro analysis, the authors should perform bulk RNAseq analysis in order to properly analyze the effects of addition of PDGF-BB and NPY on mural cell identity, survival and phenotypic switching. This analysis should be critical for the paper.

A: Yes, we did utilize 2% FBS for the purification of mural cells, as indicated in Supplementary Figure 6A above. The rationale behind using such a low percentage of FBS is to inhibit the growth of endothelial cells, in accordance with established protocols (PMID:31851695, PMID:35226797). In the revised manuscript, we have included a schematic figure outlining the protocol for isolating mural cells.

To further address the reviewer's concerns, we conducted a repeat of this experiment using 2% FBS. The results demonstrate that PDGF-BB diminishes the expression of mural cell markers independently of the FBS percentage, as illustrated in **Rebuttal Only Figure 6**.

Rebuttal Only Figure 6. The Effect of PDGF-BB on mural cells is independent of FBS concentration.

Expression level of *Npy1r* and mural cell markers *Pdgfrb*, *Rgs5*, and *Angpt1* in isolated mural cells cultured with 2% or 10% FBS treated with 250ng/mL PDGF-BB (PMID:27608497).

Hence, it appears unlikely that the alterations observed in the transcriptional identity, as depicted in Figure 5G-K (**Supplementary Figure 8A-E** in the revised manuscript), can be attributed to the presence of 10% FBS. Instead, these changes are more likely a consequence of PDGF-BB, as evidenced by the fact that control cells cultured with 10% FBS did not exhibit a shift in their transcriptional identity.

As the reviewer suggests, bulk-seq has been used in **PMID:27608497** to demonstrate that PDGFBB can change mural cell identity- this effect is revisited in Figure 5G-K (**Supplementary Figure 8A-E** in the revised manuscript).

Furthermore, in order to thoroughly assess the impact of NPY on mural cells, we have now provided additional evidence in **Figure 3A-B**, demonstrating that NPY has the capacity to stimulate the proliferation of mural cells. Therefore, we can conclude that NPY contributes to the sustenance of mural cells by promoting their proliferation, which we hope effectively addresses the reviewer's concerns.

R3.7. Along those lines, it is important to perform single cell RNA seq of mural cells in NPY genetic ablation in sympathetic neurons in order to understand the mechanisms by which this signalling pathway may impact brown adipose tissue formation.

A: *We agree with the reviewer that it is important to determine the mechanism by which NPY affects mural cells and brown adipose tissue formation. We thus collaborated with Shingo Kajimura to investigate this mechanism: first, we isolated mural cells from adipose tissue and conducted additional in vitro experiments to show that NPY promoted the proliferation of mural cells (Figure 3A-B above), which can, in turn, differentiate into thermogenic adipocytes in BAT even under physiological conditions (Figure 2H-I, Supplementary Figure 6A-D, and Schematic Figure 2).*

Figure 2H-I. Thermogenic adipocytes can be lineage-traced to NPY1R⁺ cells.

(H-I) Confocal images of vessels dissected from the BAT of ND-treated, RT-housed 12-week-old *Npy1r^{Cre}; Rosa26^{tdTomato}* mice stained with anti-UCP1 (cyan), Scale bar = 100 μ m. (I) Zoomed-in images of H, scale bar = 50 μ m.

Supplementary Figure 6A-D. Mural cells can differentiate into thermogenic adipocytes

(A) Schematic of isolating mural cells from adipose tissue. (B) The expression level of mural cell markers *Des*, *Pdgfrb*, and *Acta2*. Red dashed line indicates the expression level of reference gene *Hprt*. (C-D) The expression level of (C) thermogenesis genes *Cidea*, *Pgc1a*, and *Prdm16*, and (D) beiging gene *Pparg* in adipocytes differentiated from the stromal-vascular fraction of adipose tissue and isolated mural cells. All values are expressed as mean \pm SEM, * $p < 0.05$, ** $p < 0.01$, *** $p < 0.001$, **** $p < 0.0001$, Student T-tests.

Schematic Figure 2. Schematic of our findings demonstrating that NPY1R⁺ mural cells are a source of beige cells, and that NPY promotes the proliferation of NPY1R⁺ mural cells.

Secondly, we demonstrate that during in vitro differentiation, NPY facilitates beiging by increasing the maximal respiratory capacity (Figure 3C-D).

Figure 3F-G. NPY facilitates beiging in vitro by increasing the maximal respiratory capacity.

(F) Schematic of Kajimura's protocol for differentiating isolated Lin^- , $CD81^+$ progenitors. (G) Oxygen consumption in adipocytes differentiated from $CD81^+$ progenitors with or without NPY ($n=5$). All values are expressed as mean \pm SEM, * $p < 0.05$, ** $p < 0.01$, *** $p < 0.001$, **** $p < 0.0001$, Student T-tests.

Thirdly, our experiments involving $Th^{Cre}; Npy^{flox/flox}$ cKO mice clearly demonstrate that the removal of NPY from sympathetic axons results in a reduction of mural cells, as illustrated in **Figure 4A, C-D, and F**. Consequently, this decrease in mural cells leads to a reduction in the pool of progenitors for thermogenic adipocytes within BAT. Consequently, this depletion contributes to the whitening of BAT and its impaired functionality in thermogenesis, as observed in **Figure 5A-F**.

Figure 4C-D&F. Loss of NPY from sympathetic neuron decreases mural cells.

(C) Light-sheet image of cleared iWAT of 30-week-old, ND-treated mice stained against PV1, a marker of endothelial leakiness (red) and the mural marker TAGLN (cyan). Scale bar=500 μ m. Scale bars of the zoomed-in images are 200 μ m. (D) The percentage of NG2⁺/PDGFR α ⁻ mural cells and BATs, iWATs, and vWATs of 30-week-old, ND-treated mice measured by FACS. (F) The percentage of DES⁺ mural cells in BATs, iWATs, and vWATs of 17-week-old, HFD-treated mice measured by FACS. All values are expressed as mean \pm SEM, * p <0.05, ** p <0.01, *** p <0.001, **** p <0.0001, Student T-tests.

Figure 5A-F. Loss of function of NPY from sympathetic neurons whitens BAT before the onset of obesity, decreases energy expenditure and thermogenesis.

(A) The picture of BATs dissected from ND-treated 12-week-old male WT (top) and *Th^{Cre}; Npy^{flox/flox}* (bottom) mice. The dashed lines encircle the lobes of BATs. (B-C) The expression levels of (B) beiging genes *Pparg* ($n=7&8$), and (C) thermogenesis genes *Prdm16* ($n=8$), *Pgc1a* ($n=7$) and *Cidea* ($n=8$) in the BATs of ND-treated 12-week-old male *Th^{Cre}; Npy^{flox/flox}* mice and WT mice ($n=7$). (D) Daily energy expenditure of 7-week-old male *Th^{Cre}; Npy^{flox/flox}* mice and *Npy^{flox/flox}* WT mice measured using an indirect calorimetry system ($n=7$). (E-F) The change in (E) BAT temperature and (F) body weights of 12-week-old mice after a 14-hour fast ($n=6$ for *Th^{Cre}; Npy^{flox/flox}* mice and $n=3$ for both *Th^{Cre}* and *Npy^{flox/flox}* WT mice).

Lastly, previous studies have established that of NPY-NPY1R activation leads to downstream *Pi3K* and *MAPK* activation (PMID:24917132, PMID:25817573), which were demonstrated to promote cell proliferation (PMID:32466671, PMID:11942415).

To summarize, previous publications and our new experiments demonstrate that NPY sustains the mural progenitor pool of thermogenic adipocytes by promoting its cell proliferation (**Schematic Figure 2** above) via Pi3K-MAPK pathway. We hope this new data and references to literature are sufficient to address the reviewer's concern.

Minor:

R3m.1. Please provide tables for the single cell RNAseq analysis.

A: All the code used for scRNA-seq has been uploaded to GitHub (<https://github.com/PLAVRVSO/scRNA-Seq-analysis-of-BAT-SVF>), and we have uploaded the tables containing the differentially expressed genes in each cluster.

R3m.2 In Fig. S1, what proportion of Th neurons express NPY?

A: We have quantified the percentage of NPY⁺ neurons in TH⁺ neurons (**Supplementary Figure 1F**), to show that about 40% TH⁺ neurons are NPY⁺.

Supplementary Figure 1D-F. Npy is expressed by about 40% of sympathetic neurons.

(D-E) Confocal image of cryo-sectioned (D) superior-cervical ganglion and (E) stellate ganglia of wild type male 8-week-old mice stained with anti-TH (red) and anti-NPY (cyan). Scale bars=100 μm. (F) The percentage of NPY⁺ sympathetic neurons quantified based on images as in D&E (n=5).

R3m.3. In Fig. S5, please increase the number of mice to N=4-5 for the analysis shown in panel A.

A: As per the reviewer's suggestion we have now increased the N to 5-6 (**Supplementary Figure 10A** in the revised manuscript).

Supplementary Figure 10A. The expression of *Npy* and *Th* in SCG is not affected by HFD-induced obesity.

(A) The expression of *Npy* and *Th* in the superior cervical ganglia (SCG, $n=6&5$) of ND- and HFD-treated 17-week-old WT male mice.

R3m.4. The authors show stain for an axonal marker in mice where NPY is deleted in sympathetic neurons to show that innervation of the adipose tissue is not affected (Fig. S7).

A: We have used anti-TH to label sympathetic axons in WAT and BAT, and there is no difference between WT and $Th^{Cre}; Npy^{flox/flox}$ KO mouse (Supplementary Figure 11D-F).

D

Supplementary Figure 11D-F. TH⁺ sympathetic innervation is not altered in the Th^{Cre}; Npy^{flox/flox} KO mouse.

(D) anti-TH (green) and anti-CD31 (red). Scale bar=500 μ m. (E-F) Light-sheet image of cleared BAT of 12-week-old Th^{Cre}; Npy^{flox/flox} mice and Npy^{flox/flox} male mice stained with (E) anti-NPY (cyan), scale bar= 500 μ m, or (F) anti-TH (green), scale bar=200 μ m.

R3m.5. In Fig. S8 the authors show that NPY depletion causes reduction of CD31. How do the authors explain these data?

A: The reviewer brings an interesting point to our attention that we now mention in the discussion section while alluding to literature that explain the effect. Previous research has demonstrated that pericyte/mural cells are necessary for angiogenesis in a pro-inflammatory environment (See **Figure 1** in **PMID:31685607**). Since there is a depletion of mural cells in HFD-treated mice (**Supplementary Figure 9E-F, Rebuttal Only Figure 5**), and because obesity promotes adipose inflammation, the CD31⁺ cells are reduced. We hope this explanation addresses the reviewer's concern.

F **Mural Cell Coverage**

Reviewer Reports on the First Revision:

Referees' comments:

Referee #1 (Remarks to the Author):

The authors provided detailed responses to the reviewers' comments. There are also several new pieces of data, including experiments showing a striking increase in vascular leakiness in Npy KO animals, as assessed by PV1 levels (Fig 4C). However, there remains inadequate support for the central conclusions regarding the selective effect of NPY on brown fat progenitor cells. It is also unclear if the loss of mural cells underlies the whitening of BAT and metabolic effects. Other effects of Npy deficiency such as vascular leakiness, changes in blood pressure, inflammation, etc may underlie the observed phenotypes.

Specific comments:

1. Mural progenitors (PDGFRb+) also give rise to white adipocytes as reported in several papers from the Gupta and Graff labs. How do the authors' explain the selective effects on brown/beige cells?
2. Fig 2. In the NPY1r tracing model- is there contribution of Npy1r+ cells to white adipocytes in WAT depots? Also staining of adipose tissue sections should be shown to visualize a larger area.
3. Figure 3. (related to above points). The expression of adipocyte genes should be evaluated and shown for this experiment (ie. expression levels of adipocyte and thermogenic genes in CD81- and CD81+ cells). What is the molecular phenotype of the cells being assayed for OCR? Is there an equivalent level of adipocyte differentiation in CD81- and CD81+ cells? How do the authors explain the NPY-induced increase in max respiration? Also, was this experiment also performed in BAT-derived cells?
4. Figure 5D. How is there a decrease in energy expenditure with no change in body weight?
5. Figure 5. Is there a decrease in brown/beige fat activity in NPY KO mice upon cold exposure? Fasting reduces brown fat function, so the drop in body temperature under these conditions may not reflect changes in BAT activity.

Referee #2 (Remarks to the Author):

The authors have done a professional job in responding to the previous critiques. They have recruited additional collaborators and have done significantly more experiments. They have addressed all of my previous concerns.

Referee #3 (Remarks to the Author):

In the revised manuscript # 2023-02-01744B, the authors have added a large number of additional experiments and clarifications to address the concerns raised by the reviewers. These new experiments and clarifications have significantly strengthened the manuscript. However, there are still some minor issues that need to be addressed by the authors as follows:

1. Mechanisms of NPY action on mural cells - The authors have shown that NPY increases modestly the proliferation of mural cells since its addition increases the number of EdU+ cells. Based on previous published studies, the authors assume that NPY induces PI3K - MAPK kinase signaling. However, this is not formally tested in Figure 3. The authors need to show that NPY addition changes the phosphorylation of MAPK and perform functional studies to demonstrate the link between NPY - MAPK - Proliferation in mural cells.
2. Demonstration that loss of NPY function from sympathetic neurons increases vascular leakiness (Figure 4) or high fat diet increases vascular leakiness (Supp. Fig. 9). The authors shown in Figure 4 that PV-1 (PLVAP) are increases in the mutant vasculature: however, they have NOT formally tested that the vasculature is leakier. The authors need to inject intravenously tracers (Dextran 70 kDa-TMR) and demonstrate increased leakiness in the adipose tissue in the mutant.
3. In Supp. Figure 2 the authors demonstrate nicely that NPY+ neurons are associated with arteries, but not veins. This association suggests that NPY may regulate contraction of the arteries and arterioles and as a consequence the blood flow in the adipose tissue. This needs to be tested in vivo by the authors since it may have a critical role for blood flow. The authors state in response to Rev #1 concern that a previous study did not show any effect of NPY on blood flow; however, they need to test this effect in their mutant mice or HFD.

Author Rebuttals to First Revision:

REVIEWER #1

The authors provided detailed responses to the reviewers' comments. There are also several new pieces of data, including experiments showing a striking increase in vascular leakiness in Npy KO animals, as assessed by PV1 levels (Fig 4C). However, there remains inadequate support for the central conclusions regarding the selective effect of NPY on brown fat progenitor cells. It is also unclear if the loss of mural cells underlies the whitening of BAT and metabolic effects. Other effects of Npy deficiency such as vascular leakiness, changes in blood pressure, inflammation, etc may underlie the observed phenotypes.

We thank the reviewer for recognising that our results are detailed and striking; we are willing to provide additional information to further mitigate the remaining concerns.

The selective effect of NPY on the beige progenitors is due to the specific expression of Npy1r in beige progenitors. According to our data shown in Figure 3D, Npy1r expression is much higher in the identified progenitors for beige cells compared with the progenitors for white adipocytes. We also observed that only multi-locular UCP1⁺ thermogenic adipocytes can be lineage-traced to NPY1R (Supplementary Figure 6A-B).

We agree with the reviewer that inflammation caused by NPY-loss-induced mural cell loss can also contribute to adipose metabolism, and we are also aware of the fact that inflammation can negatively affect thermogenesis and beiging, but it is technically challenging to decouple mural cells loss and inflammation since there is reciprocal interplay between these two entities: mural cell loss can cause inflammation (PMID:33653955), and inflammation can also cause mural cell loss (PMID:26868537).

Concerning blood pressure, a recent study indicates that there is no effect of sympathetic-neuron-specific NPY-cKO on blood pressure (PMID:38236776). It is thus unlikely that the obesity phenotype of NPY-cKO mice is caused by changes in blood pressure. A reverse causality would be more plausible.

Specific comments:

1. Mural progenitors (PDGFRb⁺) also give rise to white adipocytes as reported in several papers from the Gupta and Graff labs. How do the authors explain the selective effects on brown/beige cells?

We thank the reviewer for bringing up the publications from Gupta and Graff labs, that we now discuss vis-a-vis our results.

In Gupta's paper published in Nature Communication in 2018 (PMID: 29497032), PDGFR β ^{CRE} was used for lineage tracing of adipocytes that protect against pathological visceral expansion. However, as demonstrated in our analysis of scRNA-Sequencing of WAT and BAT (Figure 2A&C of the manuscript), Pdgfrb is also expressed by PDGFR α ⁺ fibroblasts, which were previously demonstrated to give rise to white adipocytes (PMID: 22482730,

PMID:23434825). Thus, the discrepancy noted by the reviewer is explained by lack of perfect overlap between the different markers chosen for lineage-tracing.

Figure 2A&C. Npy1r expression is more specific to mural cells than Pdgfrb, which is also expressed in fibroblasts. Dot plots of single-cell RNAseq showing the expression of *Npy1r*, *Des*, *Myh11*, *Acta2* (aSMA), *Tagln*, *Cspg4* (NG2), *Pdgfrb* (red arrow), *Pdgfra*, and *Pecam1* (CD31) in the stromal vascular fraction (SVF) of (A) iWAT (PMID:32755590) or (B) iBAT (PMID:33846638).

Despite this discrepancy, and consistent with our model, Gupta's PMID 29497032 does not exclude a role of PDGFR β ⁺ preadipocytes in sustaining thermogenesis since data in Figure 6r&q (excerpted below) shows that loss of function of *Pparg* from PDGFR β ⁺ preadipocytes diminished the thermogenic response of vWAT to the PPAR-agonist Rosi.

Figure 6q-r excerpted from PMID: 29497032 showing that PDGFR β ⁺ preadipocytes also sustain thermogenic gene expression.

Graff's paper (PMID: 25437556) used SMA-Cre^{ERT2}; R26R^{RFP} mice for lineage tracing of adipocytes. A-SMA (*Acta2*) is specific to mural cells (Figure 2A&C of the manuscript excerpted below), but their reporter shows some leakiness since their cytometry data and qPCR show the expression of fibroblast marker *Pdgfra* in the RFP labelled cells immediately after TM pulsing (Figure 3G-H of PMID: 25437556 excerpted below), and before the chasing when cells differentiate to adipocytes. This demonstrates that the SMA-Cre^{ERT2} driver is not exclusive to mural cells.

Figure 3G-H excerpted from PMID: 25437556 showing that cells labelled with SMA-RFP (SMA-Cre^{ERT2}; R26R^{RFP}) are not exclusively mural cells. (G and H) FACS analyses (G) and mRNA analyses (H) of denoted genes in FACS-isolated RFP⁺ or RFP⁻ cells from SMA-RFP (SMA-Cre^{ERT2}; R26R^{RFP}) mice immediately after TM pulsing. * $p < 0.01$ RFP⁺ versus RFP⁻ cells. Scale bar, 100 μ m.

We trust that these explanations justify why NPY1R-lineage-tracing is selective for thermogenic pre-adipocytes, and that our new data covered in our reply to comment #2 below can mitigate the reviewer's concern.

2. Fig 2. In the NPY1r tracing model- is there contribution of *Npy1r*⁺ cells to white adipocytes in WAT depots? Also staining of adipose tissue sections should be shown to visualize a larger area.

The reviewer raised an interesting question that we have addressed by following up on the suggestion of further analysing large areas of WAT sections of *Npy1r^{Cre}; Rosa26^{tdTomato}* lineage-tracing model. The answer is no: as shown below in Supplementary Figure 7A-B, we did not observe *tdTomato⁺* unilocular *UCP1⁻* white adipocytes, on paraffin-embedded sections showing an area twice as larger (Supplementary Figure 7A). On the contrary, we detected multi-locular *UCP1⁺* *tdTomato⁺* in iWAT (Supplementary Figure 7A-B). Based on these observations, we reason that the selective effect of NPY on thermogenic adipocyte is due to the selective expression of *Npy1r* in the progenitors of the thermogenic adipocytes, both in WAT and BAT.

Supplementary Figure 7. Only UCP1⁺ multi-locular adipocytes lineage-trace to NPY1R, in WAT and BAT. (A-B) Confocal images of paraffin-embedded slides (scale bars=50 μ m for A and 10 μ m for B) of iBAT and iWAT of 12-week-old male *Npy1r^{Cre}; Rosa26^{tdTomato}* mice stained with UCP1 (Cyan). B is zoom-in of A. 488nm laser-excited autofluorescence (green), which is deviated from *tdTomato* (red) and AF647 (cyan) channels, indicates the presence of adipocytes. White arrows indicate multi-locular *UCP1⁺* adipocytes in iWAT.

3. Figure 3. (related to above points). The expression of adipocyte genes should be evaluated and shown for this experiment (ie. expression levels of adipocyte and thermogenic genes in *CD81⁻* and *CD81⁺* cells). What is the molecular

phenotype of the cells being assayed for OCR? Is there an equivalent level of adipocyte differentiation in CD81⁻ and CD81⁺ cells?

The expression levels of adipocyte and thermogenic genes in CD81⁻ and CD81⁺ fractions that the reviewer is seeking have been demonstrated in Kajimura's Cell paper (**Figure 2B-F** of PMID: 32615086 excepted below). The group of our collaborator showed that CD81⁻ cells also have adipogenic ability, but no thermogenic ability, which is specific to the CD81⁺ fraction.

As per the reviewer's request, we have now included in Rebuttal Only Figure 1 the molecular phenotypes of the cells being assayed for OCR. This data also demonstrates that NPY upregulates the thermogenesis genes without upregulating adipogenesis genes.

Rebuttal Only Figure 1. NPY upregulates thermogenesis but not adipogenesis genes. The expression of thermogenesis and adipogenesis genes in iWAT-derived CD81⁺ progenitors differentiated with or without NPY, n=4.

How do the authors explain the NPY-induced increase in max respiration?

The function of NPY to increase max respiration can be attributed to its ability to promote the proliferation of the progenitors of thermogenic adipocytes, even *in vitro*.

Also, was this experiment also performed in BAT-derived cells?

The experiment was not performed in iBAT-derived CD81⁺ cells because, although BAT depots have CD81⁺ cells, the lineage tracing studies in the past suggests that they do not give rise to thermogenic adipocytes in iBAT (PMID: 32615086, Fig.2 and Sup Fig.2).

4. Figure 5D. How is there a decrease in energy expenditure with no change in body weight?

We have revised the text to emphasize that significant changes in body weight result from decreased energy expenditure. Without a corresponding rise in food intake, the consequences of decreased energy expenditure on body weight accumulate over time, leading to a gradual accumulation of body fat and ultimately culminating in severe obesity. This trend is observable in the Th^{Cre}; Npy^{flox/flox}, whose body weight consistently increases throughout adulthood, as depicted in Fig 5G below.

Figure 5. Loss of function of NPY from sympathetic neurons leads to adult-onset obesity on a normal diet (G) The body weight of ND-treated male Th^{Cre}; Npy^{flox/flox} and Npy^{flox/flox} throughout early adulthood (n=4).

Furthermore, in Figure 5F, the reduced energy expenditure of Th^{Cre}; Npy^{flox/flox} mice is clearly

demonstrated by the lesser amount of weight lost after an overnight fasting period.

F Body Weight Change

Figure 5F. Loss of function of NPY from sympathetic neurons oppose weight-loss after a 14-hour fast.(F) body weights of 12-week-old mice after a 14-hour fast (n=6 for Th^{Cre}; Npy^{flox/flox} mice and n=3 for both Th^{Cre} and Npy^{flox/flox} control mice).

We have edited the main text to make this point clearer and trust that this

mitigates the reviewer's concerns.

5. Figure 5. Is there a decrease in brown/beige fat activity in NPY KO mice upon cold exposure? Fasting reduces brown fat function, so the drop in body temperature under these conditions may not reflect changes in BAT activity.

This is a good point that we have addressed by cold exposure. Our data demonstrates that Th^{Cre}; Npy^{flox/flox} mice have lower rectal temperature under cold exposure, and their iBATs express lower level of thermogenesis genes which, together, indicates decreased BAT activity (Supplementary Figure 14F-G).

F Thermogenesis Gene (8h Cold Challenge)

G Rectal Temperature

Supplementary Figure 14F-G. Loss of NPY from sympathetic neurons decreases cold-induced thermogenesis and iBAT activity. (F) The expression of *Ucp1*, *Pparg*, and *Dio2* in the iBATs of WT and Th^{Cre}; Npy^{flox/flox} mice after 8-hour cold exposure (n=3&4). (G) Rectal temperature of WT and Th^{Cre}; Npy^{flox/flox} mice under cold exposure recorded using a rectal probe every hour (n= 3&4).

Referee #3 (Remarks to the Author):

In the revised manuscript # 2023-02-01744B, the authors have added a large number of additional experiments and clarifications to address the concerns raised by the reviewers. These new experiments and clarifications have significantly strengthened the manuscript. However, there are still some minor issues that need to be addressed by the authors as follows:

We value the reviewer's acknowledgment of our efforts to generate additional data, which have strengthened our manuscript.

1. Mechanisms of NPY action on mural cells - The authors have shown that NPY increases modestly the proliferation of mural cells since its addition increases the number of EdU+ cells. Based on previous published studies, the authors assume that NPY induces PI3K - MAPK kinase signaling. However, this is not formally tested in Figure 3. The authors need to show that NPY addition changes the phosphorylation of MAPK and perform functional studies to demonstrate the link between NPY - MAPK - Proliferation in mural cells.

We now provide functional data demonstrating that, via ERK, the MAPK signalling pathway is required for the effect of NPY on the proliferation of mural cells. Previous studies show that NPY can induce ERK phosphorylation in BAT SVF and smooth muscle cells (Figure 6 of PMID: 18641693 and Figure 4 of PMID: 22374387). Since ERK is an important mediator of the MAPK signalling pathway that regulates proliferation (PMID: 32576977), we functionally addressed the requirement of this pathway by using an ERK inhibitor (PD98059). We demonstrate that PD98059 blocks the effect of NPY to promote mural cell proliferation (Figure 3A-B). We trust that our new data and reference literature mitigates the reviewer's concerns about the link between NPY-MAPK (ERK)-proliferation in mural cells.

Figure 3A-B. NPY promotes mural cell proliferation via ERK pathway. (A) Images of isolated mural cells treated under different conditions stained with DAPI (blue) and anti-DES (cyan), scale bar=200um. EdU staining (red) was used to indicate proliferation. (B) Quantification of images as shown in (A), n=10.

2. Demonstration that loss of NPY function from sympathetic neurons increases vascular leakiness (Figure 4) or high fat diet increases vascular leakiness (Supp. Fig. 9). The authors shown in Figure 4 that PV-1 (PLVAP) are increases in the mutant vasculature: however, they have NOT formally tested that the vasculature is leakier. The authors need to inject intravenously tracers (Dextran 70 kDa- TMR) and demonstrate increased leakiness in the adipose tissue in the mutant.

In our original manuscript, we didn't perform dextran stains because PV1 (with MECA-32 antibody) was already demonstrated elsewhere to be a marker for vascular leakiness (PMID: 7626790, PMID: 34672740, and PMID: 26564856); those studies validated PV1 dextran stains that the reviewer suggests (PMID: 34672740, Figure 2F-G&3A-B).

Notwithstanding the validity of PV1 staining to assess vascular leakiness, we also followed up on the reviewer's suggestion of performing dextran-injections. We show that, relative to CRE-negative control siblings, more Dextran-70kDa leaks to the parenchyma of the adipose tissue of Th^{Cre}; Npy^{flox/flox} mice, which should mitigate the reviewers' concern (Supplementary Figure 13C-D).

Supplementary Figure 13C-D Loss of NPY from sympathetic nervous system increase the vascular leakiness in WAT. (C) intravital two-photon microscopy of epididymal WAT of 23-week-old male Npy^{flox/flox} and Th^{Cre};

Npy^{flox/flox} mice, 12 minutes after i.v. injection of 70kDa Dextran (D) Quantification of the fluorescent intensity ratio between tissue parenchyma and blood vessel based on images as in C (n=5&4).

3. In Supp. Figure 2 the authors demonstrate nicely that NPY+ neurons are associated with arteries, but not veins. This association suggests that NPY may regulate contraction of the arteries and arterioles and as a consequence the blood flow in the adipose tissue. This needs to be tested in vivo by the authors since it may have a critical role for blood flow. The authors state in response to Rev #1 concern that a previous study did not show any effect of NPY on blood flow; however, they need to test this effect in their mutant mice or HFD.

Concerning blood pressure, a recent study indicates that there is no effect of sympathetic-neuron-specific NPY-cKO on blood pressure (PMID:38236776). It is thus unlikely that the obesity phenotype of NPY-cKO mice is caused by changes in blood pressure. Whereas reverse causality would be more plausible, assessing the how blood pressure could eventually change body weight is outside the scope of this study.

Reviewer Reports on the Second Revision:

Referees' comments:

Referee #1 (Remarks to the Author):

The authors have clarified several points in their response to the reviewers' comments. The addition of the age-related body weight and cold-sensitivity phenotypes of the NPY cKO mice strengthen the study.

However, there are remaining issues regarding the central conclusion that reduced beige/brown fat cell differentiation underlies the effects of NPY loss in fat tissues. The paper provides compelling evidence that NPY regulates mural cell survival. The reduction of NPY+ nerves and mural cells in obesity is also very interesting. Deletion of NPY then leads to a marked loss of mural cell coverage and vascular leakiness (Fig 4C). Additionally, there is minimal contribution of NPY1R+ cells to beige or brown adipocytes (under homeostatic conditions) shown by with lineage tracing (Figure S7) – this is with a constitutive cre labeling cells from early development. Together, it seems likely that the loss of vascular integrity, presumably associated with inflammation and fibrosis, is a major contributor to brown fat whitening and dysfunction.

There are also remaining issues with the cell experiments. The conclusion seems to be that NPY supports the proliferation of mural cells, which serve as brown/beige adipocyte precursors. There is a statement to this effect in results: "the upregulation of beiging markers is due to NPY's effect of increasing the proportion of mural cells in the SVF, which are progenitors of thermogenic adipocytes". But, there is no data to support this conclusion. In the NPY incubation experiments that focused on browning (Fig 3F-G), the treatment is started at day 0, presumably when the cells are confluent. Does NPY treatment increase the proportion of mural cells vs. other CD81+ cell types? Also, the rationale for using CD81 as a marker is unclear. What cells are marked/isolated? Is this a mix of Pdgfra+ and mural (Pdgfrb+) cells? The authors should show Cd81 expression alongside other markers in Fig S5. The authors' response to a prior question (3) highlights the data from the Kajimura paper, in which it is very clear (and opposite to the authors' assertion) that CD81+ cells have much higher adipogenic activity than CD81(-) cells.

Finally, there is a disconnect between the in vitro studies (done in inguinal WAT cells), and the in vivo phenotyping assessments of cKO mice. What is the phenotype of the inguinal WAT in the NPY cKO animals? Are there fewer beige adipocytes?

The reply to my previous question (3) regarding whether the in vitro assays were done in BAT-derived cells also seems problematic: "The experiment was not performed in iBAT-derived CD81+ cells because, although BAT depots have CD81+ cells, the lineage tracing studies in the past suggests that they do not give rise to thermogenic adipocytes in iBAT (PMID: 32615086, Fig.2 and Sup Fig.2). If these cells don't contribute in BAT, then how could dysfunction in this pathway be a major driver of the whitened phenotype?

Overall, there are some very interesting and intriguing elements in this paper, but some of the main conclusions are not well supported.

Minor point: Pparg should not be defined as a “being” gene in figures and results. Pparg is an essential adipogenic gene for all types of fat cells.

Referee #2 (Remarks to the Author):

The authors have added significant experimental support of their proposed model. No further comments.

Referee #3 (Remarks to the Author):

In the revised manuscript # 2023-02-01744C, the authors have performed some of the experiments to address my concerns with the previous submission. However, there are still some minor issues as follows:

1) In Figure 3A, B, they demonstrate that addition of an ERK inhibitor (PD98059) inhibits mural cell proliferation driven by NYP. This is a nice experiment. However, the authors need to include the dose of NYP and the ERK inhibitor (PD98059) used for the in vitro experiments to measure the effects of NYP on mural cell proliferation (EdU+ cells).

2) Second the authors have performed in vivo imaging of Dextran 70kDa-FITC leakage in the epididymal WATs of 23-week-old male Npyflox/flox and ThCre; Npyflox/flox mice 12 minutes after i.v. injection of 70kDa Dextran at the orbital plexus. The effect with the Dextran 70kDa looks smaller compared to PV-1 staining considering that this is a larger MW tracer. Where are the controls showing that the authors added the same amount of tracer in the blood stream? The experiment needs to be explained better in the results section (page 7, line 265) and discussed briefly in the paper.

3) The authors did not understand my third concern. It has nothing to do with measurements of blood pressure, but with blood flow. The authors can measure by looking at the speed of erythrocytes from the in vivo two photon imaging movies (<https://www.sciencedirect.com/science/article/pii/S1053811909008702>). Does the blood flow change in mice? This is critical to understand some of the phenotypes.

Author Rebuttals to Second Revision:

Referees' comments:

Referee #1 (Remarks to the Author):

The authors have clarified several points in their response to the reviewers' comments. The addition of the age-related body weight and cold-sensitivity phenotypes of the NPY cKO mice strengthen the study.

A: We thank the reviewer for agreeing that the additional data we provided in our revised manuscript and our last reply to the reviewers strengthened the study.

However, there are remaining issues regarding the central conclusion that reduced beige/brown fat cell differentiation underlies the effects of NPY loss in fat tissues. The paper provides compelling evidence that NPY regulates mural cell survival. The reduction of NPY⁺ nerves and mural cells in obesity is also very interesting. Deletion of NPY then leads to a marked loss of mural cell coverage and vascular leakiness (Fig 4C). Additionally, there is minimal contribution of NPY1R⁺ cells to beige or brown adipocytes (under homeostatic conditions) shown by with lineage tracing (Figure S7) – this is with a constitutive Cre labeling cells from early development. Together, it seems likely that the loss of vascular integrity, presumably associated with inflammation and fibrosis, is a major contributor to brown fat whitening and dysfunction.

A: We do agree that the effect of NPY-cKO on the function of iBAT could be due to multiple factors, and we will amend the manuscript to include such caveat. However, it would be challenging to determine which one is the main factor contributing to the observed phenotype since vascular leakiness is closely linked to mural cell function, so manipulating one without affecting the other would be technically impossible.

There are also remaining issues with the cell experiments. The conclusion seems to be that NPY supports the proliferation of mural cells, which serve as brown/beige adipocyte precursors.

A: Yes, that is the conclusion of our study.

(1) There is a statement to this effect in results: "the upregulation of beige markers is due to NPY's effect of increasing the proportion of mural cells in the SVF, which are progenitors of thermogenic adipocytes", but there is no data to support this conclusion. In the NPY incubation experiments that focused on browning (Fig 3F-G), the treatment is started at day 0, presumably when the cells are confluent. Does NPY treatment increase the proportion of mural cells vs. other CD81⁺ cell types?

A: We agree that this conclusion needs direct verifying, and we have isolated SVF and CD81⁺ cells and treat the with NPY to check if NPY increase the proportion of mural cells using flow cytometry. The result confirmed that treating

isolated iWAT SVF or CD81⁺ cells with 1 μ M NPY for 6 days starting from the day when cells are 90% confluent increased the proportion of α SMA⁺ mural cells in SVF or CD81⁺ cells (Rebuttal Figure 1). We hope this result would be sufficient to support our conclusion

Rebuttal Figure 1. NPY increases the proportion of mural cells in iWAT SVF and CD81⁺ fraction.

(A) Schematic of isolating iWAT SVF and CD81⁺ cells for NPY treatment experiments and flow cytometry analysis. (B-C) The percentage of α SMA⁺ mural cells in (B) SVF and (C) CD81⁺ cells after incubating with or without 1 μ M NPY. (D) Schematic showing NPY treatment increases the percentage of mural cells in cultured SVF or CD81⁺ cells, and mural cells are the progenitor of thermogenic adipocytes. All values are expressed as mean \pm SEM, * p <0.05, ** p <0.01, *** p <0.001, **** p <0.0001, Student T-tests.

(2) Also, the rationale for using CD81 as a marker is unclear. What cells are marked/isolated? Is this a mix of Pdgfra⁺ and mural (Pdgfrb⁺) cells? The authors should show Cd81 expression alongside other markers in Fig S5. The authors' response to a prior question (3) highlights the data from the Kajimura paper, in which it is very clear (and opposite to the authors' assertion) that CD81⁺ cells have much higher adipogenic activity than CD81(-) cells.

A: The rationale for using CD81⁺ as a marker for beige progenitors is that Lin⁻ (CD31⁻/CD45⁻) CD81⁺ cells in iWAT were previously identified as the progenitors of beige cells (PMID: 32615086), and further analysis of these cells

indicate that they highly express mural cell markers and *Npy1r*, while the CD81⁻ cells do not, which links our study to previous findings and helps strengthen our conclusion that NPY1R⁺ mural cells are progenitors of beige cells. Here we also provide a new dot plot showing the expression of Cd81 together with other markers in different clusters of cells in iWAT. This plot confirms that CD81⁺ cells are a mix of Pdgfra⁺ and Pdgfrb⁺ cells, and NPY1R⁺ mural cells are a subgroup of Lin⁻ (CD31⁻/CD45⁻) CD81⁺ cells (Rebuttal Figure 2).

Rebuttal Figure 2. Cd81 is expressed in NPY1R⁺ mural cells

Dot plot showing the expression of *Npy1r*, *Des*, *Myh11*, *Acta2* (α SMA), *Tagln*, *Cspg4* (NG2), *Pdgfrb*, *Pdgfra*, *Pecam1* (CD31), and *Cd81* in the stromal vascular fraction (SVF) of iWAT.

The authors do not perceive any contradiction with Kajumura's results in PMID: 32615086. Moreover, the relative adipogenic capability of CD81⁻ vs CD81⁺ fractions has already been examined in Kajimura's paper, which lies beyond the scope of our study. Our research concentrates on how sympathetic-derived NPY sustains previously identified progenitors of beige adipocytes.

(3) Finally, there is a disconnect between the *in vitro* studies (done in inguinal WAT cells), and the *in vivo* phenotyping assessments of cKO mice. What is the phenotype of the inguinal WAT in the NPY cKO animals? Are there fewer beige adipocytes?

A: We agree with the reviewer that *in vivo* assessment of iWAT is needed. Therefore, we measured the expression of beige markers in iWATs of mice cold-challenged for 8 hours and found that NPY-cKO downregulated *Prdm16*, the key beige marker (Rebuttal Figure 3). This result is consistent with the *in vitro* experiment now shown in Supplementary Figure 11c-d and confirms that NPY is required for the beigeing of iWAT.

Rebuttal Figure 3. NPY-cKO inhibits beiging of iWAT.

The expression of *Prdm16* in the iWATs of 20-week-old male *Npy*^{flox/flox} and *Th*^{Cre}; *Npy*^{flox/flox} mice after 8-hour cold exposure (n=3).

Supplementary Figure 11c-d. NPY facilitates the neogenesis of thermogenic adipocytes from previously identified beige progenitors (PMID: 32615086) in iWAT.

(c) Schematic of differentiating beige adipocyte progenitors (*Lin*⁻/*CD81*⁺) isolated from iWAT. (d) The expression of thermogenic and adipogenic genes in adipocytes differentiated from iWAT beige adipocyte progenitors with or without 1 μM NPY (n=4). (e) Oxygen consumption in adipocytes differentiated from beige adipocyte progenitors (*Lin*⁻/*CD81*⁺) with or without NPY (n=5).

The reply to my previous question (3) regarding whether the in vitro assays were done in BAT-derived cells also seems problematic: “The experiment was not performed in iBAT-derived CD81⁺ cells because, although BAT depots have CD81⁺ cells, the lineage tracing studies in the past suggests that they do not give rise to thermogenic adipocytes in iBAT (PMID: 32615086, Fig.2 and Sup Fig.2).

If these cells don't contribute in BAT, then how could dysfunction in this pathway be a major driver of the whitened phenotype?

A: We now further clarify this point. Previous research indicates that CD81 is not a marker for the progenitors of thermogenic adipocytes in iBAT (PMID: 32615086, Fig.2 and Sup Fig.2), which is consistent with the fact that *Cd81* is not highly expressed in the NPY1R⁺ mural cells in iBAT (Rebuttal Figure 4). We confirmed that NPY1R⁺ mural cells in iBAT are progenitors of thermogenic adipocytes using NPY1R-lineage-tracing which demonstrates that a subgroup of UCP1⁺ thermogenic adipocytes in iBAT can be lineage-traced to NPY1R⁺

mural cells (Figure 2H-I). Furthermore, we isolated mural cells from iBAT and differentiated them *in vitro*. We thus confirm that iBAT mural cells can differentiate into thermogenic adipocytes (Rebuttal Figure 5).

These data demonstrates that NPY1R⁺ mural cells are progenitors of thermogenic adipocytes in iBAT. Since NPY sustains these NPY1R⁺ mural cells by promoting their proliferation (Rebuttal Figure 6D-F), loss of NPY in sympathetic neurons depletes mural progenitors of thermogenic adipocytes, decreases the thermogenic ability of iBAT (Rebuttal Figure 7A), and results in the iBAT whitening phenotype. We hope our explanation and new data mitigate the reviewer’s concerns.

Rebuttal Figure 4. Cd81 is not highly expressed in NPY1R⁺ mural cells in iBAT.

Dot plot showing the expression of Cd81, mural cell markers, and fibroblast markers in different clusters of cells in iBAT.

Figure 2. Thermogenic adipocytes in iBAT can be lineage-traced to NPY1R⁺ mural cells.

(H-I) Confocal images of vessels dissected from the iBAT of ND-treated, RT-housed 12-week-old Npy1r^{Cre}; Rosa26^{tdTomato} mice stained with anti-UCP1 (cyan), Scale bar=100 μm. (I) Zoomed-in images of (H), scale bar=50μm.

Rebuttal Figure 5. iBAT mural cells can differentiate into thermogenic adipocytes.

(A) The expression of *Ucp1*, *Cidea*, *Pgc1a*, and *Pparg* of differentiated SVF and mural cells isolated from iBAT (n=5&3).

Rebuttal Figure 6. NPY promotes proliferation of mural cells in both iWAT and iBAT.

(A) Schematic of isolating mural cells from iWAT. (B) Confocal image showing iWAT mural cells treated with or without 1 μ M NPY stained with DES (cyan) and DAPI, and EdU indicates proliferating cells. Scale bars=80 μ m. (C) The percentage of EdU⁺ mural cells quantified based on images as in (B) (n=10).

(D) Schematic of isolating mural cells from iBAT. (E) Confocal image showing iBAT mural cells treated with or without 1 μ M NPY stained with DES (cyan) and DAPI, and EdU indicates proliferating cells. Scale bars=80um. (F) The percentage of EdU⁺ mural cells quantified based on images as in (E) (n=3). All values are expressed as mean \pm SEM, *p<0.05, **p<0.01, ***p<0.001, ****p<0.0001, Student T-tests.

Rebuttal Figure 7. Loss of function of NPY in sympathetic neurons decreases both iBAT thermogenesis and iWAT beiging after and 8-hour cold challenge.

(A-B) The expression of thermogenic and adipogenic genes in (A) iBATs (n=3&4) and (B) iWATs (n=3) of cold-challenged *Npy*^{flox/flox} and *Th*^{Cre}; *Npy*^{flox/flox} mice. All values are expressed as mean \pm SEM, *p<0.05, **p<0.01, ***p<0.001, ****p<0.0001, Student T-tests.

Overall, there are some very interesting and intriguing elements in this paper, but some of the main conclusions are not well supported.

Minor point: *Pparg* should not be defined as a “beiging” gene in figures and results. *Pparg* is an essential adipogenic gene for all types of fat cells.

A: We thank the reviewer for this advice and have edited the figures and the text.

Referee #3 (Remarks to the Author): In the revised manuscript # 2023-02-01744C, the authors have performed some of the experiments to address my concerns with the previous submission. However, there are still some minor issues as follows:

1) In Figure 3A, B, they demonstrate that addition of an ERK inhibitor (PD98059) inhibits mural cell proliferation driven by NYP. This is a nice experiment. However, the authors need to include the dose of NYP and the ERK inhibitor (PD98059) used for the in vitro experiments to measure the effects of NYP on mural cell proliferation (EdU⁺ cells).

A: We thank the reviewer for this suggestion, and we have included the concentration of NPY (1 μ M) and PD98059 (2 μ M) in the figures and figure legends.

2) Second the authors have performed in vivo imaging of Dextran 70kDa-FITC leakage in the epididymal WATs of 23-week-old male *Npy*^{flox/flox} and *Th*^{Cre};

Npyflox/flox mice 12 minutes after i.v. injection of 70kDa Dextran at the orbital plexus. The effect with the Dextran 70kDa looks smaller compared to PV-1 staining considering that this is a larger MW tracer. Where are the controls showing that the authors added the same amount of tracer in the blood stream? The experiment needs to be explained better in the results section (page 7, line 265) and discussed briefly in the paper.

A: We thank the reviewer for guidance, and now we have amended the manuscript to provide better explanations of this experiment. The 2-photon microscopic images showing the result of this experiment demonstrate that Dextran intensity in the blood vessels is comparable between Npy^{flox/flox} and Th^{Cre}; Npy^{flox/flox} mice (Supplementary Figure 17e-f), which indicates that the same amount of dye was i.v. injected. Rebuttal Figure 8 also demonstrates that shortly after i.v. injection of Dextran, the fluorescent intensity in vessels is the same between between Npy^{flox/flox} and Th^{Cre}; Npy^{flox/flox} mice, which means same amount of Dextran was injected in WT and NPY-cKO mice. The quantification shown in Supplementary Figure 17f was normalised based on the fluorescent intensity in vessels, which can solve any inaccuracy that might be caused by slightly different amount of dye in the microvasculature of adipose tissue. We hope these explanations could mitigate the reviewer's concerns.

Supplementary Figure 17e-f Loss of NPY from sympathetic nervous system increase the vascular leakiness in WAT. (e) intravital two-photon microscopy of epididymal WAT of 23-week-old male Npy^{flox/flox} and Th^{Cre}; Npy^{flox/flox} mice, 12 minutes after i.v. injection of 70kDa Dextran (f) Quantification of the fluorescent intensity ratio between tissue parenchyma and blood vessel based on images as in (e) (n=5&4).

Rebuttal Figure 8. Same amount of Dextran was i.v. injected in WT and NPY-cKO mice.

(A) Dextran intensity in vessel over time measured with a 2-photon microscope. Arrow indicates the time when Dextran was injected. (B) Dextran intensity in the vessels of eWAT right after the stabilization of Dextran signal (n=5&4). All values are expressed as mean \pm SEM, *p<0.05, **p<0.01, ***p<0.001, ****p<0.0001, Student T-tests.

3) The authors did not understand my third concern. It has nothing to do with measurements of blood pressure, but with blood flow. The authors can measure by looking at the speed of erythrocytes from the in vivo two photon imaging movies

(<https://www.sciencedirect.com/science/article/pii/S1053811909008702>).

Does the blood flow change in mice? This is critical to understand some of the phenotypes.

A: We thank the reviewer for clarifying his/her concern. Now we have added a new measurement that was obtained using the raw Dextran-70kDa intensity data. Blood flow was calculated as the change of Dextran intensity inside the artery of adipose tissue over 100 seconds right after the stabilization of the Dextran signal following administration. As shown in Rebuttal Figure 9A, there is no significant blood flow difference between the two genotypes. This result confirms that the decreased thermogenesis and increased vascular leakiness observed in NPY-cKO mice is not due to significant decreases in blood flow.

Rebuttal Figure 9. Ablating NPY from sympathetic neurons does not affect blood flow in adipose tissue.

(A) Blood flow calculated using the average of the raw intensity of Dextran inside the artery over 100 seconds right after the stabilization of the Dextran signal following administration. (n=5&4). All values are expressed as mean \pm SEM, *p<0.05, **p<0.01, ***p<0.001, ****p<0.0001, Student T-tests.

Reviewer Reports on the Third Revision:

Referees' comments:

Referee #1 (Remarks to the Author):

The authors have addressed the remaining reviewer comments. The paper provides novel and important advances for the field.

Referee #3 (Remarks to the Author):

The authors have addressed the concerns raised by both Reviewer # 1 and #3 in the second round of revisions both at experimental level and modifications to the manuscript. The complete set of main and supplementary data in the revised manuscript provides a strong compelling evidence for the conclusions of the paper. The revised manuscript also addresses some of the concerns raised by reviewers and limitations of the study. I have no further requests for the authors.